# Attention Approximates Sparse Distributed Memory

**Trenton Bricken**
Systems, Synthetic and Quantitative Biology
Harvard University
trentonbricken@g.harvard.edu

**Cengiz Pehlevan**
Applied Mathematics
Harvard University
cpehlevan@seas.harvard.edu

## Abstract

While Attention has come to be an important mechanism in deep learning, there remains limited intuition for why it works so well. Here, we show that Transformer Attention can be closely related under certain data conditions to Kanerva's Sparse Distributed Memory (SDM), a biologically plausible associative memory model. We confirm that these conditions are satisfied in pre-trained GPT2 Transformer models. We discuss the implications of the Attention-SDM map and provide new computational and biological interpretations of Attention.

## Introduction

Used most notably in the Transformer, Attention has helped deep learning to arguably approach human level performance in various tasks with larger models continuing to boost performance [1, 2, 3, 4, 5, 6, 7, 8, 9]. However, the heuristic motivations that produced Attention leave open the question of why it performs so well [1, 10]. Insights into why Attention is so effective would not only make it more interpretable but also guide future improvements.

Much has been done to try and explain Attention's success, including work showing that Transformer representations map more closely to human brain recordings and inductive biases than other models [11, 12]. Our work takes another step in this direction by showing the potential relationship between Attention and biologically plausible neural processing at the level of neuronal wiring, providing a novel mechanistic perspective behind the Attention operation. This potential relationship is created by showing mathematically that Attention closely approximates Sparse Distributed Memory (SDM).

SDM is an associative memory model developed in 1988 to solve the "Best Match Problem", where we have a set of memories and want to quickly find the "best" match to any given query [13, 14]. In the development of its solution, SDM respected fundamental biological constraints, such as Dale's law, that synapses are fixed to be either excitatory or inhibitory and cannot dynamically switch (see Section 1 for an SDM overview and [13] or [15] for a deeper review). Despite being developed independently of neuroanatomy, SDM's biologically plausible solution maps strikingly well onto the cerebellum [13, 16].[1]

Abstractly, the relationship between SDM and Attention exists because SDM's read operation uses intersections between high dimensional hyperspheres that approximate the exponential over sum of exponentials that is Attention's softmax function (Section 2). Establishing that Attention approximates SDM mathematically, we then test it in pre-trained GPT2 Transformer models [3] (Section 3) and simulations (Appendix B.7). We use the Query-Key Normalized Transformer variant [22] to directly show that the relationship to SDM holds well. We then use original GPT2 models to help confirm this result and make it more general.

---

[1]This cerebellar relationship is additionally compelling by the fact that cerebellum-like neuroanatomy exists in many other organisms including numerous insects (eg. the Drosophila Mushroom Body) and potentially cephalopods [17, 18, 19, 20, 21].

35th Conference on Neural Information Processing Systems (NeurIPS 2021).

Using the SDM framework, we are able to go beyond Attention and interpret the Transformer architecture as a whole, providing deeper intuition (Section 4). Motivated by this mapping between Attention and SDM, we discuss how Attention can be implemented in the brain by summarizing SDM's relationship to the cerebellum (Section 5). In related work (Section 6), we link SDM to other memory models [23, 24], including how SDM is a generalization of Hopfield Networks and, in turn, how our results extend work relating Hopfield Networks to Attention [25, 26]. Finally, we discuss limitations, and future research directions that could leverage our work (Section 7).

# 1   Review of Kanerva's SDM

Here, we present a short overview of SDM. A deeper review on the motivations behind SDM and the features that make it biologically plausible can be found in [13, 15]. SDM provides an algorithm for how memories (patterns) are stored in, and retrieved from, neurons in the brain. There are three primitives that all exist in the space of $n$ dimensional binary vectors:

Patterns ($\mathbf{p}$) - have two components: the pattern address, $\mathbf{p}_a^\mu \in \{0,1\}^n$, is the vector representation of a memory; the pattern "pointer", $\mathbf{p}_p^\mu \in \{0,1\}^n$, is bound to the address and points to itself when autoassociative or to a different pattern address when heteroassociative. A heteroassociative example is memorizing the alphabet where the pattern address for the letter $a$ points to pattern address $b$, $b$ points to $c$ etc. For tractability in analyzing SDM, we assume our pattern addresses and pointers are random. There are $m$ patterns and they are indexed by the superscript $\mu \in \{1, \ldots, m\}$.

Neurons ($\mathbf{x}$) - in showing SDM's relationship to Attention it is sufficient to know there are $r$ neurons with fixed addresses $\mathbf{x}_a^\tau \in \{0,1\}^n$ that store a set of all patterns written to them. Each neuron will sum over its set of patterns to create a superposition. This creates minimal noise interference between patterns because of the high dimensional nature of the vector space and enables all patterns to be stored in an $n$ dimensional storage vector denoted $\mathbf{x}_v^\tau \in \mathbb{Z}_+^n$, constrained to the positive integers. Their biologically plausible features are outlined in [13, 15]. When we assume our patterns are random, we also assume our neuron addresses are randomly distributed. Of the $2^n$ possible vectors in our binary vector space, SDM is "sparse" because it assumes that $r \ll 2^n$ neurons exist in the space.

Query ($\boldsymbol{\xi}$) - is the input to SDM, denoted $\boldsymbol{\xi} \in \{0,1\}^n$. The goal in the Best Match Problem is to return the pattern pointer stored at the closest pattern address to the query. We will often care about the maximum noise corruption that can be applied to our query, while still having it read out the correct pattern. An autoassociative example is wanting to recognize familiar faces in poor lighting. Images of faces we have seen before are patterns stored in memory and our query is a noisy representation of one of the faces. We want SDM to return the noise-free version of the queried face, assuming it is stored in memory.

SDM uses the Hamming distance metric between any two vectors defined: $d(\mathbf{a}, \mathbf{b}) := \mathbb{1}_n^T |\mathbf{a} - \mathbf{b}|$. The all ones vector $\mathbb{1}_n$ is of $n$ dimensions and $|\mathbf{a} - \mathbf{b}|$ takes the absolute value of the element-wise difference between the binary vectors. When it is clear what two vectors the Hamming distance is between, we will sometimes use the shorthand $d_v := d(\mathbf{a}, \mathbf{b})$.

The Hamming distance is crucial for determining how many neurons read and write operations are distributed across. The optimal Hamming distance for the read and write circles denoted $d^*$, depends upon the number and distribution of patterns in the vector space and what the memories are being used for (e.g. maximizing the number of memories that can be stored versus the memory system's robustness to query noise). We provide three useful reference $d^*$ values, using equations outlined in Appendix B.5. The Signal-to-Noise Ratio (SNR) optimal $d_{\text{SNR}}^*$ maximizes the probability a noise-free query will return its target pattern [15]. The memory capacity optimal $d_{\text{Mem}}^*$ maximizes the number of memories that can be stored with a certain retrieval probability and also assumes a noise-free query. The critical distance $d_{\text{CD}}^*$ maximizes, for a given number of patterns, the amount of noise that can be applied to a query such that it will converge to its correct pattern [15].

These $d^*$s are only approximate reference points for later comparisons to Transformer Attention, first and foremost because they assume random patterns to make their derivations tractable. In addition, Transformer Attention will not be optimizing for just one of these objectives, and likely interpolates between these optimal $d^*$s as it wants to have both a good critical distance to handle noisy queries and a reasonable memory capacity. These optimal $d^*$ are a function of $n$, $r$ and $m$. For the Transformer

Attention setting [1], where $n = 64$, $r = 2^n$ and $m \leq 1024$, $d^*_{\text{SNR}} = 11$, $d^*_{\text{Mem}} = 5$, $d^*_{\text{CD}} = 15$, as derived in Appendix B.5.

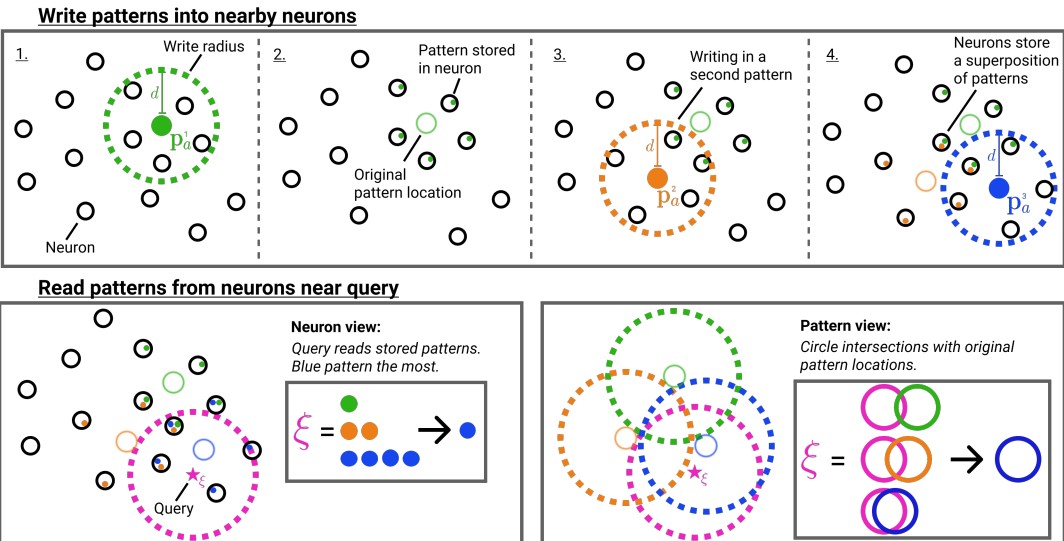

Figure 1: Summarizing the SDM read and write operations. **Top Row** three patterns being written into nearby neurons. 1. The first write operation; 2. Patterns are stored inside nearby neurons and the original pattern location is shown; 3. Writing a second pattern; 4. Writing a third pattern and neurons storing a superposition of multiple patterns. **Bottom Row** shows two isomorphic perspectives of the read operation. Neuron view (left) shows the query reading from nearby neurons with the inset showing the number of times each pattern is read. The four blue patterns are a majority which would result in one step convergence. Pattern view (right) is crucial to relating SDM to Attention and defined in Eq. 1 below. We abstract away the neurons by assuming they are uniformly distributed through the space. This allows us to consider the circle intersection between the query and the original locations of each pattern where blue has the largest circle intersection.

## 1.1   SDM Read Operation

For the connection to Attention we focus on the SDM read operation and briefly summarize the write operation: all patterns write their pointers $\mathbf{p}_p$ in a distributed fashion to all neuron addresses located within Hamming distance $d$. This means that each neuron will store a superposition of pattern pointers from those pattern addresses within $d$: $\mathbf{x}_v^\tau = \sum_{\{\mathbf{p}:d(\mathbf{p}_a^\mu, \mathbf{x}_a^\tau) \leq d, \forall \mu\}} \mathbf{p}_P$. Having stored patterns in a distributed fashion across nearby neurons, SDM's read operation retrieves stored pattern pointers from all neurons within distance $d$ of the query and averages them. This average is effectively weighted because the same patterns have distributed storage across multiple neurons being read from. The pattern weighting will be higher for those patterns with addresses nearer the query because they have written their pointers to more neurons the query reads from. Geometrically, this weighting of each pattern can be interpreted as the intersection of radius $d$ circles[2] that are centered on the query and each pattern address $\mathbf{p}_a^\mu$ for all $\mu$. A high level overview of the SDM read and write operations is shown in Fig. 1.

The $2^n$ possible neurons that have both stored this pattern's pointer $\mathbf{p}_p$ *and* been read by $\boldsymbol{\xi}$ is: $|O_n(\mathbf{p}_a, d) \cap O_n(\boldsymbol{\xi}, d)|$, where $|\cdot|$ is the cardinality operator and $O_n(\boldsymbol{\xi}, d) = \{\mathbf{x}_a \in \{0,1\}^n : d(\boldsymbol{\xi}, \mathbf{x}_a) \leq d\}$ is the set of all possible neuronal addresses $\mathbf{x}_a$ within radius $d$ of $\boldsymbol{\xi}$. Mathematically,

---

[2]In this binary space, the Hamming distance around a vector is in fact a hypercube but the vertices of an $n$ dimensional unit cube lie on the surface of an $n$ dimensional sphere with radius $\sqrt{n}/2$ and we refer to this as a circle because of our two dimensional diagrams. We adopt this useful analogy, taken from Kanerva's book on SDM [13], throughout the paper.

SDM's read operation sums over each pattern's pointer, weighted by its query circle intersection:

$$\boldsymbol{\xi}^{new} = g\left(\frac{\sum_{\mathbf{p}\in P}|O_n(\mathbf{p}_a,d)\cap O_n(\boldsymbol{\xi},d)|\mathbf{p}_p}{\sum_{\mathbf{p}\in P}|O_n(\mathbf{p}_a,d)\cap O_n(\boldsymbol{\xi},d)|}\right), \qquad g(e) = \begin{cases} 1, & \text{if } e > \frac{1}{2}, \\ 0, & \text{else} \end{cases} \tag{1}$$

and $g$ acts elementwise on vectors. The denominator normalizes all of the weights so they sum to 1 in the numerator and enables computing if the element-wise majority value is a 0 or 1, using the function $g(\cdot)$. Intuitively, the query will converge to the nearest "best" pattern because it will have the largest circle intersection weighting. The output of the SDM read operation is written as updating the query $\boldsymbol{\xi} \rightarrow \boldsymbol{\xi}^{new}$ so that it can (but is not required to) apply the read operation iteratively if full convergence to its "best match" pattern is desired and was not achieved in one update.

The circle intersection (derived in Appendix B.1) is calculated as a function of the Hamming radius for the read and write operations $d$, the dimensionality $n$, and the vector distance between the query and pattern: $d_v = d(\mathbf{p}_a, \boldsymbol{\xi})$, so we use the shorthand $\mathcal{I}(d_v, d, n)$:

$$\mathcal{I}(d_v, d, n) := |O_n(\mathbf{p}_a, d) \cap O_n(\boldsymbol{\xi}, d)| = \sum_{a=n-d-\lfloor\frac{d_v}{2}\rfloor}^{n-d_v} \sum_{c=\max(0,n-d-a)}^{d_v-(n-d-a)} \left(\binom{n-d_v}{a}\cdot\binom{d_v}{c}\right) \tag{2}$$

Eq. 2 sums over the number of possible binary vectors that can exist at every location inside the circle intersection. Taking inspiration from [27], this is a new and more interpretable derivation of the circle intersection than that originally developed [13]. Eq. 2 is approximately exponential for the closest, most important patterns where $d(\mathbf{p}_a, \boldsymbol{\xi}) \leq 2d$, which is crucial to how SDM approximates Attention. This is shown for a representative instance of SDM in Fig. 2. The details of this approximation are provided in Appendix B.2, but at a high level the binomial coefficients can be represented as binomial distributions and then approximated by normal distributions that contain exponentials. With the correctly chosen constants, $c_1$ and $c_2$, that are independent of the vector distance $d(\mathbf{p}_a, \boldsymbol{\xi})$, we can make the following approximation:

$$\mathcal{I}\big(d(\mathbf{p}_a, \boldsymbol{\xi}), d, n\big) \approx c_1 \exp\big(-c_2 \cdot d(\mathbf{p}_a, \boldsymbol{\xi})\big) \tag{3}$$

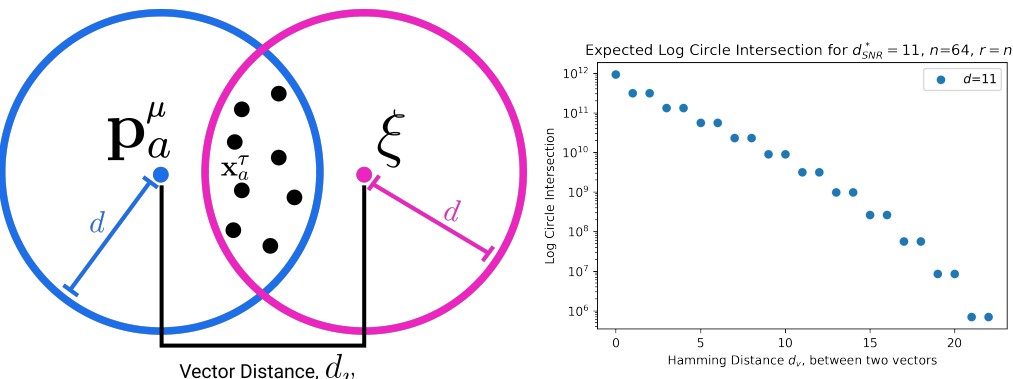

Figure 2: (**Left**) SDM's read operation using the Hamming radius $d$ (for reading and writing) and the vector distance $d_v = d(\boldsymbol{\xi}, \mathbf{p}_a^\mu)$. Recall that during the write operation, pattern addresses $\mathbf{p}_a^\mu$ write their pattern pointer $\mathbf{p}_p^\mu$ to neurons located at the addresses $\mathbf{x}_a^\tau$ (denoted here as black dots) within radius $d$. During the read operation, the query $\boldsymbol{\xi}$ reads from each neuron within radius $d$, thus creating an intersection. (**Right**) As $d_v$ between the query and pattern increases (x-axis), the size of their circle intersection falls approximately exponentially (y-axis). We use Eq. 2 with $n = 64$ and $d_{\text{SNR}}^* = 11$, while varying $d_v$ up to the distance of $d_v = 2d$ beyond which point there is no circle intersection. We plot the y-axis on a log scale to show how, because the curve is approximately linear, the circle intersection is approximately exponential. See Appendix B.2 for a formal analysis of the exponential approximation the circle intersection creates that is robust across parameters $n$, $d$, and $r$.

## 2 Attention Approximates SDM

To be able to handle a large number of patterns, we let the pattern address matrix with each pattern as a column be: $P_a = [\mathbf{p}_a^1, \mathbf{p}_a^2, ..., \mathbf{p}_a^m]$ with pointers $P_p = [\mathbf{p}_p^1, \mathbf{p}_p^2, ..., \mathbf{p}_p^m]$.

The Attention update rule [1] using its original notation is:

$$\boldsymbol{\xi}^{new} = V\text{softmax}(\beta K^T Q) = (W_v Y)\text{softmax}\big(\beta(W_k Y)^T(W_q \mathbf{q})\big),$$

where $K$, $V$, and $Q$ symbolize the "key", "value", and "query" matrices, respectively. $\mathbf{q}$ is a single query vector and $Y$ represents the raw patterns to be stored in memory. The $\text{softmax}(\beta\mathbf{x}) = \exp(\beta\mathbf{x})/\sum_{i=1}^n \exp(\beta x_i)$, where the exponential acts element-wise and Attention sets $\beta = 1/\sqrt{n}$. Softmax normalizes a vector of values to sum to 1 and gives the largest values the most weight due to the exponential function, to what extent depending on $\beta$. We can re-write this using our notation, including distinguishing continuous vectors in $\mathbb{R}^n$ from binary ones by putting a tilde above them:

$$\tilde{\boldsymbol{\xi}}^{new} = \tilde{P}_p\text{softmax}(\beta\tilde{P}_a^T\tilde{\boldsymbol{\xi}}). \tag{4}$$

We write $K = W_k Y = \tilde{P}_a$ as the raw input patterns $Y$ are projected by the learnt weight matrix $W_k$ into the SDM vector space to become the addresses $\tilde{P}_a$. Similarly, $V = W_v Y = \tilde{P}_p$ and $Q = W_q\mathbf{q} = \tilde{\boldsymbol{\xi}}$.

Showing the approximation between SDM Eq. 1 and Attention Eq. 4 requires two steps: (i) Attention must $L^2$ normalize its vectors. This is a small step because the Transformer already uses LayerNorm [28] before and after its Attention operation that we later relate to $L^2$ normalization; (ii) A $\beta$ coefficient for the softmax exponential must be chosen such that it closely approximates the almost exponential decay of SDM's circle intersection calculation.

To proceed, we define a map from binary vectors $\mathbf{a}$, $\mathbf{b}$ to $L^2$ normalized continuous vectors $\hat{\mathbf{a}}$, $\hat{\mathbf{b}}$, $h(\mathbf{a}) = \hat{\mathbf{a}}$, such that for any pair of pattern addresses the following holds:

$$d(\mathbf{a}, \mathbf{b}) = \lfloor \frac{n}{2}\big(1 - \hat{\mathbf{a}}^T\hat{\mathbf{b}}\big)\rfloor, \tag{5}$$

where $\lfloor\cdot\rfloor$ is the floor operator. We assume that this map exists, at least approximately. This map allows us to relate the binary SDM circle intersection (Eq. 2) to the exponential used in Attention (Eq. 4) by plugging it into the exponential approximation of Eq. 3:

$$\mathcal{I}\big(d(\mathbf{p}_a, \boldsymbol{\xi}), d, n\big) = \mathcal{I}\Big(\lfloor\frac{n}{2}(1 - \hat{\mathbf{p}}_a^T\hat{\boldsymbol{\xi}})\rfloor, d, n\Big) \approx c_1 \exp\Big(-c_2\lfloor\frac{n}{2}(1 - \hat{\mathbf{p}}_a^T\hat{\boldsymbol{\xi}})\rfloor\Big)$$
$$\approx c_3 \exp\big(\beta\hat{\mathbf{p}}_a^T\hat{\boldsymbol{\xi}}\big), \tag{6}$$

where $c_3$ encompasses the constants outside of the exponential. We replaced the remaining constants in the exponential with $\beta$, that is a function of $n$ and $d$ and is an approximation due to the floor operation.

Finally, these results allow us to show the relationship between Attention and SDM:

$$\tilde{\boldsymbol{\xi}}^{new} = \hat{P}_p\text{softmax}(\beta\hat{P}_a^T\hat{\boldsymbol{\xi}}) = \frac{\sum_{\mathbf{p}\in P}\exp(\beta\hat{\mathbf{p}}_a^T\hat{\boldsymbol{\xi}})\hat{\mathbf{p}}_p}{\sum_{\mathbf{p}\in P}\exp(\beta\hat{\mathbf{p}}_a^T\hat{\boldsymbol{\xi}})} \approx \frac{\sum_{\mathbf{p}\in P}\mathcal{I}\big(\lfloor\frac{n}{2}(1 - \hat{\mathbf{p}}_a^T\hat{\boldsymbol{\xi}})\rfloor, d, n\big)\hat{\mathbf{p}}_p}{\sum_{\mathbf{p}\in P}\mathcal{I}\big(\lfloor\frac{n}{2}(1 - \hat{\mathbf{p}}_a^T\hat{\boldsymbol{\xi}})\rfloor, d, n\big)}. \tag{7}$$

Alternatively, instead of converting cosine similarity to Hamming distance to use the circle intersection Eq. 2 in binary vector space, we can extend SDM to operate with $L^2$ normalized pattern and neuron addresses (Appendix B.3).[3] This continuous SDM circle intersection closely matches its binary counterpart in being approximately exponential:

$$\mathcal{I}_c\Big(\hat{\mathbf{p}}_a^T\hat{\boldsymbol{\xi}}, 1 - \frac{2d}{n}, n\Big) \approx c_4 \exp\big(\beta_c\hat{\mathbf{p}}_a^T\hat{\boldsymbol{\xi}}\big). \tag{8}$$

---

[3]Pattern pointers can point to a different vector space and thus do not need to be $L^2$ normalized. However, in canonical SDM they point to pattern addresses in the same space so we write them as also being $L^2$ normalized in our equations.

We use $\mathcal{I}_c$ to denote this continuous intersection, use Eq. 5 to map our Hamming $d$ to cosine similarity, and use coefficients $c_4$ and $\beta_c$ to acknowledge their slightly different values. Then, we can also relate Attention as we did in Eq. 7 to continuous SDM as follows:

$$\tilde{\boldsymbol{\xi}}^{new} = \hat{P}_p \text{softmax}(\beta \hat{P}_a^T \hat{\boldsymbol{\xi}}) = \frac{\sum_{\mathbf{p} \in P} \exp(\beta \hat{\mathbf{p}}_a^T \hat{\boldsymbol{\xi}}) \hat{\mathbf{p}}_p}{\sum_{\mathbf{p} \in P} \exp(\beta \hat{\mathbf{p}}_a^T \hat{\boldsymbol{\xi}})} \approx \frac{\sum_{\mathbf{p} \in P} \mathcal{I}_c\left(\hat{\mathbf{p}}_a^T \hat{\boldsymbol{\xi}}, 1 - \frac{2d}{n}, n\right) \hat{\mathbf{p}}_p}{\sum_{\mathbf{p} \in P} \mathcal{I}_c\left(\hat{\mathbf{p}}_a^T \hat{\boldsymbol{\xi}}, 1 - \frac{2d}{n}, n\right)}. \quad (9)$$

We have written Attention with $L^2$ normalized vectors and expanded out the softmax operation to show that it is approximated when we replace the exponential weights by either the binary or continuous SDM circle intersections (Eqs. 7 and 9, respectively). The right hand side of Eq. (7) is identical to Eq. 2 aside from using continuous, $L^2$ normed vectors and dropping the elementwise majority function $g(\cdot)$ that ensured our output was a binary vector. In the Transformer, while the Attention equation does not contain any post-processing function to its query update $\tilde{\boldsymbol{\xi}}^{new}$, it is then post-processed by going through a linear projection and LayerNorm [1] and can be related to $g(\cdot)$.

To fit $\beta$ to binary SDM, we convert the Hamming distances into cosine similarity using Eq. 5 and use a univariate log linear regression:

$$\log\left(\mathcal{I}\left(d(\mathbf{p}_a, \boldsymbol{\xi}), d, n\right)\right) \approx \log(c_3) + \beta(\hat{\mathbf{p}}_a^T \hat{\boldsymbol{\xi}}). \quad (10)$$

We expect the exponential behavior to break at some point, if only for the reason that if $d(\mathbf{p}_a, \boldsymbol{\xi}) \geq 2d$ the circle intersection becomes zero. However, closer patterns are those that receive the largest weights and "attention" such that they dominate in the update rule and are the most important.

In Fig. 3, we plot the softmax approximation to binary and continuous SDM for our smallest optimal $d^*_{\text{Mem}} = 5$ and largest $d^*_{\text{CD}} = 15$ to show not only the quality of the approximations but also how many orders of magnitude smaller the normalized weights are when $d(\mathbf{p}_a, \boldsymbol{\xi}) > d$. For these plots, we plug into our binary circle intersection equation each possible Hamming distance from 0 to 64 when $n = 64$ and converting Hamming distance to cosine similarity, doing the same for our continuous circle intersection. Here use our binary intersection values to fit $\beta$, creating the exponential approximation. To focus our exponential approximation on the most important, closest patterns, we fit our regression to those patterns $d(\mathbf{p}_a, \boldsymbol{\xi}) < d$ and allow it to extrapolate to the remaining values. We then normalize the values and plot them along with an smaller inset plot in log space to better show the exponential relationship. In both plots, looking at the log inset plot first, the point at which the circle intersection in blue ceases to exist or be exponential corresponds to a point in the main normalized plot where the weights are $\approx 0$.

The number of neurons $r$ and how well they cover the pattern manifold are important considerations that will determine SDM's performance and degree of approximation to Attention. Increasing the number of neurons in the circle intersection can be accomplished by increasing the number of neurons in existence, ensuring they cover the pattern manifold, and reducing the dimensionality of the manifold to increase neuron density.[4] In SDM's original formulation, it was assumed that neuronal addresses were randomly distributed and fixed in location, however, extensions to SDM [29] have proposed biologically plausible competitive learning algorithms to learn the manifold [30]. To ensure the approximations to SDM are tight, we test random and correlated patterns in an autoassociative retrieval task across different numbers of neurons and SDM variants (Appendix B.7). These variants include SDM implemented using simulated neurons and the Attention approximation with a fitted $\beta$.[5] To summarize, Attention closely approximates the SDM update rule when it uses $L^2$ normed continuous vectors and a correctly chosen $\beta$.

## 3 Trained Attention Converges with SDM

For many instantiations of SDM, there exists a $\beta$ that can be found via the log linear regression Eq. 10 that makes Attention approximate it well. However, depending on the task at hand, there are instantiations of SDM that are better than others as highlighted by the different $d^*$ optimal values. If

---

[4]This can be done by learning weight projection matrices like in Attention to make the manifold lower dimensional and also increase separability between patterns.

[5]The code for running these experiments, other analyses, and reproducing all figures is available at https://github.com/trentbrick/attention-approximates-sdm.

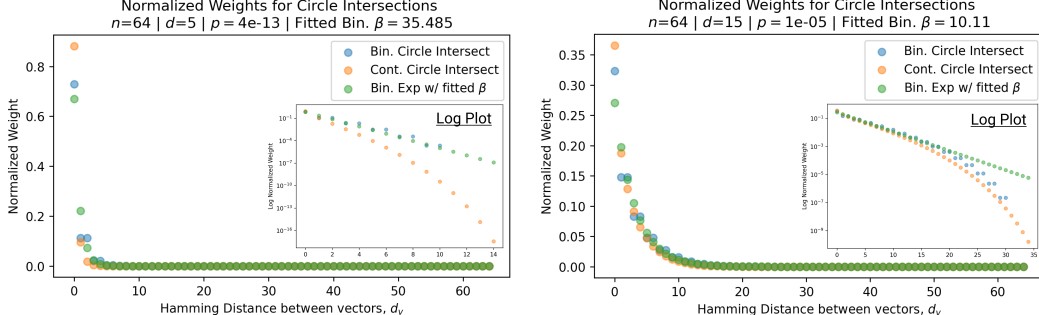

Figure 3: Representative plots showing the relationships in Eqs. 7 and 9 between the softmax approximation (green) and the normalized binary (blue) and continuous (orange) circle intersections using Transformer Attention parameters. Here the softmax approximation uses the Eq. 10 regression to fit its $\beta$ to binary SDM. We use Eq. 5 to relate Hamming distance and cosine similarity for our vector distances on the x-axis. The insets use a log y-axis to show the circle intersections are approximately exponential when $d(\mathbf{p}_a, \boldsymbol{\xi}) \leq 2d$. **(Left)** Uses $d^*_{\text{Mem}} = 5$, corresponding to the Hamming distance reading and writing to $p = 4e - 13$ of the vector space. **(Right)** Uses $d^*_{\text{CD}} = 15$. These results hold well for other values of $n$ and $d$ but we focus on the Attention setting.

Attention in the Transformer model is implementing SDM, we should expect for trained Attention to use $\beta$s that correspond to reasonable instances of SDM. We use as reference points these optimal $d^*$s.

Attention learns useful pattern representations that are far from random so this SDM $\beta$ that fits the optimal $d^*$s are only a weak reference for what $\beta$ values might be reasonable. However, because these $d^*$ definitions of optimality span from maximizing convergence with query noise, to maximizing memory capacity with noise free queries, we should expect the Transformer dealing with noisy queries and wanting reliable retrieval to interpolate between these $d^*$ values. Here, we provide empirical evidence that this is indeed the case. We analyze the $\beta$ coefficients learnt by the "Query-Key Normalization" Transformer Attention variant [22]. Query-Key Normalization makes it straightforward to find $\beta$ because it is learnt via backpropagation and easy to interpret because it uses cosine similarity between the query and key vectors. To further evaluate the convergence between Attention and SDM $\beta$ coefficients and make it more general, we also investigate the GPT2 architecture [3]. However, in this case we need to infer "effective" $\beta$ values from the size of query key dot products in the softmax. This makes these results, outlined in Appendix A.2, more approximate but

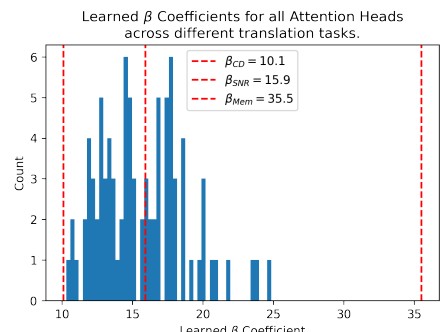

Figure 4: Histogram showing learned $\beta$ coefficients for all Attention heads across layers for the 5 translation tasks used in [22]. We plot the $\beta$ values for Attention that approximate the different $d^*$ definitions showing how the $\beta$s are interpolating between them. $\beta_{CD}$ is optimal for critical distance (maximum noise for each query); $\beta_{SNR}$ is optimal for Signal-to-Noise ratio. This assumes there is no query noise and SDM wants to minimize noise from other queries. $\beta_{Mem}$ maximizes memory capacity and also assumes no query noise.

they remain largely in agreement with the learnt $\beta$s of Query-Key Norm.

The Query-Key Norm Attention heads learn $\beta \in [10, 25]$ as shown in Fig. 4.[6] Note that the whole range of $\beta \in [10, 25]$ interpolates between the $d^*$ values well, in particular with the critical distance optimal that realistically assumes noisy queries and the SNR optimal, where having a high signal to noise ratio is desirable for both memory capacity and critical distance (see Appendix B.5).

---

[6]These values were graciously provided by the authors of [22] in private correspondence for their trained Transformer models.

In requiring that Attention vectors be $L^2$ normalized and $\beta$ fitted, SDM predicted Query-Key Norm. This is interesting because Query-Key Norm has evidence of improving Transformer performance and training speed [22, 31]. We say more about its advantages and caveats in Appendix A.1.

## 4 Transformer Components Interpreted with SDM

We can leverage how Attention approximates SDM to interpret many components of the Transformer architecture.[7] This exercise demonstrates the explanatory power of SDM and, in relating it to additional unique Transformer components, widens the bridge upon which ideas related to SDM and neuroscience can cross into deep learning and vice versa.

A crucial component of the Transformer is its Feed Forward (FF) layer that is interleaved with the Attention layers and uses approximately 2/3rds of the Transformer's parameter budget [1]. Attention's Transformer implementation and SDM deviate importantly in Attention's use of ephemeral patterns from the current receptive field. In order to model temporal dependencies beyond the receptive field, we want Attention to be able to store persistent memories. Work has compellingly shown that the FF layers store these persistent memories [32, 33, 34]. SDM can be interpreted as this FF layer because in [32] the FF layer was substituted with additional, persistent key and value vectors that Attention learnt independently rather than projecting from its current inputs. This substitution performed on par with the FF layer which, combined with deeper analysis in [33], shows the FF layers are performing Attention with long term memories and thus can be directly interpreted as SDM.

Another crucial component of the Transformer is its use of LayerNorm [28, 35, 1]. LayerNorm has a natural interpretation by SDM as implementing a similar constraint to $L^2$ normalization. However, while it ensures all of the vectors are on the same scale such that their dot products are comparable to each other, it does not constrain these dot products to the interval $[-1, 1]$ like cosine similarity. In addition to providing an interpretation for the importance of LayerNorm, this discrepancy has led to two insights: First, as previously mentioned, it predicted that Query-Key Normalization could be a useful inductive bias (more details in Appendix A.1). Second, it has provided caution to interpretations of Attention weights. Fig. 8 of Appendix A.3 shows that there are many value vectors that receive very small amounts of attention but have large $L^2$ norms that dominate the weighted summation of value vectors. This makes it misleading to directly interpret Attention weights without $L^2$ normalization of the value vectors and this has not been done in work including [1, 36, 37, 38]. Beyond helping future interpretations of Attention, we tested $L^2$ normalized value vectors as a potentially useful inductive bias by training a GPT2 model with it. Our results showed that this did not change performance but $L^2$ normalization should still be performed in cases where the Attention weights will be interpreted. See Appendix A.3 for a full discussion.

Finally, multi-headed Attention is a Transformer component where multiple instantiations of Attention operate at the same hierarchical level and have their outputs combined. Multi-heading allows SDM to model probabilistic outputs, providing an interpretation for why it benefits Attention. For example, if we are learning a sequence that can go "$A \rightarrow B$" and "$A \rightarrow Z$" with equal probability, we can have one SDM module learn each transition. By combining their predictions we correctly assign equal probability to each. This probabilistic interpretation could explain evidence showing that Attention heads pay attention to different inputs and why some are redundant post training [39, 40, 10].

An important difference between SDM and the Transformer that remains to be reconciled is in the Transformer's hierarchical stacking of Attention. This is because, unlike in the traditional SDM setting where the pattern addresses (keys) and pattern pointers (values) are known in advance and written into memory, this cannot be done for layers of SDM beyond the first that will need to learn latent representations for its pattern addresses and pointers (keys and values). Transformer Attention solves this problem by learning its higher level keys and values, treating each input token as its own query to generate a new latent representation that is then projected into keys and values [1]. This does not mean SDM would fail to benefit from hierarchy. As a concrete example, the operations of SDM are related to the hierarchical operations of [41]. More broadly, we believe thinking about how to learn the latent keys and values for the higher levels of SDM could present new Transformer improvements. A key breakthrough of the recent Performer architecture that highlights the arbitrariness of the original Transformer solution is its use of a reduced set of latent keys and values [42].

---

[7]For a summary of the components that make up the full Transformer architecture see Fig. 9 of Appendix A.4.

# 5    A Biologically Plausible Implementation of Attention

Here, we provide an overview of SDM's biological plausibility to provide a biologically plausible implementation of Attention. SDM's read and write operations have non trivial connectivity requirements described in [13, 15]. Every neuron must: (i) know to fire if it is within $d$ Hamming distance of an input; (ii) uniquely update each element of its storage vector when writing in a new pattern; (iii) output its storage vector during reading using shared output lines so that all neuron outputs can be summed together.

Unique architectural features of the cerebellar cortex can implement all of these requirements, specifically via the three way convergence between granule cells, climbing fibers and Purkinje cells: (i) all granule cells receive inputs from the same mossy fibers to check if they are within $d$ of the incoming query or pattern; (ii) each granule cell has a very long parallel fiber that stores memories in synapses with thousands of Purkinje cells [43], updated by LTP/LTD (Long Term Potentiation/Depression) from joint firing with climbing fibers; (iii) all granule cells output their stored memories via their synapses to the Purkinje cells that perform the summation operation and use their firing threshold to determine if the majority bit was a 1 or 0, outputting the new query [13, 15]. Moreover, the Drosophila mushroom body is highly similar to the cerebellum and the previous cell labels for each function can be replaced with Kenyon cells, dopaminergic neurons, and mushroom body output neurons, respectively [17].

While SDM fits the unique features of the cerebellum well, this connection has limitations. Explanations for some of the original model's limitations have been put forward to account for sparse dendritic connections of Granule cells [44] and the functions of at least two of the three inhibitory interneurons: Golgi, Stellate and Basket cells [29, 45]. However, there are futher challenges that remain, including better explanations of the inputs to the mossy and climbing fibers and outputs from the Purkinje cells; in particular, how the mossy and climbing fiber inputs synchronize for the correct spike time dependent plasticity [46]. Another phenomenon that SDM does not account for is the ability of Purkinje cells to store the time intervals associated with memories [47]. Further research is necessary to update the state of SDM's biological plausibility with modern neuroscientific findings.

# 6    Related Work

Previous work showed that the modern Hopfield Network, when made continuous and optimized differently, becomes Attention [25, 26]. This result was one motivation for this work because Hopfield Networks are another associative memory model. In fact, it has been shown that SDM is a generalization of the original Hopfield Network (Appendix B.6) [29]. While SDM is a generalization of Hopfield Networks, their specific discrepancies provide different perspectives on Attention. Most notably, Hopfield Networks assume symmetric weights that create an energy landscape, which can be powerfully used in convergence proofs, including showing that one step convergence is possible for the modern Hopfield Network, and by proxy, Attention and SDM when it is a close approximation [25, 26]. However, these symmetric weights come at the cost of biological plausibility that SDM provides in addition to its geometric framework and relation to Vector Symbolic Architectures [29, 48].

Other works have tried to reinterpret or remove the softmax operation from Attention because the normalizing constant can be expensive to compute [49, 50]. However, while reducing computational cost, these papers show that removing the softmax operation harms performance. Meanwhile, SDM not only shows how Attention can be written as a Feedforward model [15] but also reveals that through simple binary read and write operations, (the neuron is either within Hamming/cosine distance or it's not) the softmax function emerges with no additional computational cost.

Since the publication of SDM, there have been a number of advancements not only to SDM specifically, but also through the creation of related associative memory algorithms under the name of "Vector Symbolic Architectures" [51]. Advancements to SDM include using integer rather than binary vectors [52], handling correlated patterns [29], and hierarchical data storage [53]. Vector Symbolic Architectures, most notably Holographic Reduced Representations, have ideas that can be related back to SDM and the Transformer in ways that may be fruitful [54, 55, 56, 57, 58, 59].

The use of external memory modules in neural networks has been explored most notably with the Neural Turing Machine (NTM) and its followup, the Differentiable Neural Computer (DNC) [23, 60]. In order to have differentiable read and write operations to the external memory, they use the softmax

function. This, combined with their use of cosine similarity between the query and memory locations, makes both models closely related to SDM. A more recent improvement to the NTM and DNC directly inspired by SDM is the Kanerva Machine [24, 61, 62, 63]. However, the Kanerva Machine remains distinct from SDM and Attention because it does not apply the a Hamming distance threshold on the cosine similarity between its query and neurons. Independent of these discrepancies, we believe relating these alternative external memory modules to SDM presents a number of interesting ideas that will be explored in future work.

## 7 Discussion

The result that Attention approximates SDM should enable more cross pollination of ideas between neuroscience, theoretical models of associative learning, and deep learning. Considering avenues for future deep learning research, SDM's relationship to Vector Symbolic Architectures is particularly compelling because they can apply logical and symbolic operations on memories that make SDM more powerful [55, 64, 65, 66, 67]. SDM and its relation to the brain can inspire new research in not only deep learning but also neuroscience, because of the empirical success of the Transformer and its relation to the cerebellum, via SDM.

Our results serve as a new example for how complex deep learning operations can be approximated by, and mapped onto, the functional attributes and connectivity patterns of neuronal populations. At a time when many new neuroscientific tools are mapping out uncharted neural territories, we hope that more discoveries along the lines of this work connecting deep learning to the brain will be made [68, 69, 70].

*Limitations* While our work shows a number of convergences between SDM, Attention, and full Transformer models, these relationships remain approximate. The primary approximation is the link between SDM and Attention that exists not only in SDM's circle intersection being approximately exponential but also its use of a binary rather than continuous space. Another approximation is between optimal SDM Hamming radii $d^*$ and Attention $\beta$ coefficients. This is because we assume patterns are random to derive the $d^*$ values. Additionally, in the GPT2 Transformer models we must infer their effective $\beta$ values. Finally, there is only an approximate relationship between SDM and the full Transformer architecture, specifically with its Feed Forward and LayerNorm components.

## 8 Conclusion

We have shown that the Attention update rule closely approximates SDM when it $L^2$ norms its vectors and has an appropriate $\beta$ coefficient. This result has been shown to hold true in both theory and empirical evaluation of trained Transformer models. SDM predicts that Transformers should normalize their key, query and value vectors, preempting the development of Query-Key Normalization and adding nuance to the interpretation of Attention weights. We map SDM onto the Transformer architecture as a whole, relate it to other external memory implementations, and highlight extensions to SDM. By discussing how SDM can be mapped to specific brain architectures, we provide a potential biological implementation of Transformer Attention. Thus, our work highlights another link between deep learning and neuroscience.

## Acknowledgements

Thanks to Dr. Gabriel Kreiman, Alex Cuozzo, Miles Turpin, Dr. Pentti Kanerva, Joe Choo-Choy, Dr. Beren Millidge, Jacob Zavatone-Veth, Blake Bordelon, Nathan Rollins, Alan Amin, Max Farrens, David Rein, Sam Eure, Grace Bricken, and Davis Brown for providing invaluable inspiration, discussions and feedback. Special thanks to Miles Turpin for help working with the Transformer model experiments. We would also like to thank the open source software contributors that helped make this research possible, including but not limited to: Numpy, Pandas, Scipy, Matplotlib, PyTorch, HuggingFace, and Anaconda. Work funded by the Harvard Systems, Synthetic, and Quantitative Biology PhD Program.

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
