# Appendix

## A   Transformer Convergence with SDM

### A.1   Query Key Normalization

For Attention to approximate SDM, it must $L^2$ normalize its vectors. Instead of $L^2$ normalization the Transformer uses LayerNorm and this has been shown to be crucial to its performance [28, 1, 35]. If the Transformer works well because it approximates SDM then addressing this discrepancy could improve the Transformer. It turns out this idea to $L^2$ normalize the query and key vectors was already tested by "Query-Key Normalization" and is the culmination of previous attempts to improve upon LayerNorm [71, 72]. Query-Key Norm has shown performance improvements on five small translation datasets and a pseudo-code generation task but these improvements were not dramatic and it still needs to be tested on additional larger datasets [22, 31]. In the context of the default Transformer already learning roughly the correct dot product interval that is suggestive of LayerNorm and the learned projection matrices approximating $L^2$ normalization, as is shown in Appendix A.2, it makes sense that Query-Key Norm is a useful but not dramatically important inductive bias. Note that Query-Key Norm as developed still retained pre-Attention LayerNorm. It must be empirically validated but we hypothesize this is unneccesary and to not have affected the results.

Nevertheless, Query-Key Norm remains promisingly convergent with SDM's requirement of $L^2$ normalization for cosine similarities and the $\beta$s it learns via backpropagation. Moreover, independent of task performance, if Query-Key Norm can remove the need for the LayerNorm to learn $2n$ parameters and make Transformer training both more stable and fast by removing learning rate warmup, these will be substantive contributions to training speed, memory, and computational costs [71, 28].

### A.2   GPT2 SDM Approximation

In continuous SDM we assume vectors are $L^2$ normalized so their dot product lies within the interval $[-1, 1]$ and the $\beta$ coefficient is used to determine $d$ for read and write operations. However, instead

of changing $\beta$ we can fix it and change the size of our vector norms. The magnitude of the vector dot product can then give us an "effective" $\beta$ value and is the route the original Transformer architecture takes using unconstrained vector norms and a fixed $\beta = 1/\sqrt{n}$.

Computing the effective $\beta$ for the original pre-trained GPT2 models requires trying to infer the maximum vector norms used by putting many inputs into the model. This is a less accurate approach than using cosine similarity and learning the $\beta$ coefficient via backprop as "Query-Key Normalization" did [22]. While this approach gives results that are noisier and harder to interpret, they are in general agreement with those from the more exact Query-Key Normalization analysis. Therefore, these results lend additional support for the Transformer learning to use "effective" $\beta$s that not only approximate instantances of SDM well but in particular instances of SDM that are optimal in various ways under the random pattern assumption.

We used the "small", 117M parameter and "large", 774M parameter pre-trained GPT2 models provided by [3]. The GPT2 architecture was selected because it has produced some of the most compelling results in natural language processing and more recently on image tasks [8, 5] with larger models continuing to give improvements [9, 3, 4, 73]. In addition, the HuggingFace API (https://huggingface.co/transformers/index.html) made this analysis reasonably straightforward.

The models were given eight diverse text prompts for which the dot products of the keys and queries after their projections by the weight matrices were computed. An example text prompt could be: "Sparse Distributed Memory (SDM), is a biologically plausible form of associative memory". The inputs actually used are much longer and can be found in the file "text_inputs.txt" of our GitHub repo.

After computing the magnitude of the dot product and multiplying it by the fixed $\beta = 1/\sqrt{n}$ to get the magnitude of the input to the softmax (referred to as "softmax input"), we look for the largest magnitude produced by any text input for each Attention head that determines the effective $\beta$. This is because the largest input sets the maximum interval such that if we pretended the vectors were giving a cosine similarity between $[-1, 1]$, the scaling of this interval would be represented as the $\beta$ coefficient. We care about the positive values because of the shape of the exponential function where $\exp(x > 0) \in [1, \infty]$ and $\exp(x \leq 0) \in [0, 1]$ (assuming $\beta > 0$) such that its interaction with positive inputs is far more significant in determining the shape of the exponential and its range of outputs. These positive values also correspond to vectors that are closer than orthogonal to the query and will have the largest attention weights.

Figure 5 shows that both of the pre-trained GPT2 models produce effective $\beta$s across all of their Attention layers, and texts for each head on the interval $[3, 71]$ with almost all $\beta$s in the interval $[3, 12]$. Note that to produce a correct effective $\beta$, *both* the LayerNorm parameters and the vector projection matrices, $W_k$ and $W_q$ used before the softmax operation must be correctly calibrated [28]. These $\beta$s are smaller than those used in Query-Key Norm and correspond to larger Hamming radii $d$, however they are lower bounds on the true $\beta$s because of the need for inputs to infer the maximum dot product magnitude where many of these inputs are far from the maximum cosine similarity with the query. The fact that many patterns in the vector space will not be at the minimum or maximum cosine similarities (if they were $L^2$ normed) can be seen in Fig. 6 where the distribution of cosine similarities multiplied by beta is almost symmetric about the orthogonal 0 distance and has a steep drop off around this value, like the binomial distribution of Hamming distances centered at the orthogonal distance that random vectors obey.

In Fig. 7 we provide another perspective on the effective $\beta$s of the Transformer by showing their distribution across layers. For each layer we take the maximum softmax input for a given text prompt for each head. We then take the mean of this maximum for each head and plot this for each text input. Across all inputs for both small and large GPT2 we see the same pattern where the effective $\beta$ starts off large then dips before spiking and dropping off again. Recall that larger $\beta$s correspond to smaller Hamming distances that read to and write from fewer patterns. As validation of our analytic approach, we see a similar result in [25] for their BERT model (see Figure A.3 on pg. 74 of [25]), aside from their very first layer.[8] In both models the middle layers have a sudden spike where they look at very few patterns while the final layers look at a larger number of patterns. In our GPT2 analysis the very

---

[8]This difference in the first layer $\beta$s could be explained by the fact that GPT2 does not apply LayerNorm before its first Attention layer attention operation while BERT does [3, 74, 28].

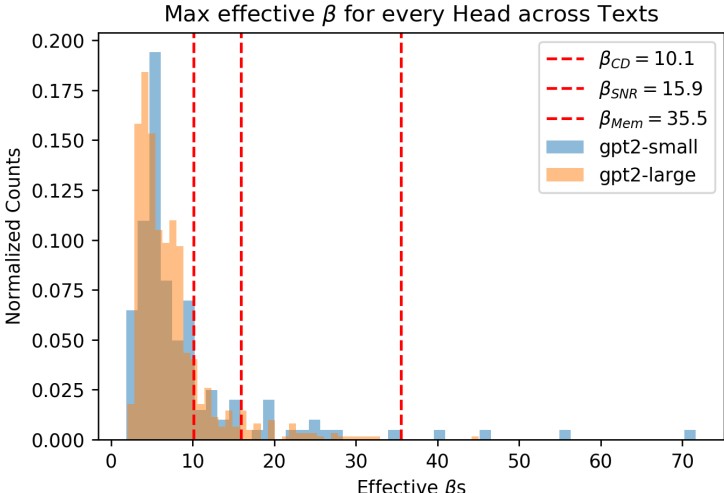

Figure 5: Histogram showing the maximum softmax input, defined as the "effective" $\beta$ for the GPT2 Small and Large models aggregated across all text data for all heads and layers. Because of the need for inputs to infer their dot product magnitude and that many of these inputs will be far from maximum cosine similarity, we assume these $\beta$s are lower bounds on their true values. Nevertheless, they remain quite convergent across heads with that of Query-Key Norm.

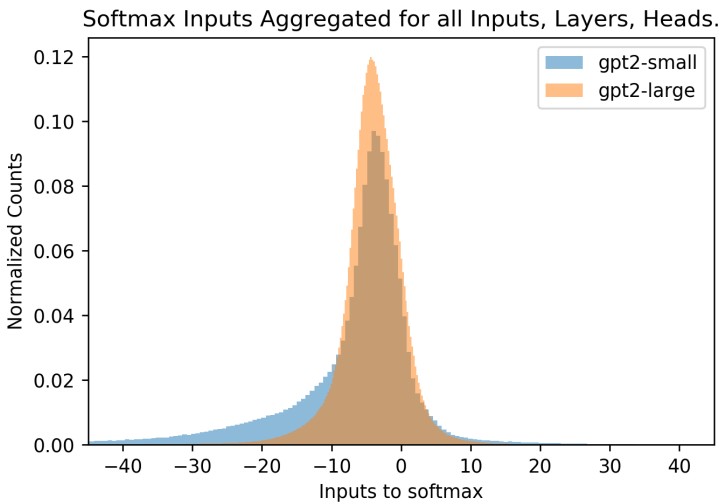

Figure 6: This figure shows the size of the softmax inputs to the models aggregated across all layers, heads, and text inputs. It is clear that the key and query are further away than the orthogonal distance a majority of the time because a majority of values are $< 0$. For both GPT2-small and GPT2-large there is an approximately normal distribution centered almost at zero but GPT2-small has a leftward skew we do not know the reason for.

first layer also looks at few patterns while in the BERT analysis they look at the largest number of patterns.

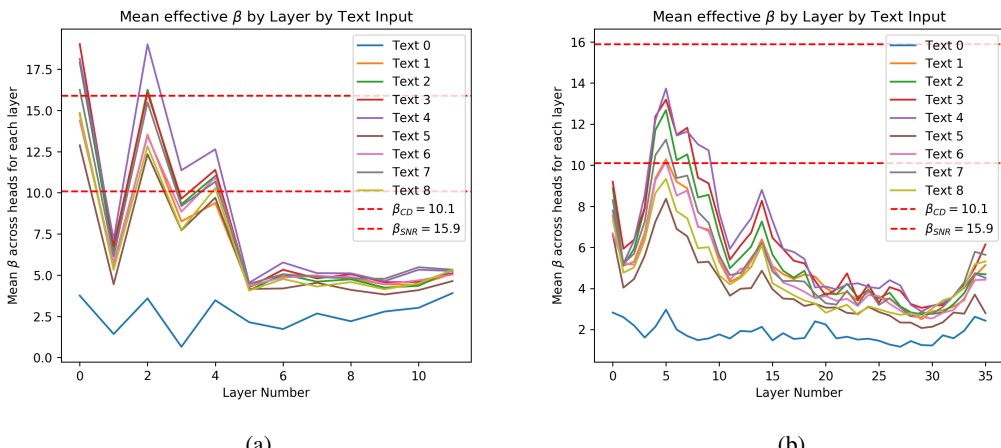

(a)

(b)

Figure 7: The effective $\beta$ values for **(a)** GPT2-small and **(b)** GPT2-large are not far off those from Query-Key Norm Fig. 4. This is the mean of the maximum softmax input seen across all heads, plotted for each layer and text input. This pattern of the effective $\beta$ starting off large, then dropping before spiking and falling off again matches that of the BERT model analyzed in [25] (for all but the first layer). The dotted red lines are the $d^*$ optimal $\beta$ values where we leave out the memory optimal $\beta = 35.5$ as it would distort the y-axis too much as was also much larger than the largest learned $\beta$s in Query-Key Norm Fig. 4. The first text input (blue line) has smaller values because it is a short text passage, only using 195 tokens as its input, while all other inputs use >700 tokens. This is an example of how we need to use many, long text inputs in order to infer the largest inputs to the softmax. Recall that a larger $\beta$ corresponds to a smaller hamming distance, $d$.

## A.3 Transformer $L^2$ Value Vector Normalization Is Crucial For Interpreting Attention Weights

The need to $L^2$ norm the value vector of Attention for it to approximate SDM has highlighted a discrepancy in the Transformer. Attention does a weighted sum of the value vectors but this sum can be skewed by the different norms of the value vectors. Using GPT2, we investigated the relationship between the attention weight and the $L^2$ norm of different tokens, finding a bias shown in Fig. 8 where low attention value vectors can have much larger norms. This bias enables certain low attention inputs to "skip the line" and override their assigned attention weight. For example an attention value of 0.1 multiplied by a vector with an $L^2$ norm of 20 gives the vector an overall weight of 2.0 in the summation, this is 2x larger than if all attention weight (a value of 1.0), was on a single value with a norm of 1. This observation is crucial for any Transformer interpretability research, however, the original Transformer paper and later work simply visualizes attention weights and uses them as a proxy for what the model is using to generate outputs to the next layer and its final prediction. By failing to account for the value vector norm, these interpretations of Transformer Attention are inaccurate [1, 36, 37, 38].[9]

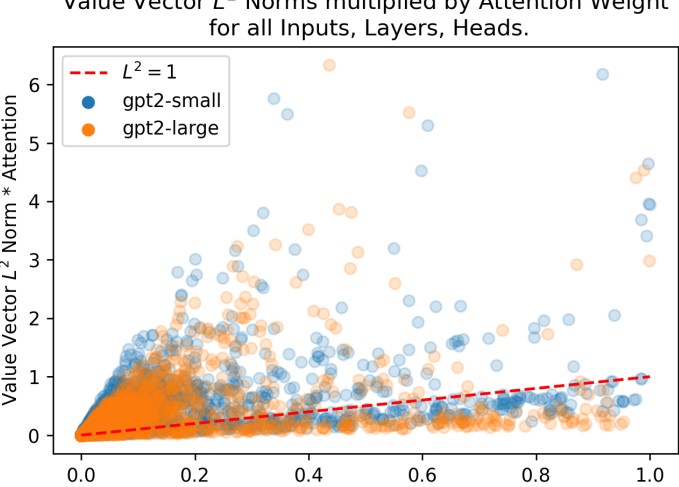

Figure 8: Input tokens that get little attention weighting can have very large $L^2$ norms in their value vectors, which biases the attention they get in their weighted summation. Each point on the scatter plots is an input token to the model, aggregated across all text data for all heads and layers. Each token has its Attention weight on the x-axis (the output of the softmax operation) and its value vector that we compute the $L^2$ norm of on the y-axis multiplied by the attention it recieves. Each point is plotted with an opacity of 0.2 and the red dotted line is the weighting that inputs would have if all of their $L^2$ norms were 1. We randomly selected 500,000 examples from each model and plotted them.

Beyond helping with interpretability, we investigated whether or not $L^2$ norming the value vectors affected performance. We trained two GPT2 models on an autoregressive word corpus benchmark, one that used $L^2$ normalization on its value vectors before the Attention weighted summation, and the other that did not. We found that their training performances were virtually identical. We had assumed that $L^2$ normalization would not lead to a dramatic change in performance because, while we outline next the hypothetical reasons why it may help or harm the model, either way there are enough free parameters for a good solution to be found.

On one hand, giving the Transformer the ability to give certain inputs large value vector norms such that they are always accounted for in the weighted summation, no matter their attention weighting, could be advantageous by allowing attention to focus instead on those inputs that are less consistently important. However, we also considered a scenario similar that of a VAE experiencing posterior

---

[9]Methods like in [40] that use a gradient based approach to identifying token importance implicitly account for the vector norms and do not have this problem, but gradient based analysis appears to be less common than just using the Attention weights when interpreting Transformers.

collapse when its decoder is too powerful, such that its latent space becomes meaningless [75]. Enforcing $L^2$ norm on the value vectors could have been a useful inductive bias that forced the Attention calculation to always learn which input tokens it should pay attention to. In the end, it appears that both approaches are equally valid. One point in favour of $L^2$ norm of the value vectors is that it allows for the Attention weights to be interpreted in the natural intuitive way that many papers have made the mistake of doing without correctly accounting for the norms of the value vectors.

## A.4 Transformer Architecture

A full formulation of the Transformer GPT architecture (used by GPT2 and GPT3) is shown in Fig. 9 [3, 4, 1]. The GPT architecture is the decoder portion of the original Transformer but without the encoder-decoder Attention submodules. We present this specific architecture because of its heteroassociative task of predicting the next input (there are compelling theories such as Predictive Coding that our brains do the same [76, 11]) and its state of the art performance in natural language and more recently image generation [5, 8]. The original Transformer architecture was used for language translation tasks [1].

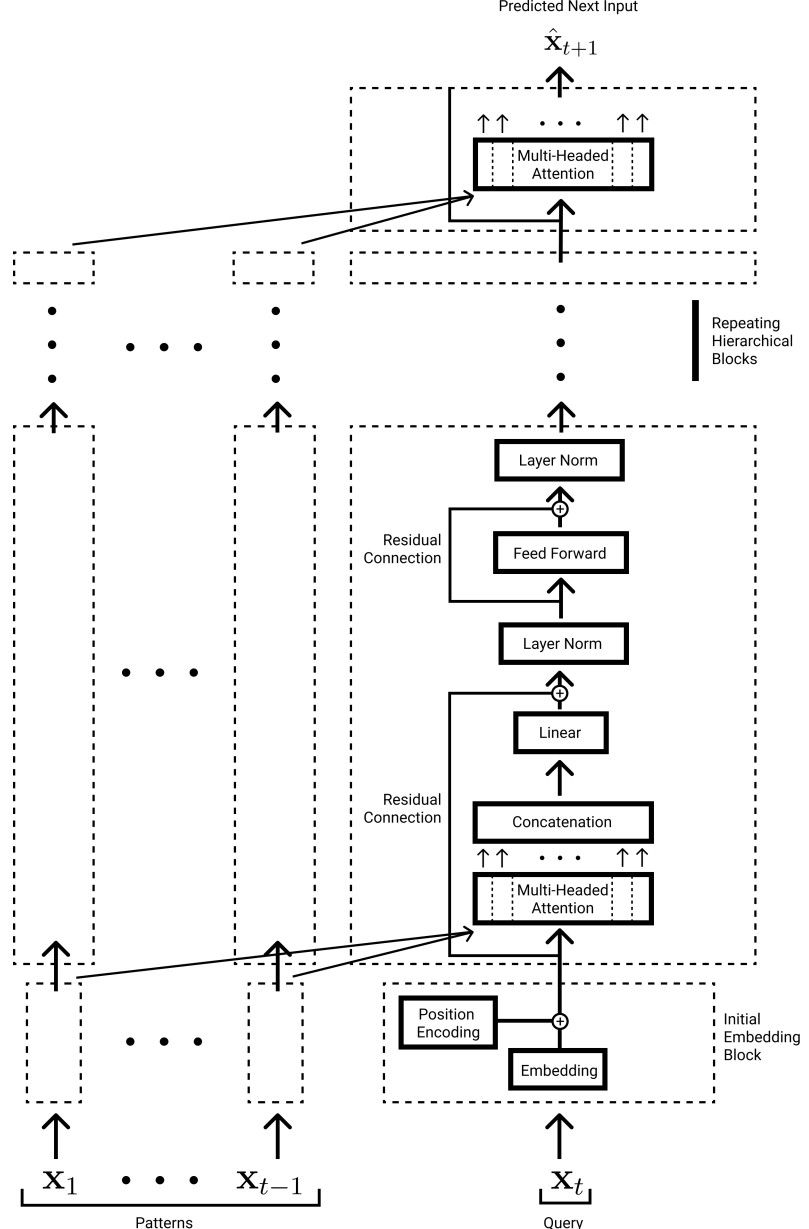

Figure 9: The GPT Transformer Architecture that is the decoder portion of the original Transformer [3, 1]. GPT takes in all inputs at previous time points (within its receptive field) $\mathbf{x}_1, \ldots, \mathbf{x}_{t-1}$ and uses these as patterns. Using the most recent input $\mathbf{x}_t$ as the query, it predicts the next input $\hat{\mathbf{x}}_{t+1}$. The attention update rule is given using SDM notation in Eq. 4. To maximize clarity, we present every operation performed on the vectors through a single Transformer block. In GPT3, the largest GPT model to date, there are 175 billion parameters and 96 repeating hierarchical blocks of which one is depicted. To emphasize the Attention operation, we use arrows showing how one of the Attention heads retrieves patterns as inputs. In the first layer these inputs have just received their initial embedding, in the later layers they use their transformed pattern representations created from when each input was itself the query. See https://jalammar.github.io/illustrated-gpt2/ and https://jalammar.github.io/illustrated-transformer/ for excellent overviews of GPT2 and the Transformer, respectively.

## B SDM Details

### B.1 Circle Intersection Calculation

The SDM book [13] presents its circle intersection equation in its Appendix B, taking up a few pages of quite involved proofs. It then derives a continuous approximation to the exact equation that is then used throughout the book (this equation still operates in the binary vector space and is unrelated to the continuous $L^2$ normed vector space version of SDM we develop that is outlined in Appendix B.3). Following inspiration from [27, 44], here we derive a more interpretable version of the exact circle intersection equation.

In confirming that the exact equation we derive matches with the exact equation derived in the book (it does), we discovered that the circle intersection results actually presented in the book that claim to have come from the continuous approximation to this exact equation are slightly inaccurate as shown in Fig. 10 [13]. We don't know for sure why this is the case but we suspect it has to do with rounding errors during the calculations.[10] When implemented, the exact calculation gives values that are larger than those presented in the book, while the continuous approximation gives values that are smaller. Ensuring these equations are exact is crucial for fitting $\beta$ to the circle intersection in our Attention approximation, computing optimal $d^*$ values, and comparing SDM to Attention in our experiments of Appendix B.7.

| | $d_v$ | Exact | Continuous Approx. | SDM Book |
|---|---|---|---|---|
| 0 | 0 | 1071.850049 | 970.769893 | 1000.0 |
| 1 | 1 | 958.436527 | 888.214545 | 894.0 |
| 2 | 10 | 795.873633 | 713.056676 | 743.0 |
| 3 | 50 | 476.873676 | 426.702117 | 445.0 |
| 4 | 100 | 285.830559 | 254.636411 | 267.0 |
| 5 | 150 | 173.627261 | 153.930182 | 162.0 |
| 6 | 200 | 103.460716 | 91.252432 | 97.0 |
| 7 | 300 | 32.323798 | 28.196980 | 30.0 |
| 8 | 400 | 7.597126 | 6.553410 | 7.0 |
| 9 | 500 | 1.101265 | 0.942394 | 1.0 |

Figure 10: The expected circle intersection sizes showing the values in the SDM book are in between the values for the exact equation and its continuous approximation. This discrepancy is important as the size of the circle intersection is foundational to the results of this work. We think the difference is due to rounding errors. These expected circle intersections use the parameters $n = 1000$, $r = 1,000,000$ and $d = 451$.

Implementing the exact SDM lune equation of the book's Appendix B ourselves gives the same results as our algorithm, and both give the same results as simulations of SDM shown in Fig. 11 so we believe this is the correct and exact algorithm. We present it's derivation below:

We are given two arbitrary $n$ dimensional binary query and pattern vectors $\boldsymbol{\xi}$ and $\mathbf{p}_a$ that will read and write using a Hamming distance of $d$ and have a vector distance of $d_v = d(\boldsymbol{\xi}, \mathbf{p}_a)$. We can group the elements of $\boldsymbol{\xi}$ and $\mathbf{p}_a$ into two categories: where they agree (of which there are $n - d_v$ elements) and where they disagree ($d_v$ elements).

$$
\begin{aligned}
\boldsymbol{\xi} &= [\overbrace{1,\ldots,1 \mid 0,\ldots,0}^{n-d_v} \mid \overbrace{1,\ldots,1 \mid 0,\ldots,0}^{d_v}] \\
\mathbf{p}_a &= [\underbrace{1,\ldots,1 \mid 0,\ldots,0}_{a+z} \mid \underbrace{0,\ldots,0 \mid 1,\ldots,1}_{b+c}]
\end{aligned}
\tag{11}
$$

---

[10]For example, when $n = 1000$ and $d = 451$, the fraction of the vector space occupied by the Hamming threshold is $p = 0.00107$. This means that when the query and target pattern are in the same location such that $d_v = 0$, in expectation we should see $p * r$ neurons in this intersection which when $r = 1,000,000$ gives us 1071 for the exact calculation. In [13] this $p$ is rounded $p = 0.001$ and $p * r$ then gives the 1000 shown in Fig. 10.

Now imagine a third vector, representing a hypothetical neuron address. This neuron has four possible groups that the elements of its vector can fall into:

- $a$ - agree with *both* $\boldsymbol{\xi}$ and $\mathbf{p}_a$
- $z$ - disagree with *both* $\boldsymbol{\xi}$ and $\mathbf{p}_a$
- $b$ - agree $\boldsymbol{\xi}$ disagree $\mathbf{p}_a$
- $c$ - agree $\mathbf{p}_a$ disagree $\boldsymbol{\xi}$

We want to compute the possible values that $a$, $z$, $b$ and $c$ can have such that it exists inside the circle intersection between $\boldsymbol{\xi}$ and $\mathbf{p}_a$. This produces the following constraints:

$$a + b + c + z = n$$
$$a + b \geq n - d$$
$$a + c \geq n - d$$
$$a + z = n - d_v$$
$$b + c = d_v$$

These constraints exist because the vector must be $n$ dimensional, it must have more than $n - d_v$ agreements with $\boldsymbol{\xi}$ to be in its Hamming threshold, and the same for $\mathbf{p}_a$. In total it can only agree or disagree with both $\boldsymbol{\xi}$ and $\mathbf{p}_a$ simultaneously at the $n - d_v$ where they agree. Similarly, it can only disagree with one of them but not the other at the $d_v$ places where $\boldsymbol{\xi}$ and $\mathbf{p}_a$ disagree.

We can derive bounds for $a$ and $c$ and use these values to compute the total possible number of neurons that exist in the neuron intersection and the fraction of the total $2^n$ vector space occupied. Because $a$ determines $z$ and $c$ determines $b$, we only need to use $a$ and $c$.

$$n - d - \lfloor \frac{d_v}{2} \rfloor \leq a \leq n - d_v$$
$$\max(0, n - d - a) \leq c \leq d_v - (n - d - a)$$

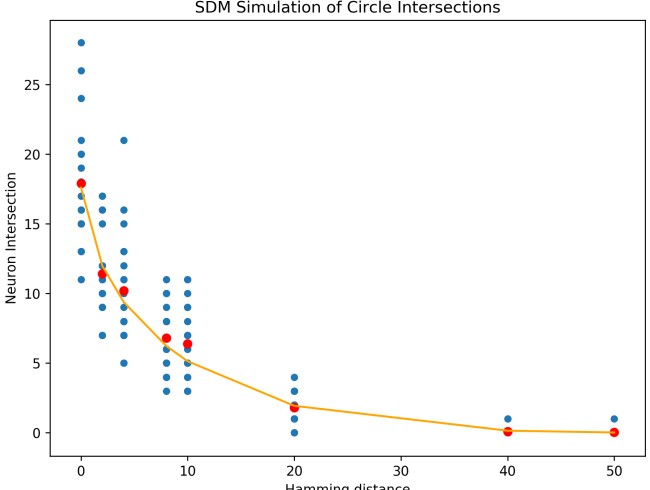

Figure 11: The size of the circle intersection for two vectors at different Hamming distances using simulations vs expectations from Eq. 2 for SDM. We ran 20 simulations with random initialization of all vectors each time for all neuron and pattern addresses. We randomly generated two addresses $d_v$ Hamming distance apart and computed the neuron intersection. Each simulation is a blue dot, the red dot is the mean, and the orange line is the analytical expectation from Eq. 2. For faster simulations, we set $n = 100$, $r = 10,000$, $m = 100$ and used Hamming distance $d = 35$ that activates $p = 0.0017$ of the space.

This results in the expected circle intersection equation:

$$|O_n(\boldsymbol{\xi}, d) \cap O_n(\mathbf{p}_a, d)| = \sum_{a=n-d-\lfloor \frac{d_v}{2} \rfloor}^{n-d_v} \sum_{c=\max(0, n-d-a)}^{d_v-(n-d-a)} \left( \binom{n-d_v}{a} \cdot \binom{d_v}{c} \right) \tag{12}$$

If we multiply $|O_n(\boldsymbol{\xi}, d) \cap O_n(\mathbf{p}_a, d)|$ by $\frac{r}{2^n}$ then we can get the expected number of neurons that will exist in this circle intersection (assuming they are randomly distributed throughout the space).

## B.2   Analytical Circle Intersection Exponential Approximation

In this section we more formally derive an analytical exponential approximation to the circle intersection with moderate success. Our empirical results from both simulations (eg. Fig 3) and real world datasets across a range of parameters (Appendix B.7) bolster these results.

Equation 2 for the circle intersection that is derived in Appendix B.1 is a summation of the product of binomial coefficients. We can approximate this equation using the largest element of the summation and converting the binomial coefficients into a binomial distribution. We can then approximate the binomial distribution with a normal distribution, which has exponential terms and is tight for the high dimensional $n$ values used with SDM. The idea to use the normal approximation is credited to the SDM book [13].

It is important to note that these exponential bounds will only hold when $d_v \leq 2d$. Otherwise, the size of the circle intersection is 0, making an exponential approximation impossible. However, there are two reasons this is not an issue:

First, the patterns that are close to the query matter the most and this is where the circle intersection approximates the exponential best. More distant patterns don't matter because they approach zero when normalized. Fig. 3, reproduced here as Fig. 12 for convenience, shows this. Looking at the inset plot first that is in log space, the binary and continuous circle intersections in blue and orange follow the green exponential line closely until around $d_v = 20$ at which point they become super exponential. In the main plot it is clear that by the time $d_v = 20$, the normalized weighting to each point is $\approx 0$ for both the softmax and circle intersections. There is empirical validation of this effect from experiments where the $\beta$ fitted Attention approximation to SDM performs as well as the circle intersection in autoassociatively retrieving patterns from noisy queries as investigated in Appendix B.7.

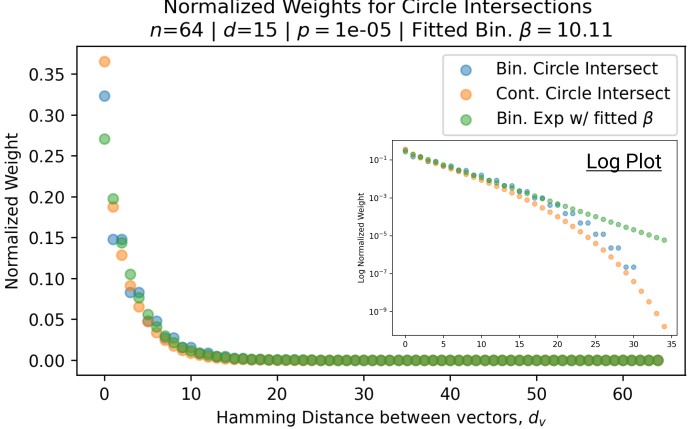

Figure 12: Reproduction of Fig. 3 to show how the point at which the circle intersection is no longer exponential, around $d_v = 20$ is when the normalized values $\approx 0$.

The second reason SDM only having an exponential approximation up to $d_v < 2d$ is not an issue applies to all instantiations of SDM that are either biologically plausible (where $r \ll 2^n$) or use

critical distance optimal $d^*$ in high $n$ dimensions, such that $d$ is close to $n/2$ in size.[11] This means that $d_v > 2d$ only for very large $d_v$ distances close to $n$ in value that are exceedingly rare because this is much greater than the orthogonal distance of $n/2$ almost all random vectors exist at.[12] Table 1 of Appendix B.5 shows optimal $d^*$ using canonical SDM parameters where $n = 1,000$, $r = 1,000,000$, $m = 10,000$, and all versions of $d^*$ are close to $n/2$.

The circle intersection Eq. 2 reproduced below as Eq. 13 for convenience accounts for the fact that if $d_v > 2d$, the intersection is 0 through its bounds on $a$. Specifically, for $d_v > 2d$, the lower bound will be larger than the upper bound on $a$ meaning there is no summation.

$$|O_n(\boldsymbol{\xi}, d) \cap O_n(\mathbf{p}_a, d)| = \sum_{a=n-d-\lfloor \frac{d_v}{2} \rfloor}^{n-d_v} \sum_{c=\max(0, n-d-a)}^{d_v - (n-d-a)} \left( \binom{n-d_v}{a} \cdot \binom{d_v}{c} \right) \tag{13}$$

The binomial coefficient's equality with the Binomial distribution is:

$$\text{Binomial}(k, n, p) = \binom{n}{k} p^k (1-p)^{n-k}$$

$$\text{Binomial}(k, n, p = 1/2) = \binom{n}{k} \left( \frac{1}{2} \right)^n$$

$$\binom{n}{k} = 2^n \cdot \text{Binomial}(k, n, p = 1/2)$$

We can approximate the Binomial, which has a mean of $np$ and variance of $np(1-p)$ with a standard Gaussian denoted $N(\cdot)$:

$$\text{Binomial}(k, n, p = \frac{1}{2}) \approx N\left( \frac{k - \frac{n}{2}}{\sqrt{\frac{n}{4}}} \right) = \frac{1}{\sqrt{2\pi}\sqrt{\frac{n}{4}}} \exp\left( -\frac{1}{2} \left( \frac{k - \frac{n}{2}}{\sqrt{\frac{n}{4}}} \right)^2 \right)$$

Therefore, we can view the binomial coefficients in the circle intersection Eq. 13 as a sum of the product of exponentials:

$$\binom{n-d_v}{a} \cdot \binom{d_v}{c} \approx 2^{n-d_v} \cdot N\left( \frac{a - \frac{n-d_v}{2}}{\sqrt{\frac{n-d_v}{4}}} \right) \cdot 2^{d_v} \cdot N\left( \frac{c - \frac{d_v}{2}}{\sqrt{\frac{d_v}{4}}} \right) \tag{14}$$

Because the product of binomial coefficients are always $\geq 0$ we can approximate this sum by using its largest element as a lower bound. The largest element is when $a$ is as close to $\frac{n-d_v}{2}$ as possible and $c$ is as close to $\frac{d_v}{2}$ (binomial coefficients are largest when $k = n/2$ for $\binom{n}{k}$). Because $a$ is bounded from below by $n - d - \lfloor \frac{d_v}{2} \rfloor$ and $d < \frac{n}{2}$,[13] it is the case that $\frac{n-d_v}{2} < n - d - \lfloor \frac{d_v}{2} \rfloor$ so the closest $a$ can get to $n - \frac{d_v}{2}$ is its lower bound. When $a$ is at its lower bound the only value that $c$ can take is the value that maximizes its binomial coefficient of $\frac{d_v}{2}$:

$$\max(0, n - d - a) \leq c \leq d_v - (n - d - a)$$

$$n - d - (n - d - \lfloor \frac{d_v}{2} \rfloor) \leq c \leq d_v - (n - d - (n - d - \lfloor \frac{d_v}{2} \rfloor))$$

$$\lfloor \frac{d_v}{2} \rfloor \leq c \leq \lfloor \frac{d_v}{2} \rfloor \tag{15}$$

For the upper bound on $c$, if $d_v$ is even then this equals $\lfloor \frac{d_v}{2} \rfloor$, if it is odd then it equals $\lfloor \frac{d_v}{2} + 0.5 \rfloor$ which gives a bound that rounds down to the nearest integer, still giving us $\lfloor \frac{d_v}{2} \rfloor$. Therefore, the

---

[11]Neither of these cases are true for the usual Attention parameters where $n = 64$ is quite low dimensional and $r = 2^n$, but the approximation still holds well for the first reason provided above.

[12]$d$ being close to $n/2$ is more common as $n$ increases because there is the Blessing of Dimensionality where a greater fraction of the vector space is closer to being orthogonal (see [13, 15] for a discussion of this effect). Greater orthogonality means an increase in $n$ causes a super-linear increase in $d$ for a given $p$ because to capture the same fraction $p$ of the space, $d$ must be that much closer to $n/2$.

[13]We never want to read or write to vectors that are greater than the orthogonal distance.

largest element of the sum, where both binomial coefficients are as large as they can be given their constraints, is when $a = n - d - \lfloor \frac{d_v}{2} \rfloor$ and $c = \lfloor \frac{d_v}{2} \rfloor$, plugging these values into Eq. 14 we get:

$$\binom{n - d_v}{n - d - \lfloor \frac{d_v}{2} \rfloor} \cdot \binom{d_v}{\lfloor \frac{d_v}{2} \rfloor} \approx 2^n \cdot N\left( \frac{(n - d - \lfloor \frac{d_v}{2} \rfloor) - \frac{n - d_v}{2}}{\sqrt{\frac{n - d_v}{4}}} \right) \cdot N\left( \frac{\lfloor \frac{d_v}{2} \rfloor - \frac{d_v}{2}}{\sqrt{\frac{d_v}{4}}} \right) \tag{16}$$

$$= 2^n \cdot \frac{1}{\sqrt{2\pi} \sqrt{\frac{n - d_v}{4}}} \exp\left( -\frac{1}{2} \left( \frac{(n - d - \lfloor \frac{d_v}{2} \rfloor) - \frac{n - d_v}{2}}{\sqrt{\frac{n - d_v}{4}}} \right)^2 \right)$$

$$\cdot \frac{1}{\sqrt{2\pi} \sqrt{\frac{d_v}{4}}} \exp\left( -\frac{1}{2} \left( \frac{\lfloor \frac{d_v}{2} \rfloor - \frac{d_v}{2}}{\sqrt{\frac{d_v}{4}}} \right)^2 \right) \tag{17}$$

$$\approx \frac{2^{n+1}}{\pi \sqrt{d_v(n - d_v)}} \exp\left( -\frac{(n - 2d)^2}{2n} \cdot \frac{1}{1 - \frac{d_v}{n}} \right) \tag{18}$$

$$\approx \frac{2^{n+1}}{\pi \sqrt{d_v(n - d_v)}} \exp\left( -\frac{(n - 2d)^2}{2n} \cdot \left( 1 + \frac{d_v}{n} + \mathcal{O}\left( \left( \frac{d_v}{n} \right)^2 \right) \right) \right)$$

$$= \frac{2^{n+1}}{\pi \sqrt{d_v(n - d_v)}} \exp\left( -\frac{(n - 2d)^2}{2n} \right)$$

$$\cdot \exp\left( -\frac{(n - 2d)^2}{2n^2} \cdot d_v \right) \cdot \epsilon\left( \frac{d_v}{n} \right) \tag{19}$$

$$\approx \frac{2^{n+2}}{\pi n} \exp\left( -\frac{(n - 2d)^2}{2n} \right) \cdot \exp\left( -\frac{(n - 2d)^2}{2n^2} \cdot d_v \right) \tag{20}$$

Moving from line 17 to 18 we set $\lfloor \frac{d_v}{2} \rfloor \approx \frac{d_v}{2}$ which is exact for even $d_v$, but an approximation off by 0.5 for odd $d_v$. This approximation allows many of the $d_v$ terms to cancel. Going from line 18 to 19 we use a first order Taylor expansion at $d_v = 0$ to make the exponential be a function of $d_v$ where it is in the numerator. This approximation is accurate for $\frac{d_v}{n} \ll 1$. We use $\epsilon(d_v/n)$ to denote the higher order terms for the Taylor expansion that are dropped going to the final line Eq. 20. During this final step we also remove $d_v$ from the constant outside the exponential by lower bounding it which occurs when $d_v = \frac{n}{2}$:

$$\min_{d_v} \frac{2^{n+1}}{\pi \sqrt{d_v(n - d_v)}} = \frac{2^{n+2}}{n\pi}.$$

Therefore, Eq. 20 denotes the exponential approximation to the largest sum in the circle intersection calculation.

To test the quality of this approximation, we plot the full circle intersection, the value of the largest sum of binomial coefficients (the left hand side of Eq. 16), the Normal approximation (Eq. 17), and the Taylor approximation (Eq. 20), which is the final desired exponential approximation to the circle intersection. The exponential approximation of Eq. 20 to the circle intersection makes a number of approximations that reduce its precision, in particular the Taylor approximation, which assumes $d_v = 0$ so we restrict our analysis to $d_v < 0.1n$. However, we stated earlier that it is these patterns closest to the queries that matter the most in approximating the softmax and Attention. Figure 13 shows the Attention setting where $n = 64$ and $r = 2^n$ with a wide range of $d$ values and their corresponding space fractions $p$. Figure 14 is the same plot but with the canonical SDM setting where $n = 1,000$ and $r = 1,000,000$. To convert to the continuous case, we can use the mapping from cosine similarity to Hamming distance given in Eq. 5 where we assume the query and key vectors being used ($\xi$ and $\mathbf{p}_a$) are $L^2$ normalized and re-write our approximation as:

$$\frac{2^{n+2} \exp(-\frac{3(n - 2d)^2}{4n})}{n\pi \sqrt{1 - (\hat{\mathbf{p}}_a^T \hat{\xi})^2}} \exp\left( \frac{(n - 2d)^2}{4n} \cdot \hat{\mathbf{p}}_a^T \hat{\xi} \right) \tag{21}$$

And the same lower bounding of the non exponential terms can be performed to remove the cosine similarity $\hat{\mathbf{p}}_a^T \hat{\xi}$ from them. This concludes how the circle intersection can be bounded by exponential functions.

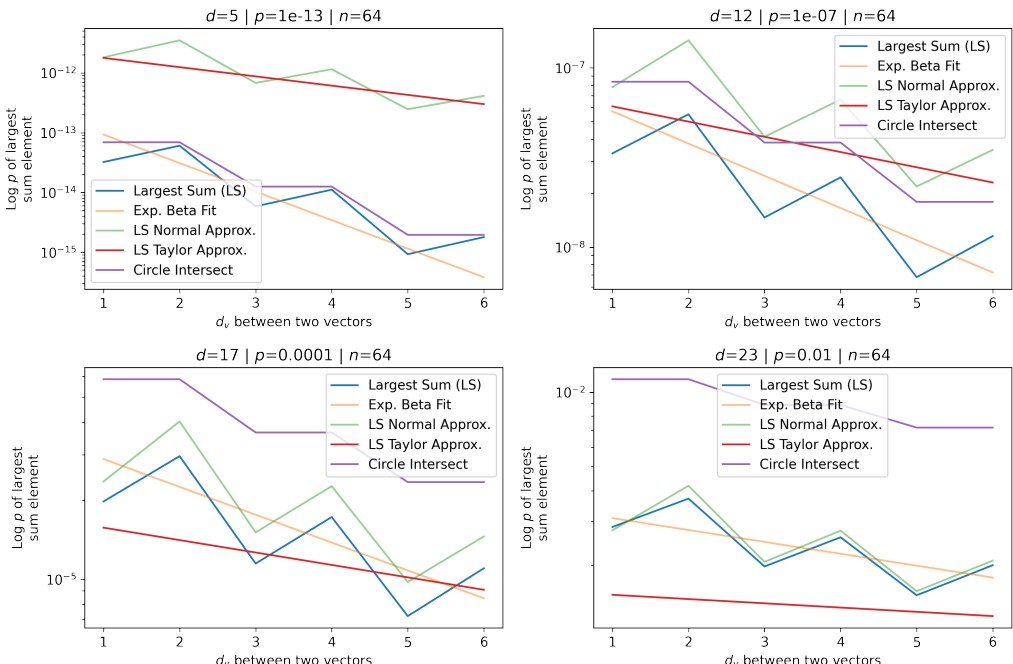

Figure 13: Showing the largest summation term of the circle intersection and its exponential approximation in comparison to the full circle intersection for a large range of $d$ values when $n = 64$. Blue is the largest sum, represented on left hand side of Eq. 16. Orange is a log linear regression fit to this line using Eq. 10 to show how this first summation term, like the full circle intersection is approximately exponential. Green is the Normal approximation in Eq. 18 to the largest sum. It slightly upper bounds the real largest sum value. Red is the Taylor approximation in Eq. 20 that becomes less accurate as $d_v$ on the x-axis increases. Purple is the full circle intersection equation to show how well the largest sum approximates it as a lower bound. It is a tight bound for smaller Hamming $d$ radii when there are fewer other terms to sum. The y-axis gives the log fraction of the vector space $p$ occupied by the largest sum (this is independent of $n$). The x-axis is up to values of $0.1n$ where the Taylor approximation holds well.

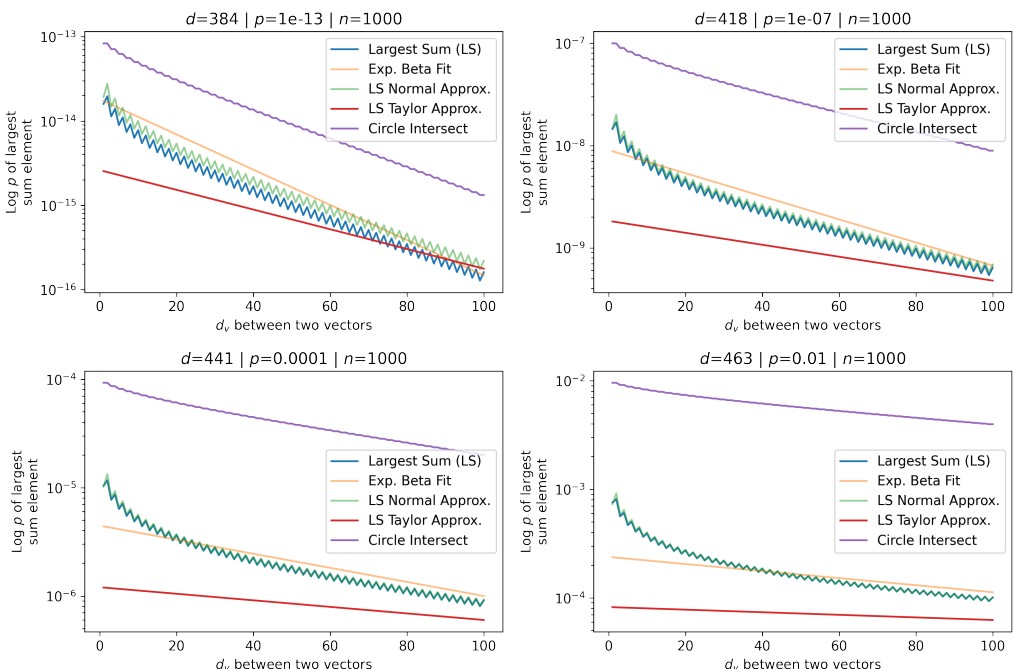

Figure 14: The same as figure 13 but using the canonical SDM setting where $n = 1,000$.

## B.3 Continuous SDM Circle Intersection

Using SDM with continuous, $L^2$ normalized vectors there are two ways to compute the number of neurons in the circle intersection. One option is computing cosine similarity between two points and mapping this back to Hamming distance using Eq. 5 before computing the circle intersection using Eq. 2. However, mapping back to Hamming distance discretizes many intermediate cosine similarities, we leave it to future work to explore under what regimes this discretization helps convergence by reducing noise or harms it by removing signal.

The other way to compute the circle intersection is in continuous space. However, unlike in the binary space where our vectors were unconstrained (they could be the all ones vector or all zeros), in continuous space all vectors are $L^2$ normed to exist on the surface of an $n$-dimensional hypersphere. This means the number of neurons in the circle intersection calculation is not the volume of intersection between two hyperspheres but instead only the intersection that exists on the surface of the hypersphere. In other words, we want the surface area of the intersection between two hyperspherical caps. Figure 15 shows the difference between the binary and continuous calculations geometrically in 3-dimensional space.

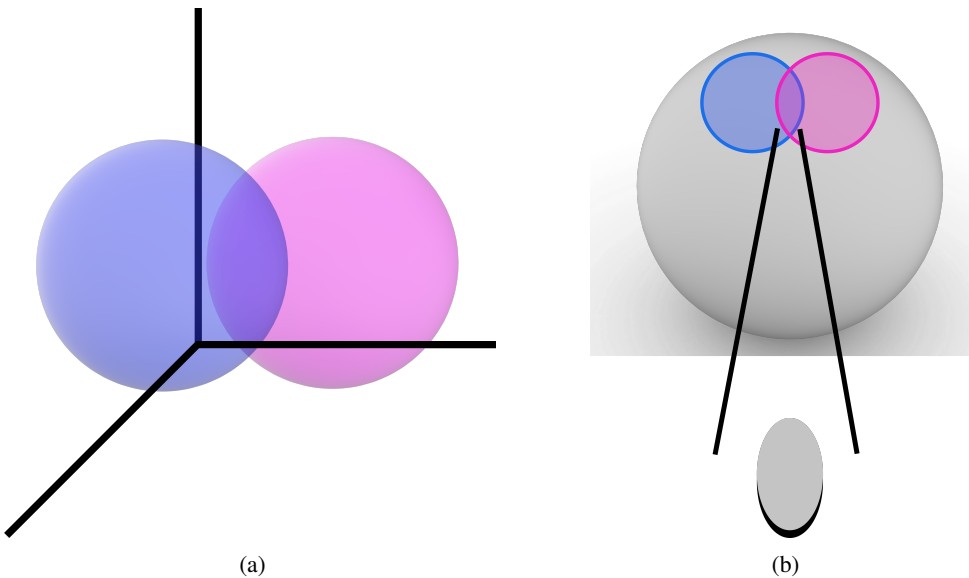

(a)                                                              (b)

Figure 15: Using 3-dimensions to depict differences in the circle intersection calculations for binary versus continuous SDM. **(a)** Binary SDM circle intersection. The intersection of hyperspheres used in binary SDM where vectors are unconstrained and can exist anywhere in the binary vector space including the all ones and all zeros vectors. **(b)** Continuous SDM circle intersection. All neurons are L2 normed so they exist on the surface of the gray sphere along with all patterns and queries. The read and write cosine similarity radii correspond to an $n - 1$ dimensional surface area depicted as the blue and pink circles on the surface of the sphere. Geometrically, neurons in this cosine radius can be represented as the cap of a cone going from the center to the surface of the sphere. We care about computing the intersection between these two cone caps.

It is important to note that due to the curvature of the hypersphere and the cone caps that define the circle intersection, the intersection continues to exist even after $d_v > 2d$ at which point the circle intersection disappears in the binary setting. This is shown both in Fig. 3 and our experiments in Appendix B.7.

Equations have been derived for this continuous circle intersection in [77] and are reproduced below. We first need to convert our two $L^2$ normalized vectors $\hat{\mathbf{a}}$ and $\hat{\mathbf{b}}$ into angles to define the size of the cones and their caps:

$$\theta_1 := \theta_2 = \arccos\left(1 - \frac{2d}{n}\right)$$

$$\theta_v := \arccos\left(\hat{\mathbf{a}}^T\hat{\mathbf{b}}\right)$$

$$\theta_{\min} := \arctan\left(\frac{\cos(\theta_1)}{\cos(\theta_2)\sin(\theta_v)} - \frac{1}{\tan(\theta_v)}\right)$$

$$l_2 = 1$$

where $\theta_1$ and $\theta_2$ map the Hamming distance radius into cosine similarity and then into the corresponding angle in radians. $\theta_v$ is the angle in radians between the two vectors. $\theta_{\min}$ is an intermediate calculation that is the radians between one of the vectors and the hyperplane that goes through their intersection. And $l_2$ is the size of the $L^2$ norm of our vectors which is equal to 1 in our case.

The full equation to get the size of the intersection is:

$$\mathcal{I}_c\left(\hat{\mathbf{p}}_a^T\hat{\boldsymbol{\xi}}, 1 - \frac{2d}{n}, n\right) := A_n(\theta_v, \theta_1, \theta_2, n, l_2 = 1) = J_n(\theta_{\min}, \theta_2, l_2) + J_n(\theta_v - \theta_{\min}, \theta_1, l_2) \tag{22}$$

$$J_n(\theta_a, \theta_b, l_2) = \frac{\pi^{\frac{n-1}{2}}}{\Gamma\left(\frac{n-1}{2}\right)} l_2^{n-1} \int_{\theta_a}^{\theta_b} \sin(\phi)^{n-2} I_{1-\left(\frac{\tan(\theta_a)}{\tan(\phi)}\right)^2}\left(\frac{n-2}{2}, \frac{1}{2}\right) \mathrm{d}\phi$$

where $\Gamma(\cdot)$ is the Gamma function and $I(\cdot)$ is the regularized incomplete Beta function.

To compute the expected number of randomly distributed neurons present in the computed circle intersection, we compute the total surface area of the hypersphere:

$$SA_n(l_2) = \frac{2\pi^{\frac{n}{2}}}{\Gamma\left(\frac{n}{2}\right)} l_2^{n-1} \tag{23}$$

And compute the number of neurons expected in our intersection as:

$$r\frac{\mathcal{I}_c\left(\hat{\mathbf{p}}_a^T\hat{\boldsymbol{\xi}}, 1 - \frac{2d}{n}, n\right)}{SA_n(l_2 = 1)}.$$

We do not attempt to derive an analytical exponential approximation to this continuous circle intersection. However, we do compare its weights to Attention softmax both in simulations (Fig. 3) and experiments with the MNIST and CIFAR10 datasets (Appendix B.7).

### B.4  SDM Variance Calculation and Signal to Noise Ratio Derivation

Here, we explain SDM's Signal to Noise Ratio (SNR) and the variance equation used to compute it that was introduced in [27]. This variance equation is a more accurate approximation than the original in the SDM book [13] (outlined in its Appendix C). This claim has been validated through simulations shown in Figure 16 and detailed at the bottom of this Section. Because it is more accurate, we present it here and use it through this work for the convergence and signal to noise ratio calculations. Notably, Kanerva's summary of SDM published in 1993 [15] also uses this improved variance equation.

Signal to noise ratio (SNR) is used to compute query convergence dynamics to a target pattern and the maximum memory capacity of SDM when we assume random patterns. As a result, the SNR is also used to compute all of the optimal Hamming distance variants, with details outlined in Appendix B.5. For a full in depth description of SDM convergence dynamics including analysis when the patterns written into memory are themselves noisy we defer to [78]. Note that while the following convergence results are important for this work, the most relevant results are those of SDM extensions and the Attention SDM approximation, which can leverage results derived from the Modern Continuous Hopfield Network paper that we defer to [25].

The signal in the SNR refers to the target pattern $p^*$ that a noisy query wants to converge to. The noise refers to interference from all other patterns present.

$$\text{SNR} = \frac{\text{expected magnitude of target pattern's circle intersection}}{\text{standard deviation of all pattern circle intersections}}$$

Taking an arbitrary element $i$ from our $n$ dimensional vector as they are all independent (assuming random patterns) the query update operation produces a weighted summation, denoted $\Sigma$:

$$\Sigma = \sum_{\mu=0}^{m} I_\mu i_\mu \tag{24}$$

where $I_\mu$ is the size of the intersection between the query and the pattern (using Eq. 2) indexed by $\mu$, and $i_\mu$ is the value that the pattern has at this position. In the equations that follow we use a $*$ to represent the target pattern we want the query to converge to. Also, to make the calculations easier we assume that the pattern values are bipolar $i \in \{-1, 1\}$ and that $i_*$ is known. These assumptions mean that $i_*^2 = 1$ and $E[i_\mu] = 0, \forall \mu \neq *$ because they are random and symmetric about zero.

We can re-write Eq. 24 splitting out the signal (related to the target pattern) and noise components as:

$$\Sigma = \mathbb{E}[I_*]i_* + (I_* - \mathbb{E}[I_*])i_* + \sum_{\mu \neq *} I_\mu i_\mu$$

Note that the neuron intersection $I$ here depends upon the Hamming distance $d_v$ between the query and the specific pattern being indexed. For example, the target pattern $p^*$ may be 100 bits away in which case $I_* = I(100)$ and all of the random patterns may be 500 bits away making $I_\mu = I(500), \forall \mu \neq *$.

To compute the SNR, we want to find $\mathbb{E}[\Sigma]$ and $\sqrt{Var(\Sigma)}$ for the numerator and denominator, respectively.

$$\mathbb{E}[\Sigma] = \mathbb{E}[I_* i_*] + \mathbb{E}[(I_* - \mathbb{E}[I_*])i_*] + \sum_{\mu \neq *} \mathbb{E}[I_\mu i_\mu]$$

$$= \mathbb{E}[I_*]i_* + (\mathbb{E}[I_*] - \mathbb{E}[I_*])i_* + \sum_{\mu \neq *} \mathbb{E}[I_\mu]\,\mathbb{E}[i_\mu]$$

$$= \mathbb{E}[I_*]i_* + 0 + 0$$

Turning to the variance, the variance of a sum is the sum of variances plus the pairwise covariances of the summands:

$$Var(\Sigma) = Var(\mathbb{E}[I_*]i_*) + Var((I_* - \mathbb{E}[I_*])i_*) + \sum_{\mu \neq *} Var(I_\mu i_\mu)$$

$$+ \sum_{\mu \neq *} Cov\Big((I_* - \mathbb{E}[I_*])i_*, I_\mu i_\mu\Big) + \sum_{a \in \mu \neq *,\, b \in \mu \neq * \cap b \neq a} Cov\Big(I_a i_a, I_b i_b\Big)$$

However, we will now show that with i.i.d $i_\mu \in \{-1, 1\}, \forall \mu \neq *$ all covariance terms are 0 leaving just the sum of variances:

$$\sum_{\mu \neq *} Cov\Big((I_* - \mathbb{E}[I_*])i_*, I_\mu i_\mu\Big) = \sum_{\mu \neq *} \mathbb{E}\Big[(I_* - \mathbb{E}[I_*])I_\mu i_\mu\Big]i_* - \mathbb{E}\Big[(I_* - \mathbb{E}[I_*])i_*\Big]\mathbb{E}[I_\mu i_\mu]$$

$$= \sum_{\mu \neq *} \mathbb{E}\Big[(I_* - \mathbb{E}[I_*])I_\mu\Big]\mathbb{E}[i_\mu]i_* - \mathbb{E}\Big[(I_* - \mathbb{E}[I_*])i_*\Big]\mathbb{E}[I_\mu]\,\mathbb{E}[i_\mu]$$

$$= \sum_{\mu \neq *} \mathbb{E}\Big[(I_* - \mathbb{E}[I_*])I_\mu\Big] * 0 * i_* - \mathbb{E}\Big[(I_* - \mathbb{E}[I_*])i_*\Big]\mathbb{E}[I_\mu] * 0$$

$$= 0 - 0 = 0$$

Looking at the other covariance term:

$$\sum_{a \in \mu \neq *,\, b \in \mu \neq * \cap b \neq a} Cov\Big(I_a i_a, I_b i_b\Big) = \sum_{a \in \mu \neq *,\, b \in \mu \neq * \cap b \neq a} \mathbb{E}[I_a i_a I_b i_b] - \mathbb{E}[I_a i_a]\,\mathbb{E}[I_b i_b]$$

$$= \sum_{a \in \mu \neq *,\, b \in \mu \neq * \cap b \neq a} \mathbb{E}[I_a]\,\mathbb{E}[i_a]\,\mathbb{E}[I_b]\,\mathbb{E}[i_b] - \mathbb{E}[I_a]\,\mathbb{E}[i_a]\,\mathbb{E}[I_b]\,\mathbb{E}[i_b]$$

$$= 0 - 0 = 0$$

Now, being able to ignore the covariance equations:

$$Var(\Sigma) = Var(\mathbb{E}[I_*]i_*) + Var\Big((I_* - \mathbb{E}[I_*])i_*\Big) + \sum_{\mu \neq *} Var(I_\mu i_\mu)$$

$$= 0 + Var\Big((I_* - \mathbb{E}[I_*])i_*\Big) + \sum_{\mu \neq *} Var(I_\mu i_\mu)$$

Looking at each term of this sum:

$$Var\Big((I_* - \mathbb{E}[I_*])i_*\Big) = i_*^2 Var\Big(I_* - \mathbb{E}[I_*]\Big) = Var\Big(I_* - \mathbb{E}[I_*]\Big)$$

$$Var(I_* - \mathbb{E}[I_*]) = Var(I_*)$$

And:

$$Var(I_\mu i_\mu) = \mathbb{E}\left[(I_\mu i_\mu)^2\right] - \Big(\mathbb{E}[I_\mu i_\mu]\Big)^2$$

$$= \mathbb{E}[I_\mu^2 i_\mu^2] - \Big(\mathbb{E}[I_\mu]\mathbb{E}[i_\mu]\Big)^2$$

$$= \mathbb{E}[I_\mu^2] - 0$$

Re-arranging using the variance equation:

$$\mathbb{E}[I_\mu^2] = Var(I_\mu) + \mathbb{E}[I_\mu]^2$$

and bringing everything together to compute the variance we have:

$$Var(\Sigma) = Var(I_*) + \sum_{\mu \neq *} Var(I_\mu) + \mathbb{E}[I_\mu]^2$$

We can assume that $I_\mu \sim$ Poisson, $\forall \mu$ and are i.i.d because a random pattern when $n$ is large will be very close to orthogonal and only have a small number of neuron intersections. This approximation allows us to state that $\mathbb{E}[I_\mu] = Var(I_\mu)$ and when $n$ is large, make the further assumption that all patterns $\mu \neq *$ are at the orthogonal distance $d_v^{\text{ortho}} = n/2$ leading to:

$$Var(\Sigma) = \mathbb{E}[I_*] + \sum_{\mu \neq *} \mathbb{E}[I_\mu] + \mathbb{E}[I_\mu]^2$$

$$Var(\Sigma) = \mathbb{E}[I_*] + (m-1)\big(\mathbb{E}[I_\mu] + \mathbb{E}[I_\mu]^2\big)$$

This Poisson approximation works well for $\mu \neq *$ but is a lower bound for $Var(I_*)$. In addition, not all random patterns will be at the orthogonal distance, with higher variance coming from the few patterns that happen to be closer and further away. Both of these factors mean we are underestimating the variance and overestimating the SNR, but simulations outlined in Fig. 16 show that this approximation is still quite accurate.

Finally, we can write the SNR equation as:

$$\text{SNR}(d_v^*, d_v^{\text{ortho}}, d, r, m, n) \approx \frac{\mathbb{E}[I_*]}{\sqrt{\mathbb{E}[I_*] + (m-1)\big(\mathbb{E}[I_{\mu \neq *}] + \mathbb{E}[I_{\mu \neq *}]^2\big)}} \tag{25}$$

We have introduced the parameters $d_v^*$, $d_v^{\text{ortho}}$, $d$, $r$, $m$, and $n$ as inputs into the SNR equation as these determine the target and random pattern intersection calculations $I_*$ and $I_{\mu \neq *}$. Recall that because we are using a lower bound on the variance, this is an upper bound on the SNR.

If we want to compute the number of memories that SDM can store in absolute terms, we must set a retrieval probability: the probability that a bit of our $n$ dimensional query is decoded correctly to the target pattern. With $n$ being large we can treat the SNR as being a standard normal distribution and use Z-scores to compute the probability of retrieval. For example, a 99% probability corresponds to $z = 2.33$ and we can write:

$$P\Big(\frac{\mathbb{E}[I_*]}{\sqrt{\mathbb{E}[I_*] + (m-1)\big(\mathbb{E}[I_{\mu \neq *}] + \mathbb{E}[I_{\mu \neq *}]^2\big)}} > z\Big)$$

and solve for $m$:

$$m = \frac{(\frac{\mathbb{E}[I_*]^2}{z^2} - \mathbb{E}[I_*])}{\mathbb{E}[I_{\mu \neq *}] + \mathbb{E}[I_{\mu \neq *}]^2} + 1 \qquad (26)$$

In order to have the z-score correspond to the probability of retrieval for the full length $n$ pattern, we use the equation:

$$z = F^{-1}(\text{prob}^{1/n}) \qquad (27)$$

where $F^{-1}()$ is the inverse CDF of a standard Gaussian and "prob" is the probability that we want for the whole target pattern to be decoded correctly. Equations 25 and 26 are used to compute different variants of optimal Hamming distances in Appendix B.5.

To compute the accuracy of the SNR approximation we ran simulations of SDM. This shows that, as expected, our approximation was a lower bound but a tight one. We ran SDM 200 times, each time randomly initializing the neuron, pattern and query vectors and using the parameters $n = 100$, $r = 10,000$, $m = 100$ and $d = 35$. We had every pattern write to each neuron and then computed the variance in the neuron value vectors. Code to reproduce this simulation is in our GitHub repository. The original SDM variance equation does not model the size of the circle intersection with the target pattern in its denominator and as a result the standard deviation is a constant horizontal line. This makes it a far looser lower bound on the standard deviation and much higher, more optimistic upper bound on the SNR.

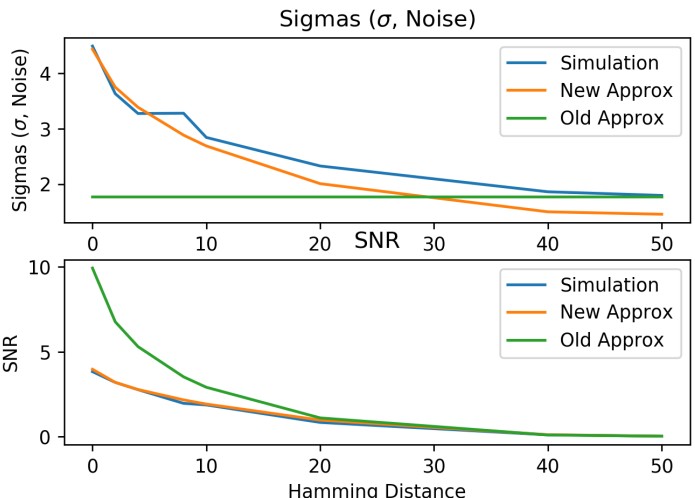

Figure 16: Simulating the real variance and signal to noise ratio of SDM to show the quality of the "New" approximation of [27] which has close agreement across Hamming distances between a query and target pattern. "Old Approx" is the variance equation presented in the SDM book [13]. (**Top Row**) The standard deviation values. (**Bottom Row**) the Signal to Noise ratio.

## B.5  Optimal Hamming Distance

For a given instantiation of SDM, the optimal Hamming distance threshold $d^*$ for reading and writing depends upon the objective of the memory system, the parameters $n$, $r$, $m$, and the distribution of patterns throughout the space. Examples of memory system objectives that will be discussed are maximizing the retrieval probability of noise-free queries, maximizing the number of patterns that can be stored in memory above a specified retrieval error, and maximizing the amount of noise that a query can tolerate while still retrieving its target pattern. For all of the analysis that follows, we assume that patterns and in turn neurons are random. This is a strong assumption that deviates from correlated real world data and the learned representations used by Transformer Attention. However, this assumption makes analysis of SDM tractable and still serves as both a useful reference point and way to think about the trade-offs associated with different SDM instantiations.

For the analysis that follows, when considering the optimal $d^*$ value it can be useful to refer to the fraction of the vector space $p$ that is within the $d$ threshold because $p$ is independent of $n$. Before discussing specific optimal values of $d^*$ that depend upon the aforementioned parameters, we discuss higher level constraints on $d$. Biologically plausible versions of SDM only consider the interval $p \in [10^{-11}, 10^{-1}]$ because they assume there are a finite number of neurons $r < 10^{11}$ and want each pattern to be within $d$ of at least one neuron in expectation.[14] This assumption that $r \ll 2^n$ possible vector addresses in the binary space is what makes SDM "sparse". For example, if $r = 1,000,000$ then we want $p > 10^{-6}$ such that there is at least one neuron present that any random pattern can read and write to. Note this assumes the cerebellum is running one instance of SDM rather than being multi-headed, which would reduce the number of neurons for each implementation. This multi-headed nature may be more likely given the discovery of cerebellar microzones and closed loop circuits with different neocortical regions making, $r = 10^{11}/i$ where $i$ is the number of SDM instances [81, 82].

Intuitively, this value of $r$ matters because having more neurons to store patterns will increase memory capacity. Specifically, having more neurons provides greater specificity in how large a given circle intersection is and thus how much each pattern is weighted in the SDM read operation. Making this intuition more formal, the circle intersection equation computes the number of all $2^n$ possible binary vector addresses that exist in the circle intersection. Dividing by $2^n$ gives us the fraction of space this circle intersection occupies. If we then multiply by $r$ we get the expected number of neurons that exist in this space fraction. Because the SDM read operation is normalized, the multiplication by $r$ should not matter as it will cancel in the numerator and denominator. However, when we are considering real implementations of SDM, where the number of expected neurons cannot be an expectation to many decimal points of precision but instead must be an integer number of neurons, large $r$ increases the decimal point accuracy of the circle intersections by turning these decimals into larger, more precise integers. This greater precision can be crucial for correctly weighting each of the pattern pointers in the update rule (see Appendix B.7 examples where enforcing the integer number of neurons causes convergence to collapse). Because Attention works with floating point numbers, it's mapping to the space of discrete binary vectors assumes that $r = 2^n$, while $n$ is relatively small here at $n = 64$, $r = 1.8 * 10^{19}$ which is many orders of magnitude greater than the number of granule cells in the cerebellum, making it biologically implausible. As will be shown, Attention having such a large $r$ has consequences for some but not all of the optimal $d^*$ values found.

Beyond the biological plausibility of $r$ and its effect on $d$ via $p$, at a high level we also want $d < n/2$, equivalent to $p < 0.5$, such that it is not reading from or writing to locations that are orthogonal. Additionally, as $m$ increases there are more random patterns to increase noise such that $d$ should get smaller to keep its SNR high enough for convergence to its target pattern.

We now turn to the definitions and derivations of optimal $d^*$. By assuming patterns are random, we can use the Signal to Noise Ratio (SNR) Eq. 25 and, by further assuming our query is noise free, we can take its derivative and find the $d$ value that maximizes the probability this noise-free query will converge to its target pattern [15]. In this setting our query is already equal to our target pattern and we want to maximize the probability that upon a query update it remains there such that it is a fixed point. This objective function is easy to derive and would be useful for noise-free settings where we want our memory system to be of maximum reliability.

---

[14]When discussing biological plausibility, the SDM book notes that there are an estimated $r = 10^{11}$ granule cells, which was one order of magnitude too large [79, 19]. 80% of the $\sim$90 billion neurons in the human brain are hypothesized to be the $r$ neurons in SDM's map to biology [80].

The SNR approximation in Eq. 25 of Appendix B.4 reproduced here is:

$$\text{SNR}(d_v^*, d_v^{\text{ortho}}, d, r, m, n) \approx \frac{\mathbb{E}[I_*]}{\sqrt{\mathbb{E}[I_*] + (m-1)\big(\mathbb{E}[I_{\mu \neq *}] + \mathbb{E}[I_{\mu \neq *}]^2\big)}},$$

where to make taking the derivative tractable we assume the Hamming distance between our query and target pattern, $d_v^* = 0$, while our orthogonal pattern is a distance of $d_v^{\text{ortho}} = n/2$ such that $\mathbb{E}[I_{\mu \neq *}] = p^2 r$ and $\mathbb{E}[I_*] = pr$. This is because, in expectation, two vectors that are equal will have a circle intersection taking up $p$ of the space while two orthogonal vectors will have an intersection by chance of $p^2$. Thus we can write:

$$\text{SNR}(d_v^*, d_v^{\text{ortho}}, d, r, m, n) \approx \frac{pr}{\sqrt{pr + (m-1)\big(p^2 r + (p^2 r)^2\big)}} = \frac{pr}{\sqrt{pr\big(1 + p(m-1)(1 + p^2 r)\big)}},$$

and for large $m$ we can let $m \approx m - 1$. We can then find the optimal $p$ that maximizes the SNR by taking the first derivative with respect to $p$ and setting it to equal 0. This gives us the optimal $p^*$ solution:

$$p^* = \frac{1}{\sqrt[3]{2mr}}.$$

We can then use the Binomial (or Gaussian for large $n$) inverse CDF with the dimensionality of our space $n$ to get our value of $d^*$ that corresponds to the Hamming distance capturing $p^*$ of the space.

$$d^* = \mathcal{B}_{\text{CDF}}^{-1}\left(p^*, n, \text{prob} = 1/2\right), \tag{28}$$

where $\mathcal{B}_{\text{CDF}}$ is the inverse CDF of the Binomial distribution.

In the canonical, biologically plausible setting, where we assume that $n = 1,000$, $m = 10,000$ and $r = 1,000,000$, we get $p = 3.7 * 10^{-4}$ and $d^* = 447$. Meanwhile, in the biologically implausible Attention setting where $n = 64$, $m = 1024$, and $r = 2^n$ we get $p = 2.98 * 10^{-8}$ and $d^* = 11$. We put the results for our canonical SDM and Attention parameters with the other optimal $d^*$ variants that will be discussed next in Table 1.

Turning to the next optimal $d^*$, for maximizing memory capacity, we perform the same analysis and assume that our query is noiseless to derive an equation that gives us the maximum $m$ for a given level of error tolerance. Reproducing Eq. 26 of Appendix B.4 for convenience:

$$m = \frac{\big(\frac{\mathbb{E}[I_*]^2}{z^2} - \mathbb{E}[I_*]\big)}{\mathbb{E}[I_{\mu \neq *}] + \mathbb{E}[I_{\mu \neq *}]^2} + 1$$

where $z$ is the Z-score that corresponds to the probability of successfully retrieving the target pattern at a single bit. Setting $\mathbb{E}[I_{\mu \neq *}] = p^2 r$ and $\mathbb{E}[I_*] = pr$, taking a derivative with respect to $p$, and solving for where the equation equals 0 gives us:

$$p^* = \frac{1}{2}\left(\frac{z^4}{\sqrt[3]{2r^4 z^2 + r^3 z^6 + 2\sqrt{r^8 z^4 + r^7 z^8}}} + \frac{\sqrt[3]{2r^4 z^2 + r^3 z^6 + 2\sqrt{r^8 z^4 + r^7 z^8}}}{r^2} + \frac{z^2}{r}\right)$$

and we can use Eq. 28 to retrieve the optimal value of $d^*$. We assume that the total retrieval probability should be 99% and use Eq. 27 to work out the corresponding z-score per bit with the results presented in Table 1.

Another optimal definition for $d^*$ is to maximize the "critical distance": the amount of noise that a query can tolerate, while still retrieving its target pattern. To compute the critical distance, we must use the SNR Eq. 25 to compute fidelity $f$, the probability that the $i$th bit of the query is updated to match that of the target pattern it is closest to. Fidelity is computed using the Gaussian standard normal CDF $F(\cdot)$:

$$f := P(\boldsymbol{\xi}_i = [\mathbf{p}_a]_i^* | d_v^*, d_v^{\text{ortho}}, d, r, m, n) = F\big(\text{SNR}(d_v^*, d_v^{\text{ortho}}, d, r, m, n)\big)$$

In expectation, we will decode correctly $f * n$ bits meaning our updated query will have a new Hamming distance of $n - (f * n)$. We can find the critical distance of $d_v = d(\mathbf{p}_a^*, \boldsymbol{\xi})$ where our update diverges such that the updated query is further away from the target pattern than before,

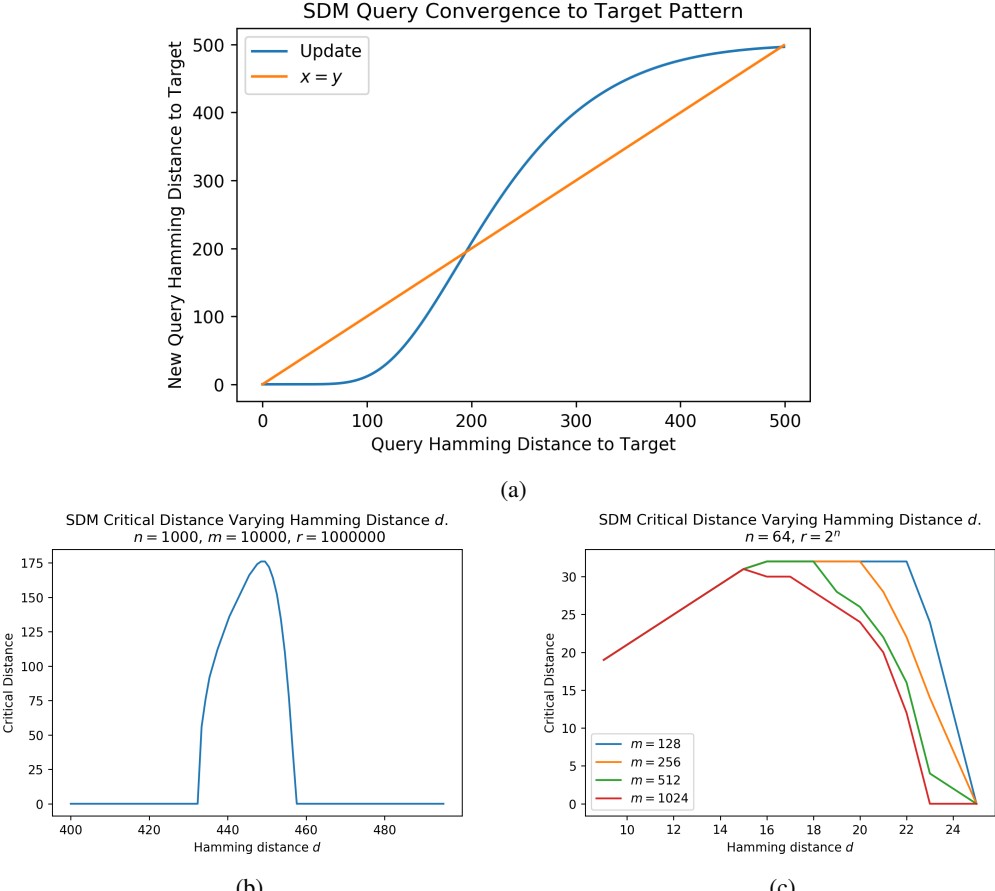

(a)

(b)                                    (c)

Figure 17: **(a)** Plot showing how the current (x-axis) vs updated (y-axis) query hamming distances to a target pattern. Where the blue line crosses the orange is the critical distance, which is $d_v = 188$. We assume $n = 1,000$ dimensions, $r = 1,000,000$ neurons, $m = 10,000$ patterns, and a $d = 451$ distance threshold. This is a reproduction of Figure 7.3 from the SDM book [13] but uses the slightly more accurate standard deviation calculation outlined in Appendix B.4. **(b)** By generating plots like in (a) and recording the critical distance, doing so for many $d$ values we can find the optimal critical distance $d^*_{CD}$. This plot shows the critical distance as a function of $d$ using the canonical SDM settings $n = 1,000$, $m = 10,000$ and $r = 1,000,000$, giving $d^* = 448$. **(c)** The same as (b) but for Attention parameters $n = 64$, $r = 2^n$, and varying $m$ corresponding to different numbers of pattern inputs. When $m = 1024$ (the maximum for GPT2 Transformer inputs), $d^* = 15$, otherwise when $m \leq 512$, $16 \leq d^* \leq 22$ as depicted.

$d(\mathbf{p}^*_a, \boldsymbol{\xi}^{\text{new}}) > d(\mathbf{p}^*_a, \boldsymbol{\xi})$. For a given instance of SDM we can generate a convergence plot like in Fig. 17(a) where the location of the blue line crossing the orange is the critical distance. We can find the critical distance for different instances of SDM by varying $d$ in order to find the $d^*$ that maximizes it for example in Fig. 17(b) and (c).

Comparing between the different optimal $d^*$ values for each of the three methods in Table 1, we see that for the canonical SDM setting all of the optimal values are almost the same. This is because of both the much smaller $r$ value and also the large $n$ that makes changes in $p$ have less of an impact on $d$. For larger $n$, more vectors are at the orthogonal distance of $n/2$ so $d$ in turn must be closer to $n/2$ in order to capture the same fraction of the space. For example, the SNR and Crit. Dist. Attention $p$ values are the same order of magnitude like all of the canonical SDM values but change far more than the corresponding $d$ values.

Table 1: Optimal $d^*$ values for different SDM instantiations and optimality definitions

| SDM Parameters | | | | Optimal Values $d^*$ ($p^*$) | | |
| Name | $n$ | $r$ | $m$ | SNR | Memory | Crit. Dist. |
| --- | --- | --- | --- | --- | --- | --- |
| Canonical | 1,000 | 1,000,000 | 10,000 | 447 ($3.7 * 10^{-4}$) | 444 ($2.18 * 10^{-4}$) | 448 ($5.58 * 10^{-4}$) |
| Attention | 64 | $2^n$ | 1,024 | 11 ($2.98 * 10^{-8}$) | 5 ($2.67 * 10^{-13}$) | 15 ($1.22 * 10^{-8}$) |

For the Attention row, we assumed that $m = 1024$, however, this is its upper bound and smaller $m$ values for both SNR and critical distance $d^*$ variants will result in larger $p$ and $d$ values. This $m \leq 1024$ for two reasons, first this is the maximum size of the receptive field, if there are fewer inputs available then it will be smaller. Second, as discussed in the main text, because Attention learns to represent patterns such that they are far from being randomly distributed, and the majority are further than orthogonal away as shown in Fig. 6. Combining these two reasons we should expect that $m$ is always quite a bit less than 1024 patterns. If this is true and optimal $d^*_{\text{CD}} \geq 15$ this corresponds to smaller learnt $\beta$ values and could better account for the smaller inferred $\beta$s in the GPT2 Transformer Fig. 5.

It is crucial to note that these $d^*$ variants all assume random patterns and can only be considered as reference points for considering what $d$ and its approximating $\beta$ Attention should use in the Transformer. When considering which of these references is the most useful, both SNR and Memory assume that the query is noise free such that it is equal to the target pattern. The Transformer's use of Dropout alone, in conjuction with the desire for it to generalize beyond its training data, invalidates this noise-free assumption [83, 1]. Therefore, it is likely that Attention will want to approximate versions of SDM that optimize more for critical distance, thus using $d$ values closer to $d^*_{\text{CD}} = 15$.

The Modern Continuous Hopfield Network paper has a number of formal analyses for how $\beta$ (related to SDM's $d$) influences convergence in Transformer Attention for different memory spaces in the continuous setting that we point readers to [25].

With regards to concerns of SDM's random pattern assumption limiting its real world applicability, [29] has shown that the theoretical results of SDM hold with correlated patterns if neurons exist in high density regions of the space and the Hamming distance $d$ dynamically changes depending on the density of addresses around the query to activate a constant number of neurons.

## B.6 SDM is a Generalization of Hopfield Networks

Hopfield Networks [84] are a special case of SDM where there are no distributed reading or writing operations [29]. At a high level, this removes the need for the existence of neurons in the vector space and makes the weighting of patterns in the query update equation a re-scaled version of the Hamming distance without any threshold.

Hopfield Networks can be viewed as the neuron based implementation of SDM,[15] where there is no distributed writing so $d_{\text{write}} = 0$. For reading, the query reads every pattern using a bipolar re-scaled version of Hamming distance and doesn't apply any threshold [29]. Neurons are made to be patterns so $r = m$ and $X_a = P_a$ (where $X_a$ is a matrix with its columns corresponding to neuron address vectors). Moreover, because the Hopfield Network is frequently autoassociative, $P_p = P_a = X_a = X_v$ [84, 29].

Traditionally, the Hopfield Network deals with vectors $\mathbf{x} = \{-1, 1\}^n$. We can make SDM also exist in this space by mapping between Hamming distance of binary vectors and bipolar vectors with: $d(\mathbf{x}, \mathbf{y}) = \frac{n - \mathbf{x}^T \mathbf{y}}{2}$. We can also ignore our normalizing constant in the denominator because to compute the majority value we only need to check if the sum is positive or negative. Therefore, we

---

[15]The neuron based implementation of SDM is outlined here but reviewed in more depth in [13, 15] and isomorphic to the pattern based perspective used in the main text, when the number of neurons used are the same.

have the SDM reading operation that uses neurons [13]:

$$\boldsymbol{\xi}^{new} = g(P_p \underbrace{b(d_{\text{write}}, \frac{n - X_a^T P_a}{2})}_{\text{Write Patterns}} \underbrace{b(d_{\text{read}}, \frac{n - X_a^T \boldsymbol{\xi}}{2})}_{\text{Read Patterns}}))$$

$$g(\mathbf{e}) = \begin{cases} 1, & \text{if } e_i > 0 \\ -1, & \text{else} \end{cases} \qquad b(d, \mathbf{e}) = \begin{cases} 1, & \text{if } e_i \leq d \\ 0, & \text{else} \end{cases}$$

And the original Hopfield Network update rule in matrix form is:

$$\boldsymbol{\xi}^{new} = g(P_a P_a^T \boldsymbol{\xi}) \tag{29}$$

If we define $d_{\text{write}} = 0$ then patterns only write to the neurons centered at them:

$$P_p b(d_{\text{write}}, \frac{n - X_a^T P_a}{2}) = P_p I = P_p = P_a$$

where $I$ is the identity matrix. For the reading operation, by removing the binary threshold the query reads from every neuron. Also not shifting and re-scaling the dot product between bipolar vectors to be the Hamming distance results in:

$$b(d_{\text{read}}, \frac{n - X_a^T \boldsymbol{\xi}}{2}) = X_a^T \boldsymbol{\xi} = P_a^T \boldsymbol{\xi}.$$

These modifications turn SDM into the Hopfield Network in Eq. 29.

We note that, while Hopfield Networks have been traditionally presented and used in an autoassociative fashion, as long as they use a synchronous update rule they can also model heteroassociative sequence transitions with the equation $\boldsymbol{\xi}^{new} = g(P_p P_a^T \boldsymbol{\xi})$ [29]. Meanwhile, modern Hopfield Networks can handle heteroassociative tasks with a simple modification outlined in [26]. This means the heteroassociative abilities of SDM are not unique. However, traditional Hopfield Networks are biologically implausible because of the weight transport problem where the weight matrix $P_a P_a^T$ is symmetric. Meanwhile, modern Hopfield Networks lack the degree of biological detail provided by SDM. This is because SDM not only identifies the unique connectomics of the cerebellum as satisfying many of its constraints [13] but also because of the work presented here highlighting how the exponential function used in [85, 25, 26] can be implemented through simple properties of neuron activations.

Conceptually, we can interpret the Hopfield update within the SDM framework as computing the similarity between $P_a^T \boldsymbol{\xi}$ which, because of the bipolar values, has a maximum of $n$, minimum of $-n$ and moves in increments of 2 (flipping from a +1 to -1 or vice versa). This distance metric between each pattern and the query being used as a weighting for the linear combination of patterns results in our query being attracted to patterns with high similarity and repulsed away from those dissimilar. However, all patterns are considered while in SDM, patterns are weighted in proportion to the intersection of the write circle of a pattern with the read circle of the query so any patterns that are $d(\boldsymbol{\xi}, \mathbf{p_a}) > 2d$, will have no circle intersection and receive no weighting. This gives the query a hard threshold of $2d$ for patterns it assigns any weight to. An emergent property of using this circle intersection for weighting is that it is approximately exponential.

It has also been noted that small, biologically plausible modifications to SDM can allow it to maintain its original capacity while handling correlated input patterns that the Hopfield Network cannot [29]. This is because SDM can learn through competitive dynamics to have neurons that are distributed through the space to match the pattern distribution and can dynamically update the Hamming distance $d$ so that it can still converge to the right pattern [29]. This is crucial as most, if not all, real world applications of associative memory will deal with correlated patterns. However, Hopfield Networks have advantages over SDM in their simplicity and energy landscapes that can be used for convergence proofs and to solve optimization problems [86, 29]. Finally, both Hopfield Networks and SDM, while using different parameters, have been shown to have the same information content efficiency as a function of their parameter count [29].

## B.7 SDM Experiments

In this section we implement eight algorithms that are all possible variations of the SDM and Attention approximations discussed in the paper and evaluate their empirical convergence abilities in

the autoassociative setting. These experiments are both to validate the equations presented in this work and test the theoretical memory and convergence capabilities of SDM. Most importantly, these experiments show that the Attention approximation to SDM holds well.

The eight algorithms vary in using continuous or binary vectors and using the SDM circle intersections (binary or continuous) or Attention approximations that fit their $\beta$ coefficient to the SDM circle intersection. These algorithms are:

- Binary SDM - From Eq. 1. Does not enforce a finite number of neurons in the space.
- Binary SDM Limited Neurons - Same as the above but enforces that there are an integer $r$ number of neurons in the space. For example, if a circle intersection has an expected 3.6 neurons in its intersection, we map this to an integer by randomly sampling if the number of neurons is 3 or 4 with a 60% probability there are 4.
- Binary Neuron SDM - The original SDM formulation that is the most biologically plausible, where neurons are defined at specific locations in the space and store a superposition of patterns in their value vector [13].
- Binary SDM Binary Fit Attention - Using binary vectors but weighting them with the Attention softmax that has fit $\beta$ to the binary circle intersection.
- Continuous Binary SDM - Using continuous vectors but mapping their cosine similarities into Hamming distances using Eq. 5 to use the binary circle intersection for weighting.
- Continuous SDM - Using continuous vectors and the continuous SDM intersection of hypersphere caps from Eq. 22 outlined in Appendix B.3.
- Continuous SDM Binary Fit Attention - Using continuous vectors that are $L^2$ normalized with Attention softmax. $\beta$ is fitted to the binary SDM circle intersection.
- Continuous SDM Continuous Fit Attention - Same as the above algorithm but $\beta$ is to the continuous circle intersection version of SDM.

We test convergence for three datasets:

- Random patterns - Sampling from a Uniform(-1,1) distribution vectors of dimension $n = 64$ and $n = 1,000$. The theoretical results of SDM hold here.
- MNIST - Correlated dataset of handwritten digits from 0 to 9 in grayscale [87]. $n = 784$ dimensions representing flattened 28 by 28 images.
- CIFAR10 - Correlated dataset of ten different object categories including planes, cats and horses [88]. Converted to grayscale with $n = 1024$ representing flattened 32 by 32 dimensional images.

SDM's original formulation assumes that neurons are fixed in place and randomly distributed, which is not a problem for random patterns that make convergence and capacity analysis tractable. However, randomly distributed neurons work poorly for real world correlated datasets like MNIST and CIFAR10 that exist on a lower dimensional manifold. This is problematic as there will only be a few that are initialized on the data manifold by chance, reducing the number of neurons in circle intersections and their precision. Additionally, the patterns themselves are often very close in space, introducing noise. For example, if our target pattern we are trying to converge to is a particular handwritten 5 digit, there are many thousands of similar looking 5s in the MNIST dataset we may converge to instead.

We first test how SDM and its Attention approximation handle random patterns, showing these results agree with theory. We then test these algorithms on MNIST and CIFAR10 where their performance is significantly reduced when implemented directly on the raw pixels of the images because of high correlations in the dataset. In response, we learn a projection matrix like that used in Transformer Attention [1] to map the images to a lower dimensional $n = 64$ latent space that significantly increases the separability between patterns and memory capacity of both SDM and Attention. We leave tests where the neurons update their locations to cover the pattern manifold to future work. However, we do test how performance changes as a function of the number of neurons $r$ in the space. Increasing the number of neurons and learning a project matrix can not only create a lower dimensional latent space (increasing neuron density) but also expands the size of the data manifold (utilizing more of the uniformly distributed neurons) and is useful for increasing SDM's performance with real world data.

### B.7.1    Random Patterns

For these experiments, we randomly generate $m$ patterns then randomly select one of them as the target pattern $\mathbf{p}^*$, where our query is centered. We apply random perturbations of increasing magnitude to the query and see if it is able to converge back to its target pattern after a maximum of 100 iterations (this was more than sufficient for all convergences analyzed). We simulate 3 different random versions of the dataset and for each we apply 5 different random perturbations to all $m$ data points, testing every algorithm on the exact same perturbations. Because we use $m = 1,024$ this means we test a total of $3 * 5 * 1024 = 15,360$ convergences with each perturbation magnitude, for every algorithm.

For our binary algorithms we make our patterns binary by setting values to 0 or 1 if they are positive or negative, respectively. We confirm that this discretization of the vectors respects the mapping defined by Eq. 5 in that all pairwise distances between vectors in the different spaces remain approximately equal. In the continuous setting (used by the latter four algorithms that have "Continuous" as the first word in their names), we apply a cosine similarity perturbation that is equivalent to the Hamming distance perturbation. We aggregate the results and plot their mean and standard deviation as a function of perturbation magnitude. For these experiments, we test two different sets of parameters, one following the Attention setup where $n = 64$ and $r = 2^n$ [1]; the other the canonical SDM setup where $n = 1,000$ and $r = 1,000,000$ [13]. In both cases, $m = 1,024$ is used because it is the maximum number of patterns that can be input at one time into the GPT2 variant of the Transformer [3].

We use the same parameters for all experiments except for "Binary Neuron SDM", where we drop one order of magnitude in the number of neurons to $r = 100,000$ and use the lower dimensional Attention $n = 64$. This is because it is very expensive to compute Hamming distances between all neurons, patterns, and queries without highly parallelized and high memory computational resources. We compare the neuron based SDM implementation to the "Binary SDM Limited Neurons" algorithm it is isomorphic to in expectation as a sanity check.

**Random Patterns - Attention Settings**

Our theoretical convergence results for SDM that assumed random patterns are replicated well in both parameter settings. Figure 18 shows every algorithm aside from "Binary Neuron SDM", which needed fewer neurons to efficiently run and is shown separately. Each plot shows the convergence performance of a different algorithm with six different Hamming radii for the read and write operations that spread a wide range of possible values including those that are deemed optimal: $d \in \{5, 9, 11, 15, 19, 27\}$ corresponding to the fractions of the space $p \in \{1e-13, 1e-9, 1e-8, 7e-6, 3.68e-4, 0.1\}$. For the algorithms using binary vectors, we compute the Hamming distance of the converged query to its target and convert this to cosine similarity so that all plots have the same scale and y-axis. We also plot baseline values which are the cosine similarity between the target pattern and the perturbed query before it undergoes convergence. Results that are above the baseline value show convergence while results that are below show divergence. We perform perturbations up to a value of 12, any higher and perturbations often make a query closer to a different pattern than the target pattern, making successful convergence impossible.

There are a few take-aways from these plots. First, and most importantly, there is close agreement between all of the algorithms. Aside from one exception, only when using the smallest $d = 5$ at the largest perturbation is there disagreement between the algorithms and this can be explained by SDM theory. Also note that $d = 5$ is the optimal $d^*$ for maximum memory capacity and assumes no query noise, which is the opposite of what we are optimizing for here. This agreement is particularly strong for the third and fourth rows that show "Continuous SDM", "Continuous SDM Binary Fit Attention", and "Continuous SDM Continuous Fit Attention" where the first uses the continuous circle intersection and the latter two use softmax approximations to the binary and continuous intersections, respectively. This shows empirically that across almost all Hamming distances, the approximation between the SDM circle intersection and softmax equation with a learned $\beta$ is tight.

The second take-away is that SDM theory can explain these results. In Fig. 19, we show the theoretical maximum critical distances for each Hamming distance as a function of the number of neurons used. In this case $r = 2^n = 2e + 19$ is on the far right of the plot. $d = 27$ has a critical distance of zero meaning it never converges as is shown for every algorithm. $d = 5$ is the next worst, failing around a perturbation $\approx 11$ and we can see that all the algorithms using the circle intersection

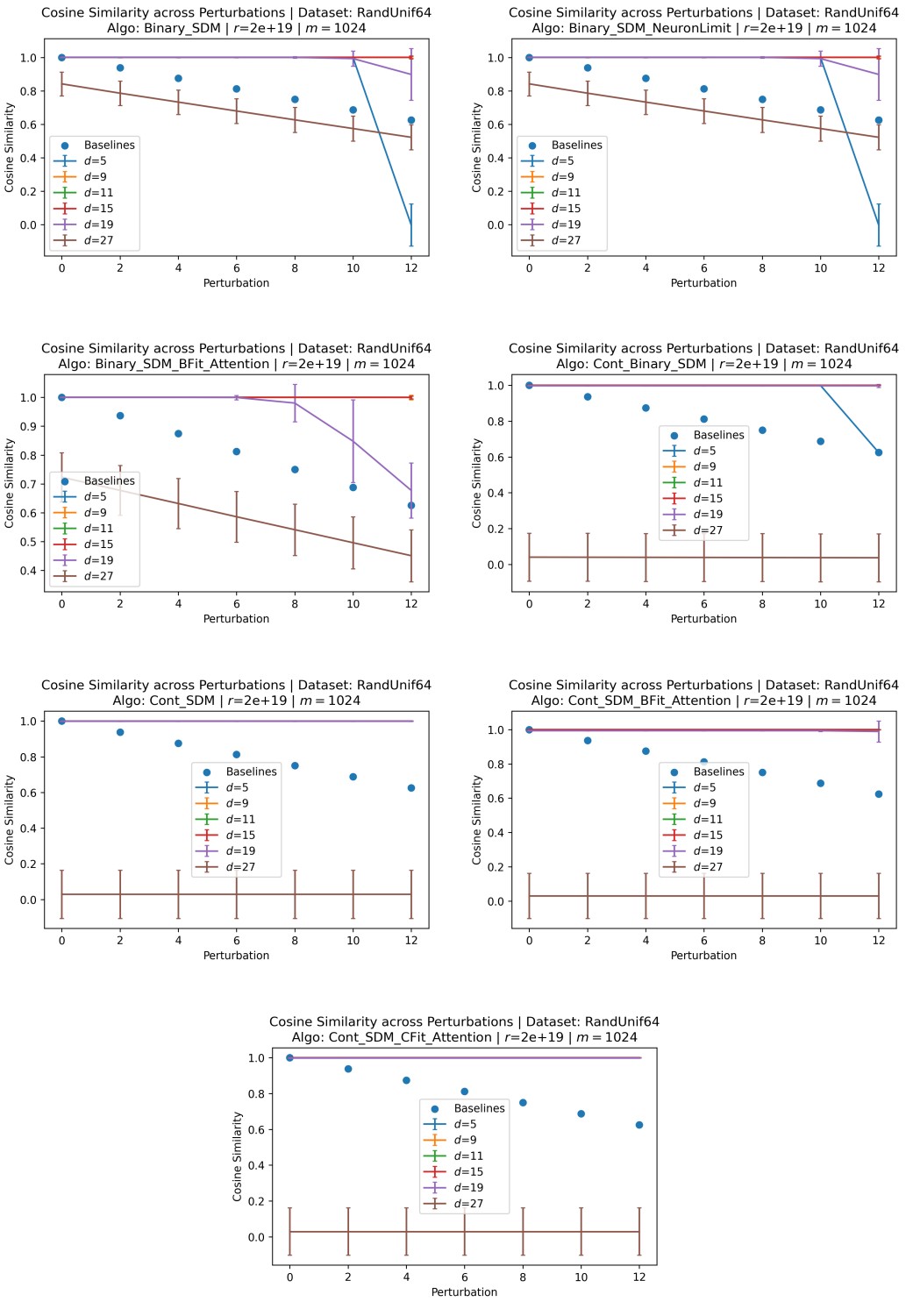

Figure 18: All of the SDM and Attention approximations closely agree using Random Patterns and the Attention parameters: $n = 64$, $r = 2^n$ and $m = 1024$. These results also largely agree with the critical distance convergence theory in Fig. 19.

in binary space (both plots in the top row and the second row on the right) show this drop off. A violation of the theory is the drop in performance of $d = 19$ at a perturbation of 12, however, the

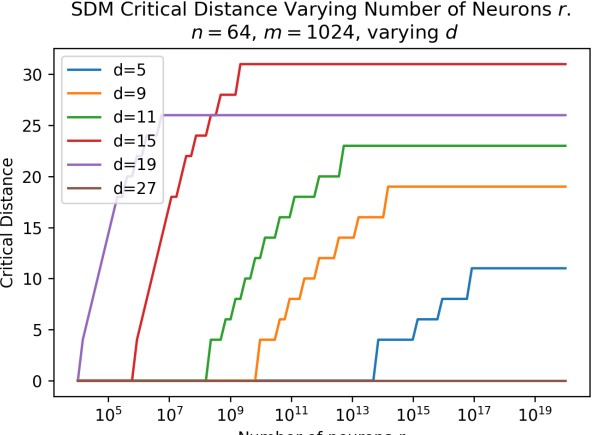

Figure 19: Showing how the critical distance increases with the number of neurons in the space for Hamming read and write radii, $d \in \{5, 9, 11, 15, 19, 27\}$. These maximum critical distances hold in the experimental results of Figs. 18 and 18.

error bards indicate this is only statistically significant in the case of the second row left plot "Binary SDM Binary Fit Attention". Here, performance remains above the baseline but why it falls is unclear, we hypothesize it may have to do with the use of the softmax pattern weightings when combined with binary vectors. Note that in the Binary setting on the top row $d = 5$ falls to a cosine similarity of 0.0 at the largest perturbation while in the second row right plot that uses the same circle intersection but with continuous vectors it only falls to the baseline value. This is entirely explained by our implementation of the binary versus continuous algorithms. In the continuous setting when there is a lack of any circle intersections as is the case here, to avoid NaN values we reset our updated query to its version before it was updated, in this case setting the query back to its perturbed state. This is instead of becoming an all 0s vector which is what happens in the binary space due to the lack of any circle intersections (weighting each pattern by a weight of 0). Because our binary vectors are generated to be half 0s and half 1s in expectation, our all 0s vector corresponds to an orthogonal cosine similarity of 0 as shown. Note that $d = 5$ being too small is not an issue when using the continuous SDM circle intersection equation as shown by the fact it does not fall in performance. This is because the curvature of the $L^2$ normalized n-dimensional hypersphere enables intersections to continue existing for larger distances between vectors as is discussed in Appendix B.3.

As a third take-away, the "Binary SDM" and "Binary SDM Limited Neurons" (top row) perform identically, this is because, while we enforce a finite number of neurons in the latter algorithm, there are a large $r = 2^n$ number of neurons occupying every possible location in the binary vector space, which is more than sufficient. This is not the case for later experiments.

We plot the real SDM neuron implementation separately in Fig. 20 that uses only, $r = 100,000$ neurons to make the computation more tractable. The only difference between "Binary Neuron SDM" and "Binary SDM Limited Neurons" is that the former simulates real neurons at random, uniformly distributed locations, while the latter computes the circle intersection with an expected, finite number of neurons present. This difference should be largely removed in expectation over the many simulations run but "Binary Neuron SDM" will be higher variance, due to its random neuron initialization. We confirm that the expected result is what we get with both algorithms agreeing. This helps validate that all of our algorithms are correctly implemented in that they agree with the original and most biologically plausible SDM implementation [13]. These results also agree with the maximum critical distance theory of Fig. 18 where we are operating with $r = 10^6$ neurons such that $d = 19$ is the only distance with a non-zero critical distance. We still ran and plotted all of the other $d$ values to show that both algorithms fail in the same ways. For "Binary Neuron SDM" $d = 19$, at the largest perturbation value drops off slightly (this is partly obstructed by the legend) but we stated that we should expect higher variance from this algorithm and the standard deviation error bar shows this difference is not statistically significant.

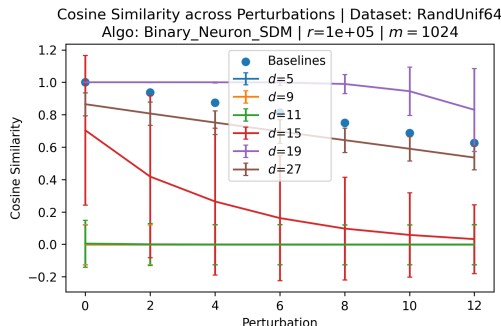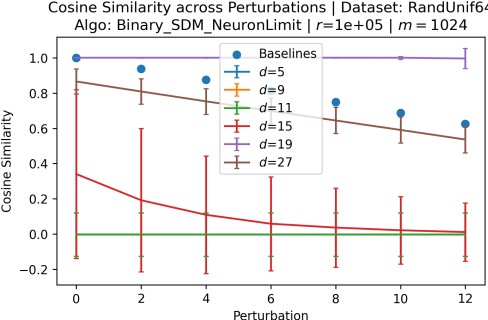

Figure 20: The biologically plausible "Binary Neuron SDM" algorithm **(Left)** agrees with "Binary SDM Neuron Limited" **(Right)**, which is isomorphic in expectation. This provides good validation that all the algorithms we consider are correctly implemented. "Binary Neuron SDM" is made tractable to compute by reducing the number of neurons to $r = 100,000$ and using the Attention parameters $n = 64$ and $m = 1024$. We use the same Hamming radii $d$ as in the previous figure 18.

### Random Patterns - Canonical SDM Settings

Here we show the same plots as above but using the canonical SDM setting of $n = 1,000$ and $r = 1,000,000$, while $m$ remains equal to $1,024$. Our perturbations sizes are again close to as large as they can be at a max value of 375, while still ensuring each perturbed query remains closest to its target pattern. We use the same fractions of the space as in the $n = 64$ setting $p \in \{1e-13, 1e-9, 1e-8, 7e-6, 3.68e-4, 0.1\}$ that corresponds to $d \in \{384, 405, 411, 431, 447, 480\}$. Note these values are a significantly wider range than the optimal $d^*$ values $d^* \in \{444, 447, 448\}$ provided in Table 1 but that these optimal $d^*$ are for $m = 10,000$ not $m = 1,024$ so are not directly comparable.

Figure 21 shows similarly good convergence between algorithms with a few notable discrepancies. First, in the top row, there is a difference in performance between "Binary SDM" and "Binary SDM Limited Neurons" that is explained by the small, finite number of neurons in the latter case where the only successful $d = 447$, as is exactly predicted by the theoretical critical distances of Fig. 22 where $r = 10^6$ and performance begins to drop around 300. This includes predicting that the critical distance of $d = 431$ is $\approx 100$ when we begin to see a performance decline.

Another discrepancy is where "Continuous SDM" and "Continuous Binary SDM" that use the circle intersection outperform all of the Attention fits for $d = 447$ at larger perturbations. The Attention fits match the theoretical critical distance prediction of Fig. 22 in declining at 300 but this does not hold in the actual SDM circle intersection computations. We cannot explain this difference currently aside from that we should expect for disagreements with our Attention approximation and its $\beta$ fit to appear at larger perturbations because this is when the circle intersection exponential approximation is less accurate.

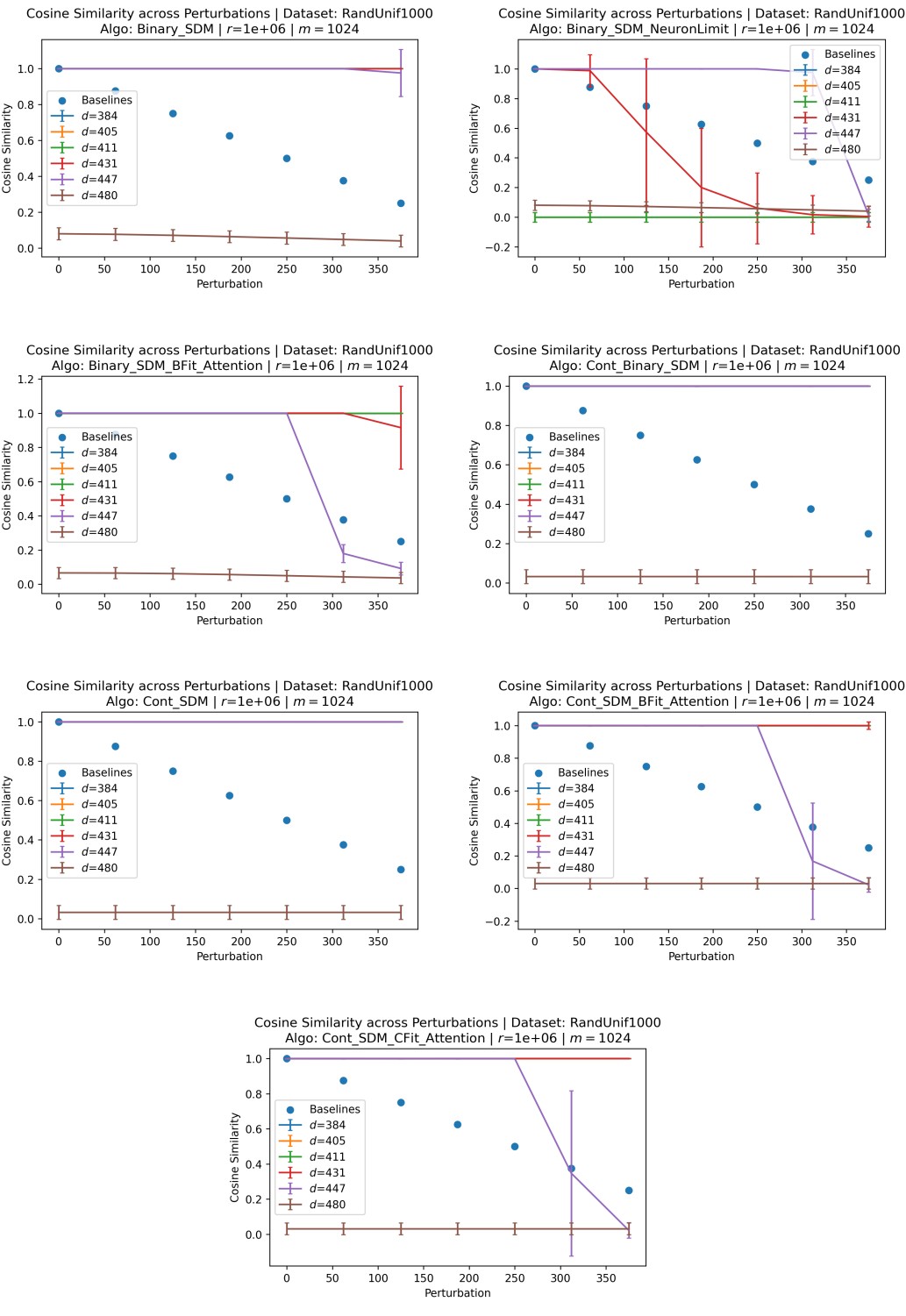

Figure 21: SDM algorithms largely converge in the canonical SDM setting with $n = 1,000$, $m = 1,024$, $r = 1,000,000$ across Hamming radii $d$. It is hard to see due to the overlapping lines aside from "Binary SDM Neuron Limited" every $d$ performs perfectly aside from where it is otherwise clear. For "Binary SDM Neuron Limited", only $d = 431$ and $d = 447$ show any convergence with the other algorithms all giving cosine similarities of $\approx 0$. This result where the number of neurons is finite agrees with the theoretical critical distances shown in Fig. 22 and with the definitions of optimal $d^*$ where only $d = 447$ works well across all algorithms when $r = 10^6$.

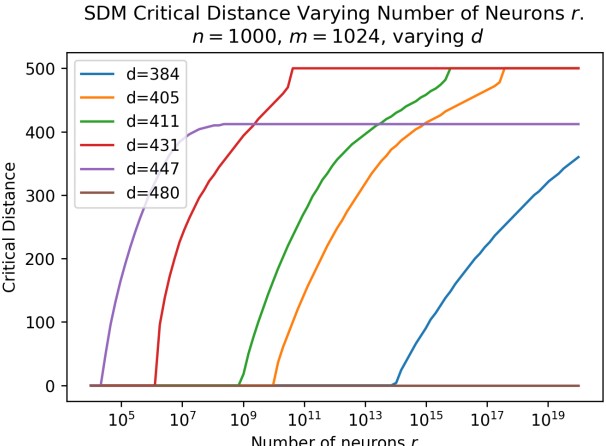

Figure 22: Theoretical results showing how the critical distance relates to the number of neurons in the space. In these experiments $r = 10^6$ making many of the $d$ values have critical distances of 0.

**MNIST**

SDM's theoretical convergence results were made tractable by assuming the patterns were randomly distributed throughout the space [13]. However, this is not the case with real world data and we test changes in convergence performance with the MNIST handwritten digits dataset [87].

Here, we do not allow our neurons to learn the MNIST data manifold, which would theoretically allow for performance to be the same as with the random pattern dataset [29]. Instead, we use random neurons to test to what extent the random pattern assumption holds with correlated data. For all of our experiments we set $n = 784$ as this is the dimensionality of flattened, 28x28 dimensional MNIST images. We tested making our MNIST digits binary using the simple rule of setting any non-zero pixel value to a value of 1. However, this does not respect the pairwise distance preservation between binary and continuous spaces making all pairwise distances on average closer and the problem harder in the binary setting. We thus only use the four continuous vector algorithms. These algorithms still have poor performance on the raw MNIST digits. We are able to significantly improve performance in the next section by learning projection matrices.

Note that for all of these results, when performance of the associative memory fails to converge it does not collapse to an orthogonal cosine similarity of 0, like with the random patterns. We reason that this is because of the correlated nature of the patterns where they may converge to a superposition of similar patterns. In Fig. 23, we perform the same analysis as with the random patterns where $m = 1,024$ and using the same $p$ fractions of the space and their corresponding Hamming radii. Only the smallest Hamming radius $d = 290$ is able to converge for smaller perturbation magnitudes which we believe is due to the patterns being so correlated in the space. Note that unlike with the Random Patterns dataset we no longer confirm that the perturbed query is still closest to its target pattern. This allows us to explore larger perturbation magnitudes than otherwise and how well convergence can happen to closely related patterns.

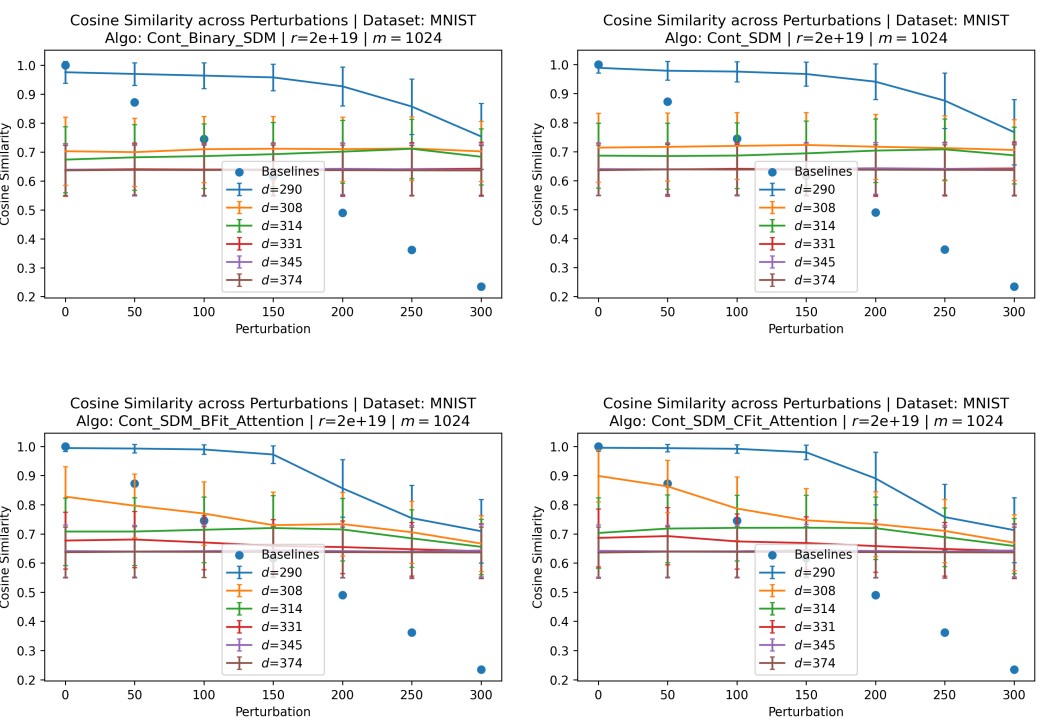

Figure 23: Using correlated MNIST digits significantly reduces SDM and Attention performance. All of the continuous algorithms used agree well showing that Attention's approximation to SDM still holds. Only the smallest Hamming radius of $d = 290$ is at all successful and we hypothesize this is due to the many correlated patterns that introduce noise when larger radii are used. Performance is dramatically improved when a projection matrix is learnt, especially for larger $d$ as is shown in the next sub-section.

### B.7.2 Learnt Projections for MNIST and CIFAR

In order to better assess the capabilities of SDM with correlated patterns, we learn a projection matrix to map image pixels into a latent $n = 64$ dimensional continuous vector space. This not only reduces the dimensionality of the space, making the randomly distributed neurons better fill the space but also can increase the separability of similar patterns by spreading out the data manifold, reducing noise during convergence. We also assess how performance changes as a function of many different numbers of neurons, $r \in \{3e + 09, 1e + 13, 3e + 16, 1e + 20\}$ and with an unconstrained, non-finite number.

For these experiments, we work with the continuous vector algorithms. We use Pytorch auto-differentiation to train via backpropagation a weight matrix with the loss function being the mean squared error between the target pattern and converged query. Training proceeds as follows: the projection weight matrix is random initialized. A random batch of 128 MNIST ($m = 60,000$) or CIFAR10 ($m = 50,000$) images from the training dataset are selected to be target patterns. These target patterns are converted into a query by randomly perturbing them to be somewhere between the cosine similarity equivalent of 10 and 100 Hamming distance away. These queries and the whole training dataset are projected into the $n = 64$ dimensional latent space and then $L^2$ normalized. Each query computes its cosine similarity with every pattern by taking a dot product and uses the "Continous SDM Binary Fit Attention" algorithm to weight each pattern. This applies Attention's softmax function with a learned $\beta$ value fit to the binary SDM circle intersection determined by the chosen Hamming radius. A weighted sum of the original pixel images (their pattern pointers in an auto-associative setting that are not $L^2$ normalized) is then performed to update the query and return it to the original image dimensionality. During training each query is given one update step to converge before its mean squared error with its target image is computed and its loss backpropagated to train the projection weight matrix. While the query is only updated once during training, during testing, if the query has not converged, it is projected again using the same weight matrix and can update itself up to 10 times.

Upon training the projection matrix for 10 epochs (iterations through the entire training dataset) we test its convergence on both the train and test datasets with all four continuous algorithms using the binary and continuous circle intersections and the softmax Attention approximations to each. We use the Limited Neuron versions of the circle intersections which allows us to test performance as a function of neuron number. The algorithms use are: "Continous Binary SDM Limited Neurons", "Continous SDM Limited Neurons", "Continous SDM Binary Fit Attention", and "Continous SDM Continuous Fit Attention". Training of the projection matrix is repeated 3 times for each Hamming radius used to ensure that its random initialization does not create anomalous performance. Repetitions occur for each Hamming radius as this determines the $\beta$ value used in the softmax during training and the size of the circle intersections. This in turn influences what projections should be learnt.

Note that we perform only limited hyperparameter tuning of our projection matrices, for example, of the learning rate, number of epochs and optimizer (we use Adam [89]). The purpose of these experiments is not to get optimal performance but show as a proof of concept that, analogous to Transformer Attention, a projection matrix can be learnt that significantly improves performance for correlated and complex real world datasets. Also note that during training we only perturb up to a Hamming distance of 100 while during testing we perturb up to 300, this likely explains why performance for each algorithm begins significantly declining after a perturbation distance of 100.

Figures 24 and 25 show example perturbations and convergences for each of the four continuous algorithms when applied to random images in the train and test distributions to give a sense for the size of the perturbations and corresponding differences in performance. These examples use $d = 5$ which has the highest variance in performance at larger perturbation amounts to illustrate what poor convergence looks like. We do not check that our perturbations keep our target pattern as the closest to the query so there are a number of cases where it is guaranteed that the query will not converge correctly, however, we still attempt to perform convergence and expect it to find closely related patterns.

Figures 26 and 27 show training performance for each algorithm and in every case, learning the projection matrix dramatically improves performance for the three smallest Hamming radii, $d \in \{5, 9, 11\}$. We only display these radii along with $d = 15$ as $d \in \{19, 27\}$ were both flat lines like $d = 15$ with slightly lower performance respectively and only introduce clutter to the plots. We hypothesize that these larger $d$ values are suboptimal for such large numbers of patterns $m$, especially

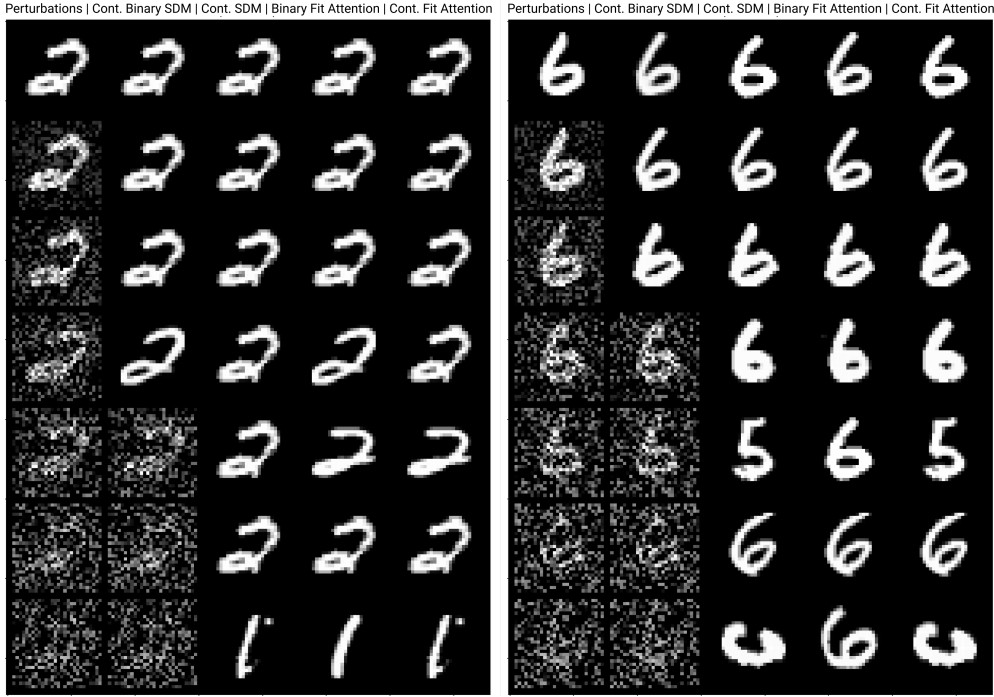

Figure 24: Perturbations applied to MNIST for two randomly chosen digits from the training distribution (**Left**) and test distribution (**Right**). The leftmost column is the perturbations of the digit that increase row-wise from Hamming distance: $\{0, 50, 100, 150, 200, 250, 300\}$ that are mapped to their equivalent cosine similarities (these represent the Baselines in Figs. 26 and 27). Each the proceeding columns are in order: "Continous Binary SDM Limited Neurons", "Continous SDM Limited Neurons", "Continous SDM Binary Fit Attention", and "Continous SDM Continuous Fit Attention". We are using the worst performing $d = 5$ Hamming radius that has the biggest difference in performance and assuming there are infinite neurons. Recall that the projection matrix was only trained on perturbations up to 100 (the third row) so any larger perturbations are out of distribution.

when dealing with correlated patterns and not learning neuron locations. Note that these plots show the non-finite neuron setting, making the same assumption of infinite neurons as Transformer Attention and also to assess their maximum performance abilities.

For both MNIST and CIFAR, there is also close convergence in performance between the algorithms. The only major difference in performance is when $d = 5$ for the "Continous Binary SDM Limited Neurons" algorithm. We believe this performance drop is related to the same performance drops seen in the Random Pattern cases (Fig. 18) where at larger distances $d = 5$ is too small for the magnitude of the perturbations, as was explained by the theoretical convergence result of Fig. 19. In this case, we also use much larger perturbations and do not check if the target remains the closest so the poor performance of $d = 5$ is more apparent. As noted in the Random pattern section, $d$ being too small is not an issue when using the continuous SDM circle intersection equation because the curvature of its $L^2$ normalized n-dimensional hypersphere space.

While the cosine similarity performance across the whole dataset for MNIST and CIFAR shown in Figs. 26 and 27 looks similar, it is clear from the randomly generated convergence examples in Figs. 24 and 25 along with other examples not shown that convegerence to the target is much worse for CIFAR. Failing to find the target image but still having high cosine similarity suggests that CIFAR datapoints are more correlated. CIFAR also overfits to its training data more than MNIST as shown in the test distribution image used on the right in Fig. 25. It makes sense that CIFAR test images are more out of distribution than those of MNIST, which are much lower complexity.

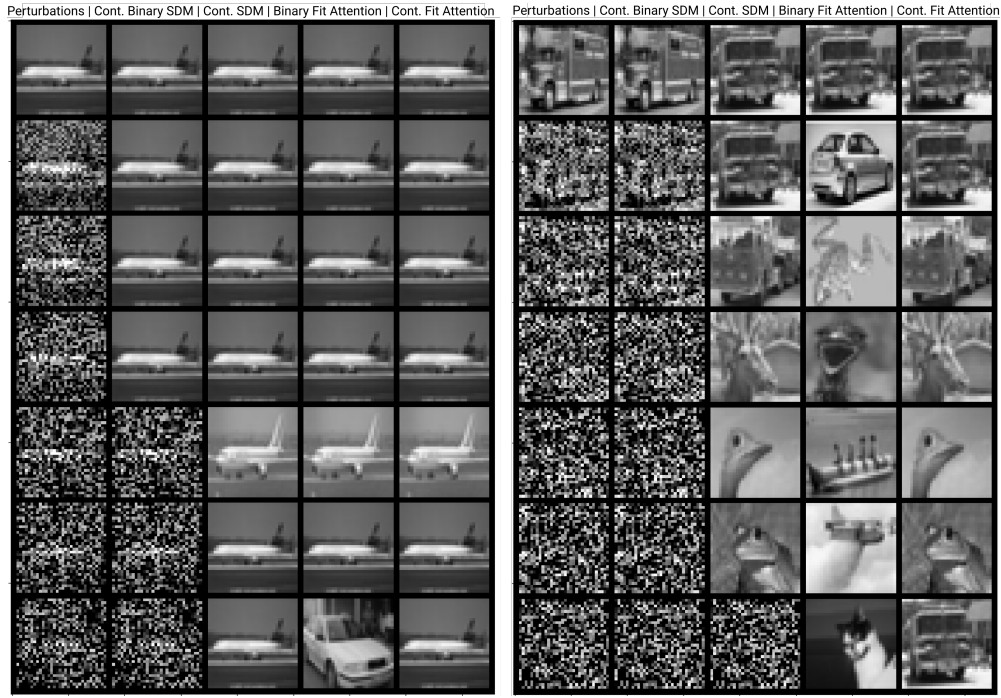

Figure 25: The same as the above Figure 24 but for the CIFAR dataset. We note that the test distribution (**Right**) performs much worse than the training distribution (**Left**).

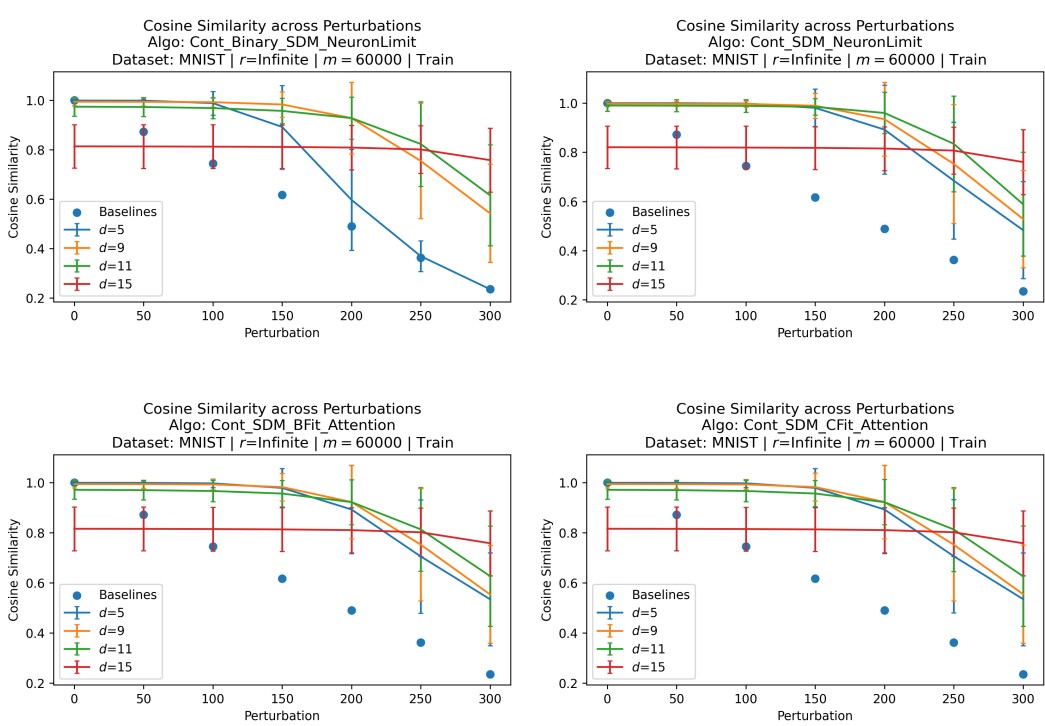

Figure 26: Training a projection matrix significantly improves MNIST critical distances for a number of the smaller Hamming radii. There is also close agreement between algorithms aside from $d = 5$, which is explained by the binary SDM circle intersection ceasing to exist at larger perturbation magnitudes for this particularly small Hamming radius.

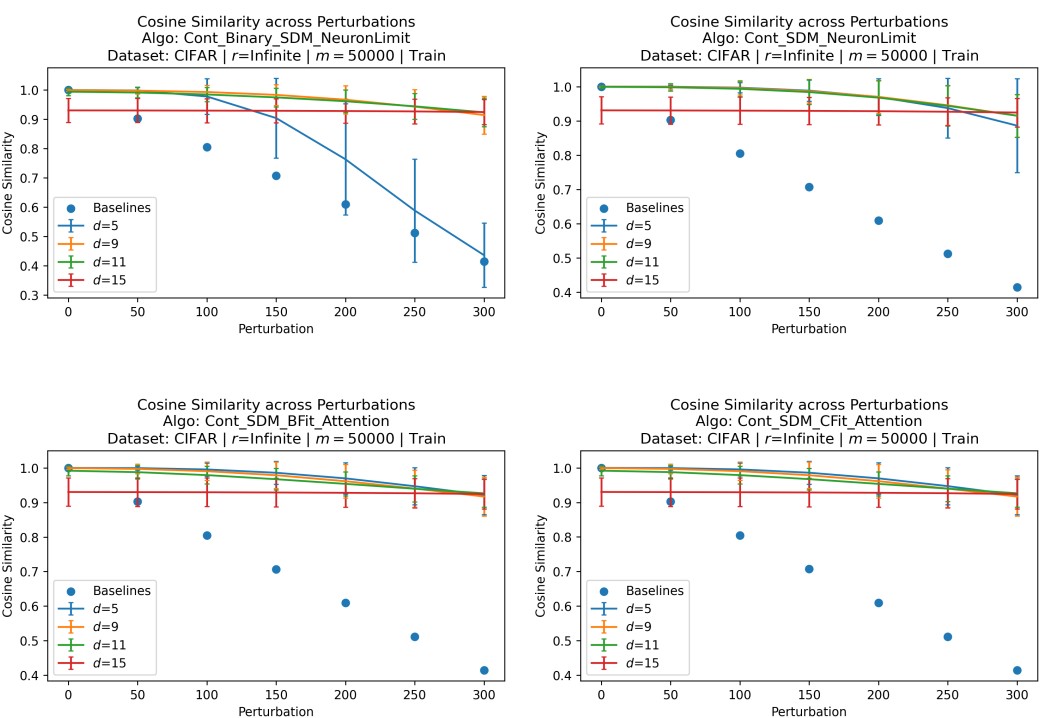

Figure 27: The gains in performance from learning the projection matrix apply to CIFAR10 in addition to MNIST with similar convergences between algorithms.

In Figures 28 and 29 we show how performance changes as a function of neuron number. Infinite neurons means that we do not enforce any integer value on the decimal places of precision in the weightings for each pattern, allowing maximum information in how each is relatively weighted that is analogous to Attention. The color scheme across plots is consistent but we do not show neuron numbers when there are too few for even the largest expected circle intersection to contain at least one neuron. We show each plot for the three most performant Hamming radii, $d \in \{5, 9, 11\}$ for each row with the columns corresponding to "Continuous Binary SDM Limited Neurons" and "Continuous SDM Limited Neurons", respectively.

These results confirm that reducing the number of neurons harms performance in a gradual way. As an additional observation, in the MNIST and CIFAR figures, the bottom left plot where $d = 11$ and the binary SDM circle intersection is used it is surprising the orange $r = 3e + 09$ performs above baseline, while it fails to for the continuous circle intersection in the same row on the right. The same effect can be seen in the middle row with the green $r = 1e + 13$. The circle intersections in continuous space are smaller such that enforcing a finite number of neurons has a larger effect on precision of its circle intersection weightings. In support of this, comparing rows we can see in general how for the same $d$ there are fewer valid numbers of neurons for the continuous circle intersection algorithm column on the right.

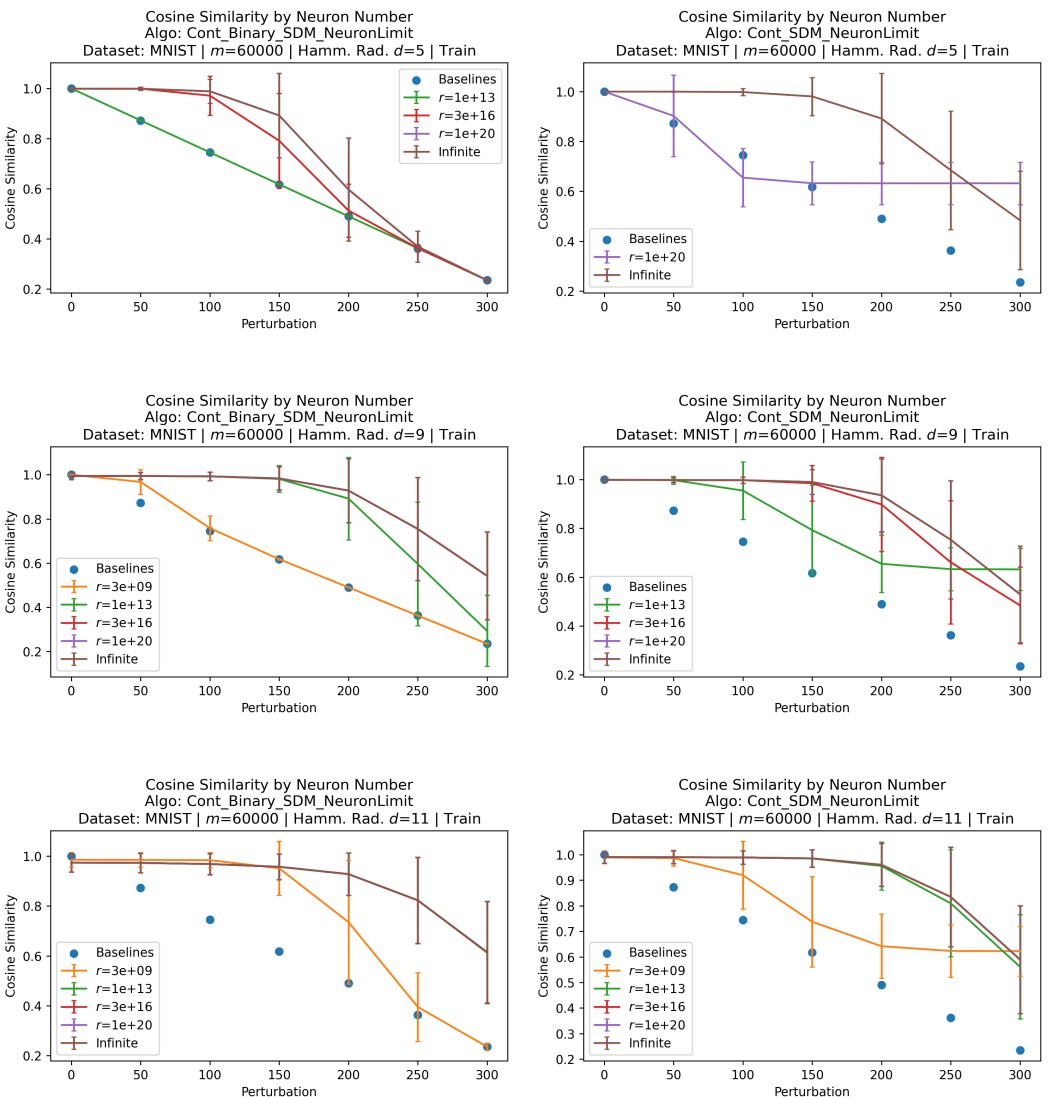

Figure 28: MNIST convergence across different numbers of neurons. We show each plot for the three best performing Hamming radii, $d \in \{5, 9, 11\}$ for each row with "Continuous Binary SDM Limited Neurons" in the left column and "Continuous SDM Limited Neurons" in the right. When a given neuron number line cannot be seen it is always hidden behind the best performing line that is visible in the plot.

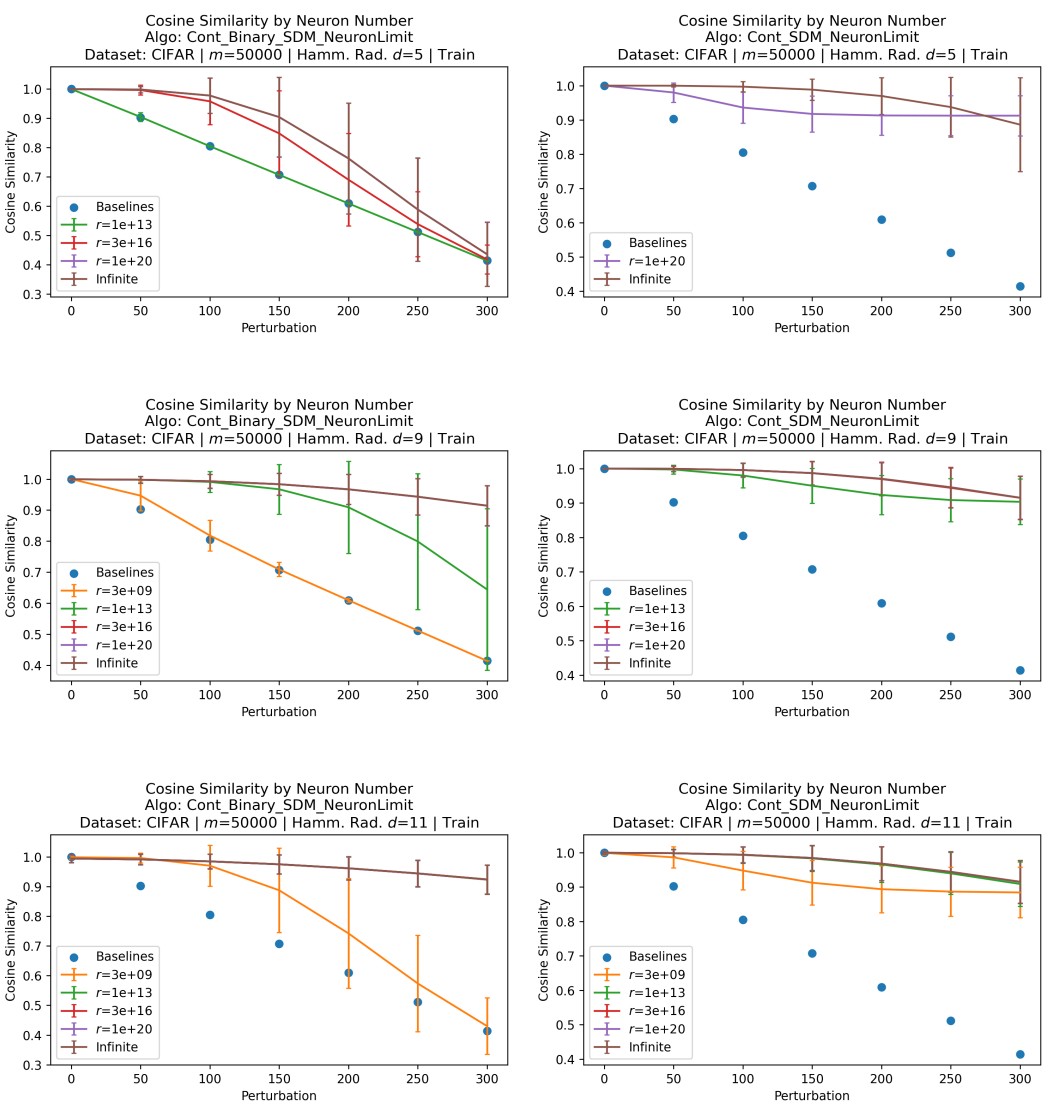

Figure 29: CIFAR convergence as a across different numbers of neurons. We show each plot for the three best performing Hamming radii, $d \in \{5, 9, 11\}$ for each row with "Continuous Binary SDM Limited Neurons" in the left column and "Continuous SDM Limited Neurons" in the right. When a given neuron number line cannot be seen it is always hidden behind the best performing line that is visible in the plot.

For the sake of completeness, Figures 30 and 31 show the same plots as Figures 26 and 27 but for the test datasets instead of train ones that the projection matrices were fit to. Performance is clearly worse showing that the projection matrix does not generalize well. However, we restate that learning the projection matrices here was a proof of concept and not optimized to avoid overfitting. In addition, the projection matrix is capable of handling $m = 60,000$ for MNIST and $m = 50,000$ for CIFAR training datasets, approximately $50\mathrm{x}$ more patterns in the space than in the $m = 1,024$ setting used for the raw MNIST images of Fig. 23.

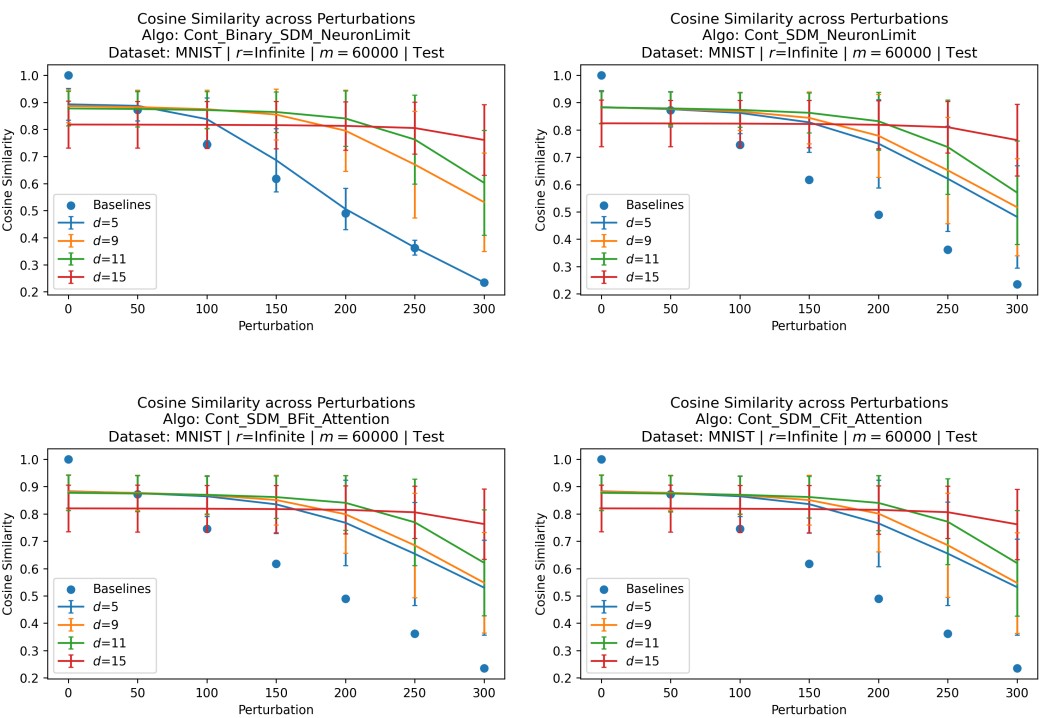

Figure 30: MNIST Test dataset performance showing the projection matrix does not generalize well. The number of patterns $m$ corresponds to the number that this projection matrix was trained on, not the number of patterns contained in the test set which is 10,000.

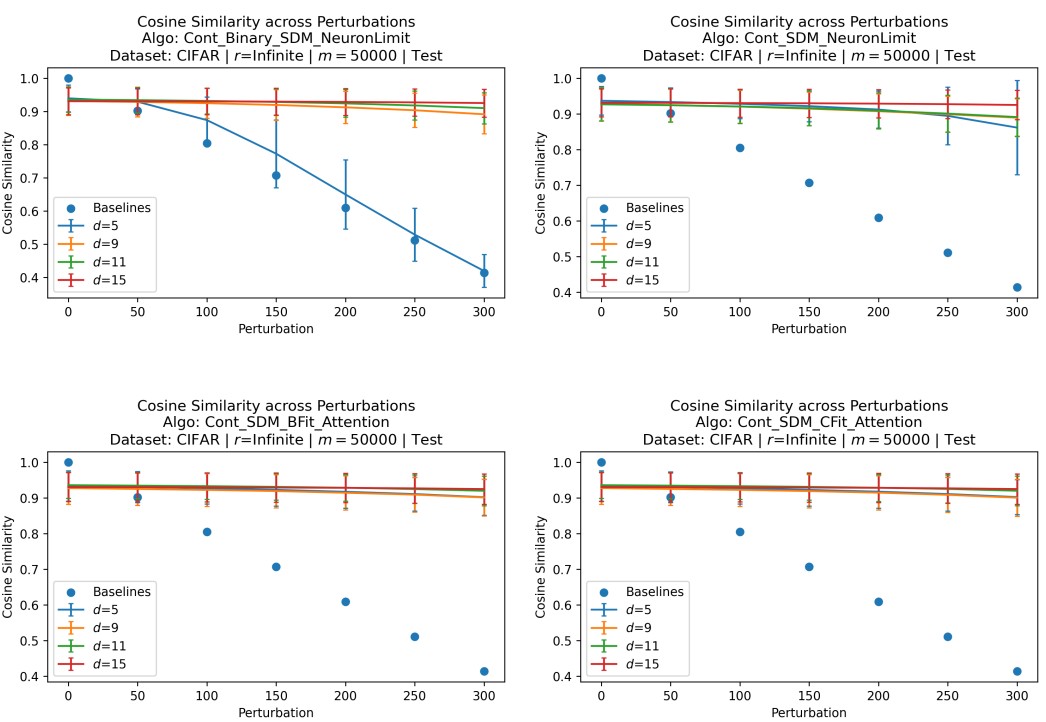

Figure 31: CIFAR Test dataset performance showing the projection matrix does not generalize well. The number of patterns $m$ corresponds to the number that this projection matrix was trained on, not the number of patterns contained in the test set which is 10,000.