# OpenReview forum: "Attention Approximates Sparse Distributed Memory"
_NeurIPS.cc/2021/Conference — NeurIPS 2021 Poster_

### Official Review · Reviewer_PZYv · 2021-07-16

**Rating:** 5
**Confidence:** 3

**Summary:**

The authors showed that the attention mechanism used in Transformers is related to the Sparse Distributed Memory, a classical computational model of associative memory in the brain.

**Limitations And Societal Impact:**

I don’t see clear limitations and potential negative societal impact beyond what’s mentioned above.

**Main Review:**

Originality
This result bears some similarity with the Ramsauer 2020 paper that connects “modern Hopfield Networks” with Attention. But other than that, it appears original.

Quality
The authors clearly spent a lot of time on this submission. The Supplemental Materials is 55 page long.

Clarity
The description of SDM seems unnecessarily complex and somewhat confusing. I thought I understood SDM when I read Kanerva’s 1992 book, but I was kind of confused after reading the authors’ description. If SDM is almost equivalent to Attention, and Attention can be described in one line (line 122-123), why does it take two pages to describe SDM?

Some description of SDM’s relationship to the brain seems unnecessarily long and not particularly insightful. For example, section 5 SDM’s Mapping to Cerebellum. It’s not clear what this section adds.

Significance
It is not clear what significant insights we can obtain by relating SDM with Transformer Attention. Many models are similar to one another if we make some approximations and assumptions. Attention without softmax is equivalent to a neural network with a single hidden layer and no nonlinearity.


**Time Spent Reviewing:**

2

---

> ### Author Response · Authors · 2021-08-10
> **Response to Reviewer**
>
> We thank the reviewer for their time, comments, and feedback. We kindly ask the reviewer to reconsider his/her score based on our replies below. Replying to each comment in turn.
>
> - **”Clarity - The description of SDM seems unnecessarily complex and somewhat confusing. I thought I understood SDM when I read Kanerva’s 1992 book, but I was kind of confused after reading the authors’ description. If SDM is almost equivalent to Attention, and Attention can be described in one line (line 122-123), why does it take two pages to describe SDM?”**
>
> Thank you for this suggestion. It is true that a one line description of SDM is possible, for example Equation 8, however, we thought this would fail to provide background on SDM to unfamiliar readers and the necessary framework to connect SDM to Attention. This connection relies upon explaining the read and write circle intersections that enable SDM to approximate the exponential of Attention’s softmax function.
>
>
> We will clarify our discussion of SDM and have gone about:
> - Adding a new figure that should provide better intuition for the SDM read and write operations and how these relate to the pattern query circle intersection.
> - Better explaining and expanding upon the definitions of “optimal” Hamming distances that SDM can use.
> - Focussing more on the 64 dimensional setting that is used by Transformer Attention to make the approximation more immediately apparent.
> - Making general improvements to writing style such as removing run-on sentences and extraneous information that has saved half a page in space.
>
> We would be grateful if the reviewer has any additional ways the readability of our work can be improved.
>
>
> - **”​​Clarity - Some description of SDM’s relationship to the brain seems unnecessarily long and not particularly insightful. For example, section 5 SDM’s Mapping to Cerebellum. It’s not clear what this section adds.”**
>
> We thank the reviewer for the criticism. We agree that our current section 5 needs to be rewritten to be more succinct and clear about its insights. We will revise the section significantly.
>
> To clarify the purpose of this section, we believe SDM’s mapping to the cerebellum is one of the crucial reasons why SDM’s relationship to Attention is valuable. This mapping provides SDM, and by proxy Attention, with biological plausibility and introduces the hypothesis that Attention and Transformers more broadly have been a dominant neural network architecture because they are performing the same crucial cognitive operations of cerebellar-like architectures that are ubiquitous across biological organisms.
>
> The paper “Large Associative Memory Problem in Neurobiology and Machine Learning” by Krotov and Hopfield (2021), highlights a desire in the research community for biological interpretability and attempts to provide an interpretation for the modern continuous Hopfield Network that has been mapped onto Attention. However, SDM goes further in both outlining precisely how different cells in the cerebellum implement each of SDM’s architectural requirements and being free of the Hopfield Networks’s weight transport problem. This mapping to specific cell types and their connectivity goes beyond the low bar of simply showing how SDM can be written as a single hidden layer feedforward network and assuming that this is biologically plausible. Granule cells can compute Hamming distances and store patterns, climbing fibers can write in patterns via LTP/LTD, and Purkinje cells can perform the summation and binarizing operations.
>
> While revising this section, we will broaden the biological link and its significance. Note that the Drosophila mushroom body is highly similar to the cerebellum and the previous cell labels for each function can be replaced with Kenyon cells, dopaminergic neurons, and mushroom body output neurons, respectively.
>
> Further, we will better discuss the limitations of the biological link. We have acknowledged in our response to Reviewer #1 (ID: XvDB) the need to better highlight the limitations of the cerebellar mapping in the main text and what remains to be explained by this model.
>
> - **”Significance - It is not clear what significant insights we can obtain by relating SDM with Transformer Attention. Many models are similar to one another if we make some approximations and assumptions. Attention without softmax is equivalent to a neural network with a single hidden layer and no nonlinearity.”**
>
> We appreciate the reviewer's criticism and concern. These are indeed important points and need to be addressed directly to better place our paper in the literature.
>
> Regarding the larger purpose of connecting SDM specifically to Attention, we point to both the model’s aforementioned biological plausibility, and its formalism that provides a geometric perspective, convergence proofs, and interpretations for the Transformer architecture more broadly. In addition, SDM retroactively predicted Query-Key Normalization (leading to small empirical gains in Transformer performance) and may lead to additional improvements to Transformer architectures that are being explored as future work. These explorations include:
> - Improvements to the long term memory functionality of the Transformer’s Feed-Forward layers.
> - New ways of learning keys and values in higher layers of stacked Attention/SDM.
> - Leveraging SDM’s connection to Vector Symbolic Architectures to introduce novel inductive biases.
>
> In addition, SDM provides a new perspective on previous work by being a generalization of Hopfield Networks and can help interpret the modern Hopfield Network modifications that improved it and enabled its connection to Attention (covered in Appendix Sections B.5 and B.6). We will clarify these insights provided by SDM in our main text.
>
> Finally, we believe it is incorrect that Attention can be written without its softmax as a single hidden layer feedforward network. For Attention, without its softmax operation and no activation function, to be written as the single hidden layer, each hidden unit would need to represent an input token with its input weights equal to the key vector and output weights equal to the value vector. However, this formulation would require the hidden units and their weights to dynamically update for every change in input tokens and would not be trained via backpropagation. This dynamic, input dependent modification of the hidden units is why Attention without the softmax operation has been shown to map to Fast Weights and RNNs rather than feedforward models (“Linear Transformers Are Secretly Fast Weight Programmers” by Schlag et al. (2021) and “Transformers are RNNs: Fast autoregressive transformers with linear attention.” by Katharopoulos et al. (2020)).
>
> In addition, these papers show that while reducing computational cost, removing the softmax operation harms performance. Meanwhile, SDM not only shows how Attention can be written as a feedforward model (using its neurons to store superpositions of input tokens during training) but also reveals that through simple binary read and write operations, (the neuron is either within Hamming/cosine distance or it’s not) the softmax function emerges and need not be removed.

---

> > ### Comment · Reviewer_PZYv · 2021-08-25
> > **More discussion points**
> >
> > I thank the authors for their response. I have also had the chance to read through the other reviews and their responses. I'm glad to read that the authors will clarify the description of SDM, its connection to Attention, and the biological relevance part. I have increased my score from 4 to 5 because of this. I also think I understand the manuscript better now.
> >
> > I should clarify that when I said "Attention without softmax is equivalent with a single hidden layer feedforward network with no nonlinearity" indeed I meant it's equivalent to a synaptically feedforward network with plasticity (or fast weights). Thanks for pointing that out.
> >
> > I think some of my key concerns are not addressed, at least to the level I understand the response. I'd appreciate if the authors can answer several questions below:
> >
> > (1) In SDM, the key matrix is random and fixed (right?), while in Attention, the key matrix is dynamically generated based on the inputs. How did the authors resolve this discrepancy in their approximation?
> >
> > (2) About SDM's comparison to cerebellum. The authors point out as a merit of SDM that there is no need for weight transport. Isn't that because SDM assumes the Key matrix is random and fixed?
> >
> > (3) Transformers process all queries (corresponding to input tokens) in parallel. SDM and all biologically similar architectures (cerebellum, mushroom body) essentially processes one query at a time. How did the authors resolve this discrepancy in their approximation?
> >
> > Minor issues:
> > Line 163, authors say " we test every SDM variant in comparison to Attention in an autoassociative retrieval task on random patterns and the MNIST dataset (Appendix D.8)." But there's no Appendix D.8.

---

> > > ### Author Response · Authors · 2021-08-27
> > > **Response to Reviewer**
> > >
> > > # Reply to Reviewer
> > >
> > > We have many thanks for the reviewer! Thanks for taking the time to read all of our reviews and discuss them. Thanks also for deciding to raise your score and for the additional follow up questions.
> > >
> > > - **"(1) In SDM, the key matrix is random and fixed (right?), while in Attention, the key matrix is dynamically generated based on the inputs. How did the authors resolve this discrepancy in their approximation?"**
> > >
> > > The reviewer is correct in that there is a discrepancy, which we should have clarified.
> > >
> > > We resolve this discrepancy in our approximation through the following reasoning. The number of neurons and how well they cover the pattern manifold will determine how well SDM functions. Eq. 4 of our work shows how SDM approximates the exponential function in Attention by counting the expected number of neural addresses that exist in the circle intersection between the query and patterns. Our approximation becomes more precise as more neurons are in this intersection either through more neurons existing in the space or the neuronal addresses better covering the pattern manifold. We expect learned keys (neural addresses) to populate the data manifold, making our approximation valid.
> > >
> > > Our approximation works even in the case where keys (neural addresses) are randomly assigned. We tested this approximation with experiments by varying both the number of neurons and how well they covered the pattern manifold in Appendix B.7. Both tests changed the precision of the circle intersection calculation but showed SDM's approximation holds. In changing the number of neurons, we tested two SDM algorithms, one assumed that a neuron exists at every position in the binary vector space (Algorithm "Pattern Based SDM Infinite Neurons"), the other assumed a restricted number of neurons (Algorithm "Pattern Based SDM Limited Neurons"). These algorithms did have performance differences but they were minor as discussed in the Appendix.
> > >
> > > In changing the randomness of the patterns,  experiments of Appendix B.7 assumed the neurons are randomly distributed and investigated how random patterns and the correlated MNIST dataset perform with SDM. SDM is able to store MNIST digits, however, it performs much worse than with the random patterns.
> > >
> > > Finally, we expect learned keys (neural addresses) to populate the data manifold because in an extension to SDM  it is put forward that the neuronal addresses can be learnt by a competitive learning algorithm ("Comparison Between Kanerva's SDM and Hopfield-type Neural Networks" by Keeler (1988) and  "Feature Discovery by Competitive Learning" by Rumelhart and Zipser (1985)). Additionally, if the addresses can be learnt, and the Hamming distance used for read and write operations can dynamically scale so that it writes to a constant number of neurons (like it would do in expectation with random neuron addresses and patterns) then the original capacity and performance guarantees of SDM will still hold.
> > >
> > > We will add a discussion of these points. We will also add a new figure to the paper that provides better intuition for the SDM read and write operations and how these relate to the pattern query circle intersection. We have made this figure accessible on our anonymous Github repo at: https://github.com/attention-sdm/attention-approximates-sdm/blob/main/reviewer_reply/SDM_Neuron_and_Pattern_Perspectives.png.
> > >
> > >
> > > - **"(2) About SDM's comparison to cerebellum. The authors point out as a merit of SDM that there is no need for weight transport. Isn't that because SDM assumes the Key matrix is random and fixed?"**
> > >
> > > Yes this is correct. We will make this explicit in the text where it is discussed.
> > >
> > > We also want to acknowledge that the competitive learning algorithm for the neuronal addresses to cover the pattern manifold mentioned in our answer to Q1 do not require symmetric keys and values. Therefore, it also avoids the weight transport problem.
> > >
> > > - **"(3) Transformers process all queries (corresponding to input tokens) in parallel. SDM and all biologically similar architectures (cerebellum, mushroom body) essentially processes one query at a time. How did the authors resolve this discrepancy in their approximation?"**
> > >
> > > We thank the reviewer for bringing this point up. The parallelism of transformers does not affect our approximation or results. Our approximation acts separately on each query. However, the parallelism vs sequential processing is an important distinction between biologicaly and artificial systems that we will emphasize.
> > >
> > > We also want to push back slightly that Transformers process all inputs in parallel. At the hardware level, there is sequential computation of each query. This is especially the case for the more efficient TransformerXL that sequentially processes a sequence, re-using all previous computations for each input that remains in its receptive field ("Transformer-XL: Attentive Language Models Beyond a Fixed-Length Context" by Dai et al. 2019).
> > >
> > > - **"Minor issues: Line 163, authors say " we test every SDM variant in comparison to Attention in an autoassociative retrieval task on random patterns and the MNIST dataset (Appendix D.8)." But there's no Appendix D.8."**
> > >
> > > Thank you for spotting this error and apologies for it. Appendix D.8 was meant to refer to the experimental results in Appendix B.7 and we have rectified this error.

---

> > > > ### Comment · Reviewer_PZYv · 2021-09-03
> > > > **response of response of response of response**
> > > >
> > > > Thanks for the response again.
> > > >
> > > > I understand that under certain assumptions, learned and data-dependent keys (Transformers, etc.) can be similar to random keys (SDM), but I'm not convinced that this approximation holds well in many important use-cases. For example, when the number of keys is low.
> > > >
> > > > About the Transformer processing queries all together point. Agreed, Transformer-XL is closer to processing one query at a time. There are still some differences (e.g, TransformerXL needing concatenation of previous and current segments), but they are perhaps not fundamental.

---

> > > > > ### Author Response · Authors · 2021-09-08
> > > > > **Reply to Reviewer**
> > > > >
> > > > > We thank the reviewer for their continued engagement, discussions, and improvements to the paper.
> > > > >
> > > > > * **"I understand that under certain assumptions, learned and data-dependent keys (Transformers, etc.) can be similar to random keys (SDM), but I'm not convinced that this approximation holds well in many important use-cases. For example, when the number of keys is low."**
> > > > >
> > > > > The reviewer is correct that we should make the limits of our approximation clear. Therefore, we propose running experiments like those in Appendix B.7 on the convergence abilities of SDM and Attention for random patterns and MNIST but varying the number of neurons and quantifying the approximation accuracy. We will also apply this analysis to the more complex CIFAR dataset with and without learned projection matrices that we proposed performing to Reviewer #1 (ID: XvDB).
> > > > >
> > > > > Utilizing the learned projection matrices will be particularly useful for simulating the performance of realistic versions of SDM where the neurons can learn the data manifold. This should significantly reduce the number of neurons that are necessary for the approximation to hold well.
> > > > >
> > > > > At risk of providing too much information, it is worth nothing that our Appendix already contains initial results showing how SDM performance changes as a function of neuron number and how well the approximation to Attention holds. For example, Figure 21 shows theoretically how SDM performances change as a function of neuron number. This theory is validated by the experiments of Figs. 18-20, most notably Fig. 19 where the Hamming distance is d=405 and the top right figure where there are $r=10^6$ neurons fails while all of the other approaches with $r=2^{1000}$ neurons succeed, as predicted by the blue line in Figure 21.
> > > > >
> > > > > On one hand, this result shows that SDM is not a good approximation to Attention when there are few neurons. This would highlight the extent to which our deep learning models can have a higher memory capacity/performance than the brain. On the other hand, there are three pieces of evidence supporting the quality of SDM's approximation:
> > > > >
> > > > > First, real world data will be correlated, existing on a lower dimensional manifold. Extensions of SDM enable neurons to learn this manifold. Moreover, the very fact the data exists on a lower dimesional manifold means the neurons can cover it more easily than if they were random patterns distributed throughout the whole vector space.
> > > > >
> > > > > Second, there are many neurons ready to learn this data manifold: 3/4rds of all neurons in the human brain are granule cells in the cerebellum (~60 billion of a total ~85 billion) ("Ch. 7 Cerebellum". In Shepherd GM. The Synaptic Organization of the Brain). SDM hypothesizes that these granule cells are its neurons (keys).
> > > > >
> > > > > Finally, the different lines on Figure 21 show that the number of neurons for good SDM performance depends upon the chosen Hamming distance. When $d=447$ (the signal to noise optimal Hamming distance), having more than $10^8$ neurons has steeply diminishing returns, this is supported by Figure 19 where the neuron limited SDM and Attention perform similarly. Does Attention's beta coefficient approximate Hamming distances where there are similar diminishing returns to additional neurons? Figure 3 provides weak support for this idea by showing that pre-trained GPT2 Attention and SDM largely agree on the Hamming distance (Beta coefficient) used.
> > > > >
> > > > > The results from the Figures mentioned and present experiments in the Appendix are incomplete and need the further analysis we have proposed. However, the above reasons provide avenues for how the approximation can hold.

---

### Official Review · Reviewer_sTq7 · 2021-07-16

**Rating:** 7
**Confidence:** 4

**Summary:**

This paper shows that Transformer's Attention closely approximates Sparse Distributed Memory (SDM), a biologically plausible associative memory model, under certain conditions on the data normalization. The authors first provide a review of Kanerva’s SDM. Then, they show that Attention approximates SDM theoretically and empirically via the pretrained GPT2 architecture. Here, the intersection between 2 Hamming circles in SDM is approximated by an exponential function of key and query, which forms the softmax function in Attention.  This close relationship between Attention and SDM leads to a hypothesis that Attention is performing the same associative memory operations as the cerebellum. Moreover, there are discussions on related works including Hopfield Network, Memory-augmented Neural Networks and Kanerva Machine.

**Limitations And Societal Impact:**

Yes

**Main Review:**

While Attention has become an important mechanism in deep learning, it remains unclear why it works so well. The finding of this paper may provide one explanation for Attention’s success. Prior works have pointed out the relationship between Attention and Hopfield Network, Fast-weight or Associative Memory. SDM has been used as motivations for memory-augmented neural networks.  However, this is the first time that someone have shown the close relationship between Attention and SDM both theoretically and empirically. Key mathematical techniques are the approximation of binomial coefficients using exponentials (equation (4)) and transforming the non-continuous Hamming distance to a continuous version including the cosine similarity (equation (6)). Those techniques, yet simple, are crucial to building a bridge between SDM and Attention.
The paper is well-written and easy to follow. People who are not familiar with the topic can still get the key ideas. Figures are clear and informative.
Questions to the authors:
1.	SDM is well-known for the ability of iterative reading, i.e. to feed the previous output of the read operation into the read operation again to get a better-match pattern until convergence. Does Attention have this ability? If yes, can it be shown theoretically or via experiments?
2.	Section 4 shows that components of a Transformer block are closely related to SDM. However, Transformer encoder and decoder consist of many blocks stacked together. This makes the model more powerful. How can this be explained in terms of SDM?

**Time Spent Reviewing:**

4h

---

> ### Author Response · Authors · 2021-08-10
> **Response to Reviewer**
>
> We thank the reviewer for their time, comments and questions. We have re-copied the reviewer’s questions with our replies below:
> - **“SDM is well-known for the ability of iterative reading, i.e. to feed the previous output of the read operation into the read operation again to get a better-match pattern until convergence. Does Attention have this ability? If yes, can it be shown theoretically or via experiments?”**
>
> In short, yes, you can do the same with Attention.
>
> On the experimental front, we tested iterative convergence using the original Attention operation in Appendix Section B.7 for random patterns and MNIST showing both that it is possible and approximates the analogous SDM operations.
>
> However, very often Attention is capable of one step convergence. The reviewer acknowledged paper “Hopfield Networks Is All You Need” provides a detailed proof (Theorem 4) inherited from “On a model of associative memory with huge storage capacity” by Demircigil et. al (2017). Their result shows that indeed very often Attention is capable of one step convergence. We already state that Hopfield Networks “can be powerfully used in convergence proofs” on lines 288 and 289 of our paper but we will make this specific one-step convergence finding more explicit.
>
> As an aside, this iterative convergence is only useful in the auto-associative setting but Transformers are used for prediction tasks where they operate hetero-associatively. However, the fact they have strong convergence properties remains useful for their ability to transition to the next hetero-associative pointer.
>
> - **“Section 4 shows that components of a Transformer block are closely related to SDM. However, Transformer encoder and decoder consist of many blocks stacked together. This makes the model more powerful. How can this be explained in terms of SDM?”**
>
> The hierarchical stacking of SDM modules is important and indeed not captured in the current work. This is because, unlike in the traditional SDM setting where the pattern addresses (keys) and pattern pointers (values) are known in advance and written into memory, this cannot be done for layers of SDM beyond the first that will need to learn latent representations for its pattern addresses and pointers (keys and values). Transformer Attention learns its higher level keys and values by treating each input token as its own query to generate a new latent representation of the token that is then projected into keys and values. We will add a statement on this discrepancy with stacked Attention to our paper.
>
> We are exploring ways of stacking SDM and novel ways to learn these latent key and value representations that will be presented in future work. Below are two specific ideas that point towards the plausibility and opportunity of stacked SDMs. We will comment on these ideas in our paper.
>
> Abstractly, like the low vs high level features learnt by progressively higher layers of a convolutional network, we believe the same effect of hierarchy should enable higher layers of SDM to extract higher level features of the input. As a concrete example, the operations of SDM are related to the “Sparse Manifold Transform” of Chen et al. (2018), which shows theoretically how a doubly stacked SDM-like model operating on video data can use the first layer to extract spatial redundancies from each still image and the second layer to extract temporal redundancies across images. The first SDM level of the sparse manifold transform can approximately be thought of as having multiple queries from an image simultaneously read from neurons that have learnt to cover a data manifold and operate auto-associatively. The second layer of SDM could then learn the hetero-associative mapping between sets of queries that correspond to the transitions between video frames.
>
> More broadly, we believe thinking more about how to learn the latent keys and values for the higher levels of SDM could present new Transformer improvements. The Transformer learns its higher level keys and values by treating each input token as its own query. We believe the fundamental breakthrough with the recent Performer architecture is that it uses a reduced set of latent keys and values in its “latent Transformer” (“Rethinking Attention with Performers” by Choromanski et al. (2020)). This highlights the arbitrariness of the original Transformer solution using queries, keys and values for every token at every layer and presents opportunities for new ways to learn the keys and values of higher layers.

---

### Official Review · Reviewer_1c9u · 2021-07-16

**Rating:** 6
**Confidence:** 3

**Summary:**

The authors focus on attention mechanisms in neural networks, and show that under certain conditions the attention mechanism in Transformer networks can approximate a type of associative memory.


**Main Review:**

Main merits:
- The work presented is novel and it seems solid.

Main limitations:
- The paper may be organized better, as it is a bit hard to read in its present form.
- A more thorough experimental validation would be useful..

**Time Spent Reviewing:**

0.75

---

> ### Author Response · Authors · 2021-08-10
> **Response to Reviewer**
>
> We thank the reviewer for their time, comments, and feedback. Replying to each comment in turn.
>
> - **”The paper may be organized better, as it is a bit hard to read in its present form.”**
>
> We are planning to make a number of modifications to the paper that may help with readability. These include:
> - Adding a new figure that should provide better intuition for the SDM read and write operations and how these relate to the pattern query circle intersection.
> - Better explaining and expanding upon the definitions of “optimal” Hamming distances that SDM can use.
> - Focussing more on the 64 dimensional setting that is used by Transformer Attention to make the approximation more immediately apparent.
> - Making general improvements to writing style such as removing run-on sentences and extraneous information that has saved half a page in space.
>
> We would appreciate it if the reviewer can add any additional ways the readability of our work can be improved.
>
> - **”A more thorough experimental validation would be useful.”**
>
> With regards to additional experimental validation, Reviewer #1 (ID: XvDB) stated concerns around SDM’s ability to handle datasets more complex than MNIST. We have replied to these concerns and proposed running further analyses with the CIFAR10 dataset. We are happy to consider other experiments suggested by the reviewer.

---

### Official Review · Reviewer_XvDB · 2021-07-20

**Rating:** 8
**Confidence:** 3

**Summary:**

This paper shows that Transformer Attention approximates Sparse Distributed Memory (SDM) model under given conditions.  It further confirms that those conditions are satisfied on pre-trained GPT2 Transformer models.  In addition to theoretical analysis, some results are provided using various datasets including MNIST.  Furthermore, there is a section on biological interpretation of Attention and SDM.

**Main Review:**

This paper shows that the relationship between SDM and Attention exists because SDM’s read operation, when approximated as an exp sum of exponentials, it actually approximates softmax rule used in Attention.   They use Query-Key Normalized transformer to show the relationship to SDM, and original GPT2 models to generalize the results.

The connection between SDM and Attention was insightful, and nicely presented.  Evaluation was very limited, which his concerning as one of the difficulties of SDM-based models is precisely lack of accuracy when it comes to more complex image-based datasets beyond MNIST.   The idea that SDM/Attention model could be better interpreted through biology is very attractive, but that section in the paper seemed to be quite a stretch, and did not present convincing evidence that this is indeed the case.


**Time Spent Reviewing:**

1

---

> ### Author Response · Authors · 2021-08-10
> **Response to Reviewer**
>
> We thank the reviewer for their time, comments, and feedback. Replying to each comment in turn.
>
> - **”Evaluation was very limited, which is concerning as one of the difficulties of SDM-based models is precisely lack of accuracy when it comes to more complex image-based datasets beyond MNIST.”**
>
> The reviewer is correct that more complex datasets, specifically those with more correlated data, reduce SDM’s memory capacity and error tolerance. This result can even be seen using MNIST in the analysis of Appendix Section B.7, where the memory capacity for MNIST is significantly worse than random, uncorrelated patterns.
>
> However, the ability for SDM to approximate Attention is independent of the dataset and a function of the SDM circle intersection’s ability to approximate the exponential and, by proxy, softmax. This in turn depends upon the dimensionality of the vector space and the size of the Hamming distance for reading and writing. Our work shows not only that under a range of hypothetical conditions this approximation holds but also that these conditions are the ones that trained GPT Transformer models operate under when working with highly complex natural language datasets.
>
> The above answer still fails to address the question of SDM’s ability to store more complex datasets. If Transformer Attention can do it but SDM struggles to store even MNIST then what gives? This distinction lies in the fact that Transformer Attention is able to learn projections for its keys, values and queries. In our validation experiments, we did not learn representations for the MNIST digits that could have de-correlated them and Attention also struggled as much as SDM did.
>
> We will test SDM with the more complex CIFAR10 dataset. We believe that either: we use the raw pixels as input where then both SDM and Attention fail to store it well; or we learn key value projections and then both SDM and Attention perform well. This outcome will further reinforce that Attention approximates SDM.
>
> - **”The idea that SDM/Attention model could be better interpreted through biology is very attractive, but that section in the paper seemed to be quite a stretch, and did not present convincing evidence that this is indeed the case.“**
>
> We thank the reviewer for his/her criticism and incentivizing us to articulate our points better. We certainly would not want to mislead our readers. In response to this concern, we will better highlight limitations of this biological interpretation in the main text. While we acknowledge there are details that need further investigation, a number of these have been addressed in extensions to SDM and there remain striking convergences between specific cell types and SDM’s non-trivial architectural requirements.
>
> Moreover, we believe the SDM model makes more precise and testable claims on its biological plausibility than many other models in deep learning that assert biological plausibility. For example, the paper “Large Associative Memory Problem in Neurobiology and Machine Learning” by Krotov and Hopfield (2021), highlights a desire in the research community for biological interpretability and attempts to provide an interpretation for the modern continuous Hopfield Network that has been mapped onto Attention. SDM goes further in both outlining precisely how different cells in the cerebellum implement each of SDM’s architectural requirements and being free of the Hopfield Networks’s weight transport problem. This mapping to specific cell types and their connectivity goes beyond the low bar of simply showing how SDM can be written as a single hidden layer feedforward network and assuming that this is biologically plausible. Granule cells can compute Hamming distances and store patterns, climbing fibers can write in patterns via LTP/LTD, and Purkinje cells can perform the summation and binarizing operations.
>
> The connection between SDM and the cerebellum first discovered by Kanerva needs extensive context behind the neural wiring of the cerebellum. Due to a lack of available space to provide this background, it and a discussion of its limitations were placed in the Appendix Section D.1. These limitations include the sparse dendritic connections of Granule cells and needing better explanations for the functions of the three inhibitory interneurons: Golgi, Stellate and Basket cells. In addition, better explaining the inputs to the mossy and climbing fibers and outputs from the Purkinje cells. We propose moving a sentence on these limitations from the Appendix into Section 5 of the main text that covers the cerebellar relationship.
>
> However, a number of limitations are explained by extensions to SDM. The Hyperplane design of Jaeckel in “A class of designs for a sparse distributed memory” (1989) addresses the sparse granule dendrites. Meanwhile, in "Comparison Between Kanerva's SDM and Hopfield-type Neural Networks" by Keeler (1988) highlights a suggestion first made by Marr that the Golgi interneuron may be dynamically scaling the Hamming distance to implement competitive dynamics. Other limitations, such as the functioning of the Stellate and Basket cells, we believe we have recently found answers for that can significantly improve the representational capacity of SDM and will be outlined in future work. We will provide a review of these extensions in our revised work.
>
> While there are details of the cerebellum and SDM's potential implementation that still need to be further investigated (such as the precise timing of climbing fiber firing to induce LTP vs LTD), we hope the reviewer is as intrigued as us by the fact that the three way convergence between parallel fibers, climbing fibers and Purkinje cells in the cerebellum. Furthermore, the number of granule cells and the extent of their parallel fiber branching that is a good candidate for the Hamming distance calculation each neuron must perform. These convergences with the non-trivial requirements for SDM to be implemented as a circuit are highlighted in Figures: 31, 33, and 34.

---

### Decision · Program_Chairs · 2021-09-27

**Decision:**

Accept (Poster)

**Comment:**

This paper shows connections between the attention function and sparse distributed memory. The insights are interesting and could be useful for those who want to understand why it works so well as well as those who want to extend it further. All reviewers generally agree that this is a good paper. Reviewer PZYv has concerns about the clarity of the paper and the significance of the insights. While I agree that the writing could be improved to make it easier to read for general readers (which the authors promised they will address in the version), I believe that the insights from this paper is useful. The authors also promised to add more discussions based on feedback from Reviewer PZYv, as summarized in their author response. I recommend accepting the paper.